# Two types of axonal muscarinic acetylcholine receptors mediate formation of saliva cocktail in the tick *Ixodes ricinus*

Cáinà Nìng[1], James J. Valdés [2,3], Lourdes Mateos-Hernández[1], Sabine Rakotobe[1], Lianet Abuin-Denis[1,4], Nadia Haddad[1], Lívia Šofranková[1,5], Mirko Slovák[6], Khalid Boussaine[7], Alison Cartereau[7], Emiliane Taillebois [7], Houssam Attoui[8], Helena Frantová[2], Veronika Urbanová [2], Tereza Kozelková [2,9], Filip Dyčka[9], Petr Kopáček [2], Ondřej Hajdušek [2], Radek Šíma [2,10,11], Jiří Týč[2], Tomáš Bílý [2,9], Martina Tesařová[2], Marie Vancová [2,9], Jan Perner [2], Steeve H. Thany[7,12] & Ladislav Šimo [1] ✉

Hard ticks depend upon an ability to precisely and dynamically regulate their saliva to successfully evade host haemostatic and immune defences during extended blood feeding. Although pilocarpine, an exogenous muscarinic acetylcholine receptor (mAChR) agonist, can stimulate salivation experimentally, the endogenous control of saliva secretion by acetylcholine remains poorly understood. Here, we identify and characterise two pharmacologically distinct mAChRs (type A and B) in the genome of the medically important tick *Ixodes ricinus*. Molecular dynamics simulations and targeted mutagenesis reveal that type B mAChRs exhibit an atypical muscarinic profile, suggesting unconventional receptor signalling. Combining immunolabelling, in vivo pharmacology, and proteomics, we show that specific central neurons interact with distinct salivary gland regions via mAChR type-specific axons, coordinating fluid and protein secretion through separate acini and likely acting upstream of a neuropeptide-dependent cascade. This previously unrecognised mechanism of neural control offers new insights into how ticks modulate their saliva advancing our understanding of vector-host interactions, with potential implications for disrupting pathogen transmission.

Hard ticks can secrete saliva into vertebrate hosts over days or weeks, depending on tick species and developmental stage[1]. Factors within tick saliva prevent host blood coagulation, suppress host defences, remove excess fluids and ions, and thus maintain homeostasis[2–4].

Salivary molecules are critical in facilitating pathogen transmission and are produced by specialised cells in the salivary glands (SGs)[2,5].

Female tick SG contains three acinus types (I, II, and III), each with distinct regulation and functions[6–10]. Salivation can be induced

[1]ANSES, INRAE, Ecole Nationale Vétérinaire d'Alfort, UMR BIPAR, Laboratoire de Santé Animale, Maisons-Alfort, France. [2]Institute of Parasitology, Biology Centre, Czech Academy of Sciences, České Budějovice, Czech Republic. [3]Centre Algatech, Institute of Microbiology, Czech Academy of Sciences, Třeboň, Czech Republic. [4]Animal Biotechnology Department, Center for Genetic Engineering and Biotechnology, Havana, Cuba. [5]Department of Animal Physiology, Pavol Jozef Šafárik University in Košice, Košice, Slovak Republic. [6]Institute of Zoology v. v. i, Slovak Academy of Sciences, Bratislava, Slovakia. [7]University of Orleans, P2E USC-INRAE 1328, Orléans, Cedex, France. [8]Ecole Nationale Vétérinaire d'Alfort, Anses, INRAE, Laboratoire de Santé Animale, VIROLOGIE, Maisons-Alfort, France. [9]Faculty of Science, University of South Bohemia, České Budějovice, Czech Republic. [10]Bioptic Laboratory, Plzeň, Czech Republic. [11]Sikl's Department of Pathology, Faculty of Medicine in Plzen, Charles University, and University Hospital, Plzeň, Czech Republic. [12]Institut Universitaire de France (IUF), Paris, France. ✉e-mail: ladislav.simo@vet-alfort.fr

experimentally by catecholaminergic and cholinergic agents[11–14], enabling the collection of large saliva volumes. While the role of dopamine is well established[15–17], the physiological relevance of cholinomimetics remains unclear.

Currently, in ticks, the muscarinic agonist pilocarpine is known to evoke strong, sustained salivary secretion in vivo, accompanied by elevated activity in SG-innervating nerves[11,18]. This response can be blocked by atropine[11], or by severing connections between the synganglion and SGs[19], indicating that axonal mAChRs mediate saliva secretion via a non-cholinergic secretomotor mechanism[20].

Here, we identified mAChRs in the axons of peptidergic neurons projecting to type II and III SG acini[21–23], revealing the long-sought cholinoceptive pathway[4,20] controlling tick salivation. We characterised two pharmacologically distinct mAChRs in *I. ricinus* using molecular dynamics, heterologous expression, and receptor-specific antibodies. We demonstrated in vivo that coordinated activation of both receptors is required for complete salivary secretion, including its dynamic and proteomic output. We also identified additional neurosecretory cells (NSC) expressing mAChRs, indicating broader cholinergic sensitivity within the synganglion. These findings reveal an integrated cholinergic control system in ticks, linking the synganglion and SGs to coordinate secretion.

## Results

### *Ir*-mAChR-A and -B act in an excitatory manner and exhibit distinct muscarinic profiles in heterologous systems

Genome and transcriptome searches identified one type A and one type B mAChR in hard ticks. Phylogenetic analyses placed *Ir*-mAChR-A as an ortholog to mammalian receptors, while *Ir*-mAChR-B clustered within the protostome-specific clade[24] (Fig. 1a). The insect-specific type C receptor[25] was absent in ticks.

In *Xenopus* oocytes, 0.5 µM acetylcholine (ACh) induced inward currents via both *Ir*-mAChR-A (−4.62 ± 3.8 µA) and *Ir*-mAChR-B (−0.63 ± −0.27 µA) (Fig. 1b, c). Muscarine produced similar responses (−2.79 ± 2.12 µA and 0.16 ± 0.09 µA, respectively), confirming both receptors as common agonist targets. By contrast, the mammalian non-subtype-selective muscarinic agonists methacholine and oxotremorine activated *Ir*-mAChR-A (−3.69 ± 2.86 µA and −4.15 ± 4.20 µA)[26,27] but not *Ir*-mAChR-B.

We previously showed that *Ir*-mAChR-A couples to the $G_{q/11}$ family of proteins in Chinese hamster ovary (CHO) cells, whereas *Ir*-mAChR-B could not be functionally expressed[8]. Here, codon optimisation with a Kozak sequence enabled *Ir*-mAChR-B expression in CHO cells, allowing analysis with aequorin and GloSensor™ assays (Fig. 1d–f). Aequorin assays revealed robust ACh- ($EC_{50}$ 0.68 µM) and muscarine- ($EC_{50}$ 6.6 µM) induced calcium mobilisation consistent with $G_{q/11}$ activation (Fig. 1d and Supplementary Fig. 1a). Co-expression of *Ir*-mAChR-B with human $G_{\alpha15(16)}$ enhanced sensitivity ~4.6 times for ACh ($EC_{50}$ 0.15 µM) and ~2.3 for muscarine ($EC_{50}$ 2.1 µM) (Fig. 1e and Supplementary Fig. 1b). GloSensor assays in human embryonic kidney (HEK293) cells further showed that *Ir*-mAChR-B couples to the $G_s$ pathway to elevate cAMP (ACh, $EC_{50}$ 2.7 µM) (Fig. 1f and Supplementary Fig. 1c), while also mediating partial inhibition of forskolin-stimulated cAMP (Supplementary Fig. 1d–f).

A screen of 37 cholinergic drugs with known affinity for mammalian mAChR subtypes (M1–M5), revealed common activators and blockers for both *Ir*-mAChR types, but type-specific ligands only for *Ir*-mAChR-A (Fig. 1g, h). Notably, pilocarpine selectively activated *Ir*-mAChR-A while the archetypal anticholinergic drug, atropine, inhibited only *Ir*-mAChR-A, consistent with reports that atropine antagonises type A mAChRs[8,28,29]—including all five mammalian mAChR subtypes[30]—but shows no[28,29,31] or little[32] antagonistic effects on invertebrate-specific type B muscarinic receptors.

### Active-site mutations in *Ir*-mAChR-A and -B destabilise receptor conformations and alter their interactions with atropine

Both *Ir*-mAChR types share the conserved G protein-coupled receptor (GPCR) seven transmembrane (TM) fold and are structurally similar to the human M1 receptor (*Hs*-M1, PDB 6WJC)[33]. Their N-terminal extracellular regions are disordered, with *Hs*-M1 shorter than *Ir*-mAChR-A, and considerably shorter than *Ir*-mAChR-B (Supplementary Fig. 2). Amino termini are critical for both GPCR export from the Golgi and membrane integration. Specifically, a conserved Y−S pair proximal to TM1[34] is present in *Ir*-mAChR-A (Y42-S43) but absent in *Ir*-mAChR-B, which instead carries a W77−Q78 pair conserved in *Hs*-M1 (W23-Q24, Supplementary Fig. 3a).

Comparison with the *Hs*-M1 active site that binds atropine[33], showed full conservation in *Ir*-mAChR-A but distinct substitutions in *Ir*-mAChR-B (Fig. 2a−c). To test their effects in silico, the four functionally divergent *Ir*-mAChR-B residues were individually introduced into the *Hs*-M1 model, and atropine coordination was analysed by classical molecular dynamics simulations (Fig. 2d). Simulations revealed that the most divergent substitution, A193I, caused the largest deviation of atropine (4.1 ± 0.6 Å, vs. 2.0 ± 0.4 Å in *Hs*-M1 WT). The Y106H substitution also induced large fluctuations (3.5 ± 0.3 Å). The remaining substitutions, S109C (3.1 ± 0.4 Å) and N382H (2.6 ± 0.5 Å), likewise produced higher deviations than the *Hs*-M1 WT (Fig. 2d). After ~300 ns, A193I and Y106H showed opposite shifts in atropine coordination.

Molecular dynamics simulations of the *Hs*-M1[Ir4B] variant, which incorporates all four *Ir*-mAChR-B-to−*Hs*-M1 substitutions (Fig. 2a−c) produced an average atropine deviation of 3.3 ± 0.3 Å after 15 ns compared to the *Hs*-M1 WT (2.1 ± 0.3 Å) (Fig. 2e). The *Hs*-M1[Ir4B] also produced structural changes in transmembrane helices and loops (Supplementary Fig. 3b). Residues 132−134 of intracellular loop 2 (IL2) showed the largest displacement, deviating up to 7.0 ± 0.6 Å after 200 ns vs 3.0 ± 0.5 Å in the *Hs*-M1 WT. By 450 ns, additional deviations appeared in residues 164−166 of TM4 and 179−193 of extracellular loop 2-TM5, which includes the A193I mutation (Supplementary Fig. 3b).

In silico analyses predicted that replacing the four conserved active-site residues of *Ir*-mAChR-A with those of *Ir*-mAChR-B (Y128H, S131C, A216I, and N510H) would destabilise atropine binding and reduce affinity, whereas introducing the reciprocal substitutions in *Ir*-mAChR-B (H160Y, C163S, I247A, and H765N) were predicted to enhance affinity. To test these predictions, we generated the corresponding constructs encoding *Ir*-mAChR-A[4B] and *Ir*-mAChR-B[4A] (Fig. 2f and Supplementary Fig. 2 and 3c), and compared their properties in CHO cells using nine cholinergic agonists at 45 µM (Fig. 2g, h), the maximal dose for *Ir*-mAChR-B in our assays (Fig. 1e). All ligands activated the *Ir*-mAChR-A WT, with similar potency to ACh (equivalent to 100%). Muscarine activation of the *Ir*-mAChR-A[4B] approximately doubled the response compared to ACh, whereas carbachol, arecoline, aceclidine, and methacholine elicited responses less than half of the ACh response (<50%), and pilocarpine and bethanechol were almost inactive (<10%) (Fig. 2g). In contrast, both muscarine and arecoline elicited activation of the *Ir*-mAChR-B[4A] with potency equal to or greater than that of ACh. Strikingly, methacholine and oxotremorine−previously selective for *Ir*-mAChR-A (Fig. 1b, g)−elicited responses in the *Ir*-mAChR-B[4A] nearly equivalent to ACh (~80−120%) (Fig. 2h).

We next examined dose-dependent inhibitory effects of atropine on WT and mutant *Ir*-mAChRs stimulated with muscarine or arecoline (Fig. 2i, j, and Supplementary Fig. 3d−h), the most potent activators of each mutant (Fig. 2g, h). In the *Ir*-mAChR-A[4B] mutant, atropine antagonism was completely abolished, whereas higher atropine doses paradoxically potentiated muscarine responses. In contrast, the reciprocal *Ir*-mAChR-B[4A] mutant enabled partial dose-dependent inhibition by atropine.

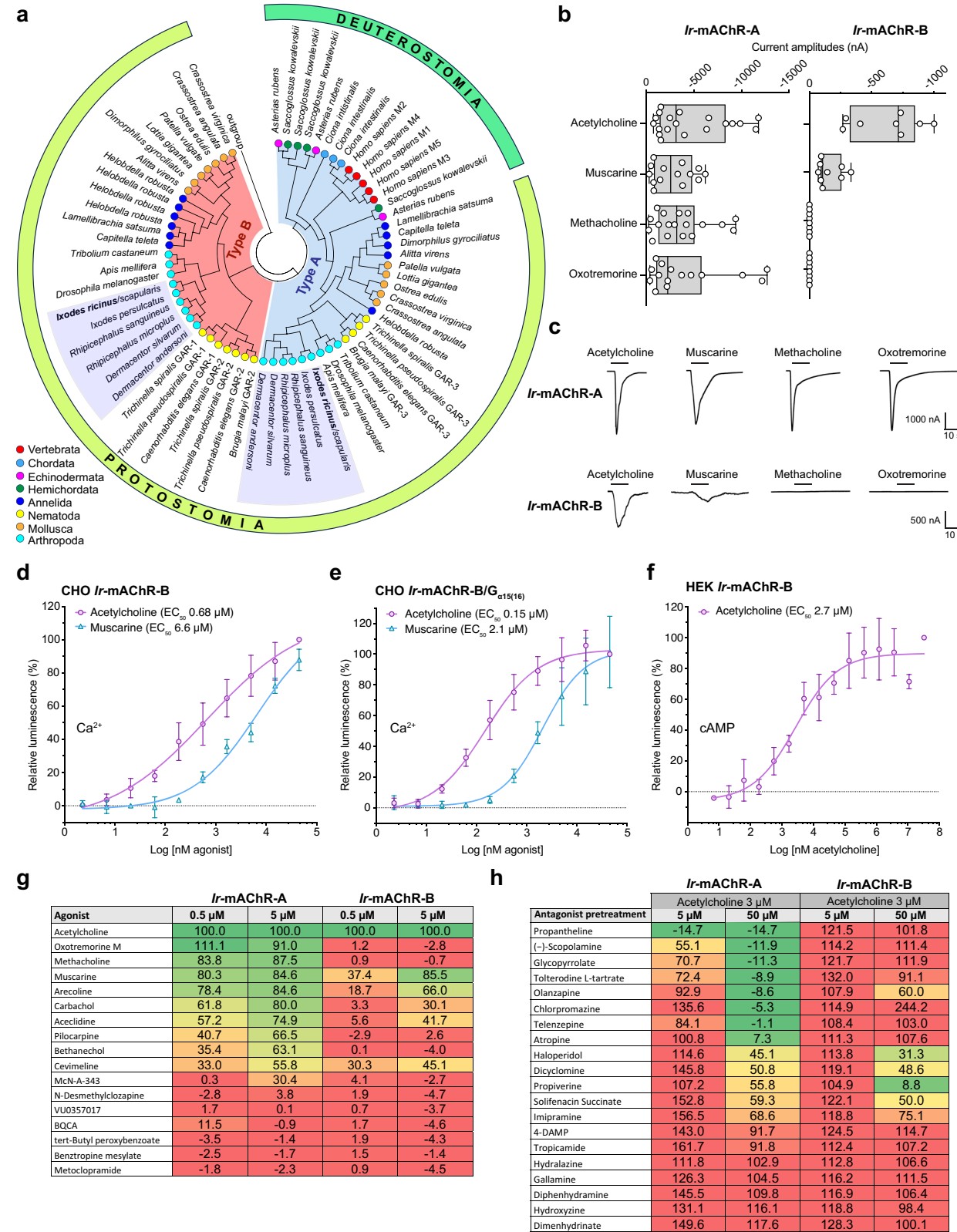

### mAChR-A and -B expressing neurons and their axonal projections in the synganglion and associated nerve trunks

To investigate muscarinic regulation of neural pathways from the synganglion to the SGs[11,19], we used *I. ricinus* type-specific antibodies to localise mAChR-A and -B in the synganglia of females (Fig. 3a–l). Both antibodies labelled distinct neuronal populations, some with axons

projecting through different nerve trunks, while others remained confined within the synganglion.

The cells with the strongest *Ir*-mAChR-A-positive reaction were protocerebral lateral NSC (PcLNS$_{1,2}$), which formed dense axonal arborisation on the dorso-lateral synganglion surface (Fig. 3a, b, f), and opistosomal SG-innervating neurons (OsSG$_{1,2}$) projecting to SG nerves

**Fig. 1 | Functional and pharmacological characterisation of *Ir*-mAChR-A and -B. a** Circular cladogram showing the phylogenetic relationships of type A and type B mAChRs across the animal kingdom; tick taxa are highlighted (see Supplementary Dataset 1 for protein sequences). **b** Mean inward ion currents evoked by 0.5 μM drug stimulation of *Ir*-mAChR-A and -B expressed in *Xenopus* oocytes. For *Ir*-mAChR-A, $n = 19, 14, 18$, and 15 from six independent oocyte batches ($N = 6$); for *Ir*-mAChR-B, $n = 8, 9, 8$, and 8 from seven batches ($N = 7$). Multiple recordings were sometimes obtained from a single oocyte. The box plots show the median (central line), interquartile range (bounds of the box), and the full range from minimum to maximum values (whiskers). **c** Representative current traces recorded upon

administration of various drugs. Dose-response curves of calcium mobilisation mediated via *Ir*-mAChR-B activation in CHO cells without (**d**) or with (**e**) $G_{\alpha 15(16)}$ expression. **f** Dose-response curves of cAMP elevation via *Ir*-mAChR-B activation in HEK cells. Heat map showing agonistic (**g**) and antagonistic (**h**) activity of cholinergic agents on *Ir*-mAChR-A and -B in CHO/$G_{\alpha 15(16)}$ aequorin assay. Data in (**d**–**f**) are from three independent transfections ($N = 3$); (**g**, **h**) show means of two biological replicates ($N = 2$), normalised to the ACh-evoked response (100%) (Supplementary Fig. 1g). Statistics: (**d**–**f**) nonlinear regression. Data are mean ± SD. Source data are available in the Source Data file.

2–4 (Fig. 3a, f). Additionally, *Ir*-mAChR-A immunoreactivity was observed in lateral segmental organs in a three-day-fed female (Fig. 3d, f). In contrast, *Ir*-mAChR-B antibodies identified medial protocerebral NSC (PcMNS$_{1-5}$) with arborisations on the dorso-lateral synganglion (Fig. 3h, i, l), as well as protocerebral SG-innervating neurons (PcSG) projecting to the SG nerve 1 (Fig. 3h, j, l). These data show that *Ir*-mAChR-A and -B are expressed in distinct neuronal populations, including NSC and SG-innervating neurons, pointing to type-specific muscarinic circuits in tick neural regulation.

### Lateral and medial peptidergic NSC express distinct mAChR types

Double immunolabelling with *Ir*-mAChR-A and -B antibodies distinguished lateral and medial NSC, each extending prominent axons that arborise at the synganglion surface (Fig. 3m). In a *Rhipicephalus appendiculatus* study, antibodies against myosuppressin (MS) and FMRFamide neuropeptides recognised medial and lateral NSC[22], corresponding to those identified here, thus confirming their peptidergic identity. Double immunolabelling of *Ir*-mAChR-A and -B with MS[35] and FMRFamide[36] antibodies verified the cholinoceptive nature of these NSC in *I. ricinus* (Fig. 3n and Supplementary Movie 1). Immunogold labelling revealed that axon terminals of PcLNS$_{1,2}$ and PcMNS$_{1-5}$ are positioned within the dorso-lateral perineurium (Fig. 3o), beneath the acellular neurilemma, which faces the haemolymph-filled periganglionic sinus (Fig. 3p). Other axons originating from these NSC extended towards internal synganglion lobes, arborising into fine terminals across medial and lateral regions (Fig. 3q). Because whole-mount confocal imaging lacks membrane resolution, the predominantly intracellular mAChR signal in axons—confirmed by ultrastructural analysis[10]—is interpreted as reflecting receptor trafficking. Most labelled profiles likely represent axonal shafts rather than synaptic terminals, where membrane receptors are concentrated.

In insects, both FMRFamide and MS belong to the FMRFamide-related peptide family, defined by a C-terminal RFamide sequence[37,38]. Hard ticks appear to encode a single ortholog that is structurally intermediate between insect FMRFamide and MS, with a C-terminal I/L-L/MHFamide motif, recently designated FMRFamide_MS-like (FMRFa_MS-L)[39]. We molecularly characterised the *I. ricinus* FMRFa_MS-L transcript and generated an antibody against one of its mature peptides (Supplementary Fig. 5a and 6b–d). Co-localisation of anti-FMRFa_MS-L and anti-*Ir*-mAChR-A in PcLNS$_{1,2}$ cells and axons indicated that insect FMRFa antibody likely recognises FMRFa_MS-L in these neurons (Fig. 4a and Supplementary Fig. 6d). By contrast, the anti-MS antibody, raised against the silkworm MS N-terminal[35], likely cross-reacts with an unknown protein in PcMNS$_{1-5}$ cells (Fig. 3n, o, q and Supplementary Fig. 6c, d, f). Instead, the neuropeptide leucokinin[40] was labelled in PcMNS$_{1-5}$ cells and axons (Fig. 4b, c and Supplementary Fig. 5b and 6e). A schematic summary presenting NSC labelling patterns is shown in Fig. 4d.

We next examined PcLNS$_{1,2}$ axon terminals and adjacent regions on the dorsal synganglion surface using double immunogold labelling with anti-*Ir*-mAChR-A and anti-FMRFa (Fig. 4e, f). FMRFa-labelled nanoparticles were abundant in the perineurium, but sparse near

neuronal somata, suggesting axon-derived peptides are trafficked towards the neurilemma and then released into the haemolymph. Low levels of FMRFa-labelled nanoparticles were also observed in neighbouring axons that possibly originate from PcMNS$_{1-5}$ or other neurons[41], (Fig. 4e), which may reflect peptide transfer, uptake, or a technical artefact.

Finally, immunostaining suggested that cholinoceptive axons from PcLNS$_{1,2}$ and PcMNS$_{1-5}$ cells in the dorsal perineurium (summarised in Fig. 4d) may release neuropeptides upon ACh stimulation. To test this, we injected unfed female ticks with 100 nL water or 100 nL of 10 μM ACh and monitored fluorescence in axons double-labelled with anti-leucokinin (PcMNS$_{1-5}$) and anti-FMRFa (PcLNS$_{1,2}$) (Fig. 4g and Supplementary Fig. 7a–c). Water injection produced a gradual fluorescence increase over 30 min, whereas ACh elicited a significantly attenuated response. We interpret fluorescence increases as likely reflecting enhanced peptide synthesis and axonal transport triggered by haemolymph pressure, whereas the reduction following ACh injection may reflect mAChR-dependent neuropeptide release. These findings indicate that PcLNS$_{1,2}$ and PcMNS$_{1-5}$ axons release neuropeptides into the haemolymph via mAChR-mediated mechanisms, linking cholinergic signalling to systemic endocrine and osmoregulatory processes.

### Peptidergic axons in type II and III SG acini express distinct mAChR types, whereas epithelial mAChRs are localised to type I acini

Tracing immunoreactive axonal projections from the synganglion to SGs revealed two pathways: (i) *Ir*-mAChR-A–positive axons from OsSG$_{1,2}$ neurons exclusively innervating type II acini (Fig. 5a); and (ii) *Ir*-mAChR-B–positive axons from PcSG neurons targeting both type II and III acini (Fig. 5b). The peptidergic identity of these neurons and projections[21,23] was confirmed by double labelling with antibodies against the insect neuropeptides orcokinin[42] and SIFamide[43] (Fig. 5a, b). SIFa-positive axons also reacted with anti-FMRFa and anti-FMRFa_MS-L antibodies, likely reflecting cross-reactivity at the conserved C-terminal Phe-NH$_2$ motif (Supplementary Fig. 5a and 8a–c), though this requires further validation.

Immunostaining showed that type II acini are the only site where *Ir*-mAChR-A and -B expressing axons form close synaptic varicosities (Fig. 5c; summarised in Fig. 6a, b). Within type II acini, *Ir*-mAChR-A terminals extended apically, while *Ir*-mAChR-B terminals remained basally restricted (Supplementary Fig. 8d, e). Examination across feeding stages (Fig. 5d) revealed more persistent *Ir*-mAChR-B immunoreactivity in type II and III acini, compared with less consistent *Ir*-mAChR-A labelling, which was confined to type II acini (Fig. 5e). Consistent with previous findings[10] and confirmed here (Fig. 5d and Supplementary Fig. 8d), anti-βIII-tubulin staining demonstrated that axon terminals from PcSG or OsSG$_{1,2}$ neurons made close contact with a single myoepithelial cell in both type II and III acini. These data indicate that *Ir*-mAChR-A and -B axons converge in type II acini but segregate spatially, with *Ir*-mAChR-B demonstrating broader persistence across acini and feeding stages, pointing to complementary receptor-type–specific roles in regulating myoepithelial cell function and secretion dynamics.

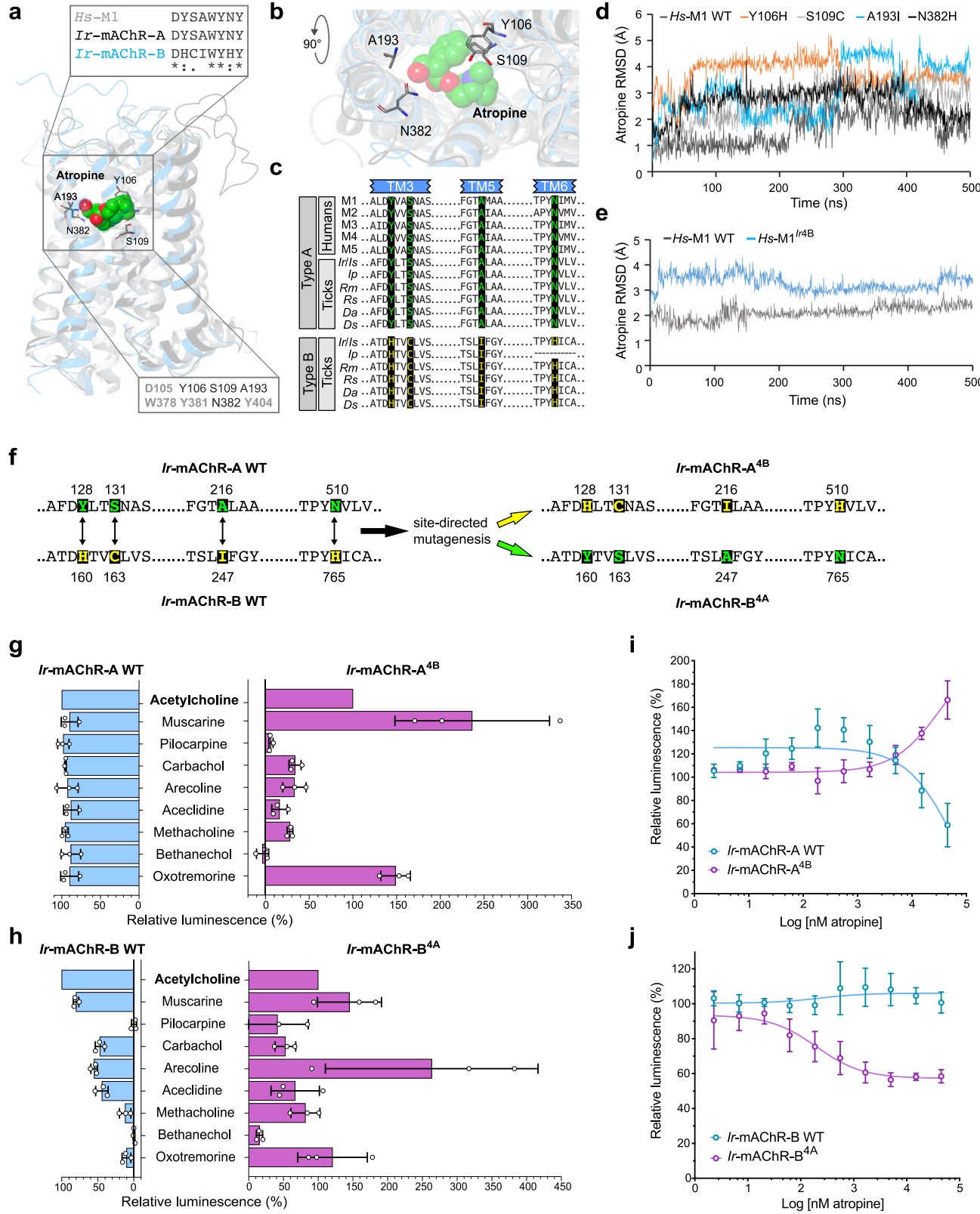

Type I acini, which have not been shown to produce saliva[7], were previously identified as targets of synaptic cholinergic signals[8]. In this study, both anti-*Ir*-mAChR-A and -B antibodies labelled type I acini epithelial cells positioned adjacent to cholinergic axon terminals as detected with an antibody against choline acetyltransferase (ChAT, Fig. 5f and Supplementary Fig. 4d). Thus, *Ir*-mAChR-A and -B are also

expressed in type I acini epithelia, suggesting neuronal cholinergic modulation of non-secretory acini.

## SGs are a major pool of non-neuronal ACh

To investigate whether cholinoceptive axon terminals in saliva-producing type II and III acini (Figs. 5a–e and 6a, b) are exposed to

**Fig. 2 | Conserved *Hs*-M1 binding site residues determine the selective affinity of *Ir*-mAChR-A and -B for atropine. a** Structural alignment of the resolved *Hs*-M1 (grey), and predicted *Ir*-mAChR-A (black) and -B (cyan) structures showing atropine (spheres: red, oxygen; blue, nitrogen; green, carbon) and key binding residues (sticks). Upper inset: sequence alignment of *Hs*-M1 atropine-binding residues with *Ir*-mAChR-A and -B (asterisk, identical; colon, strongly similar; period, weakly similar). Lower inset: *Hs*-M1 atropine-binding residues (grey) indicating substitutions between *Ir*-mAChR-A/*Hs*-M1 and -B (black). **b** A bird's-eye view (90° rotation of **a**) of superimposed *Hs*-M1, *Ir*-mAChR-A and -B showing residues coordinating atropine in the binding site. **c** Sequence alignment of human M1–M5 receptors and tick mAChR-A and -B highlighting residues in TM3, TM5, and TM6 that determine atropine selectivity. Species: *Ir, I. ricinus; Is, I. scapularis; Rm, Rhipicephalus microplus; Rs, Rhipicephalus sanguineus; Da, Dermacentor andersoni; Ds, Dermacentor*

*silvarum*. Full sequences are provided in Supplementary Dataset 1. **d** 500-ns molecular dynamics trajectories showing RMSD of atropine (y-axis) across single *Hs*-M1, reflecting *Ir*-mAChR-B substitutions, and the *Hs*-M1$^{Ir4B}$ (**e**). WT, wild type. **f** Fragments of *Ir*-mAChR-A and -B sequences showing mutation sites (green and yellow, respectively) used to generate *Ir*-mAChR-A$^{4B}$ and -B$^{4A}$ mutant variants. **g, h** Aequorin assays comparing responses of WT and *Ir*-mAChR-A$^{4B}$ and -B$^{4A}$ to 45 µM agonists; ACh responses were normalised to (100%) in each of the three biological replicates. Effect of atropine pre-treatment on 30 µM muscarine- (**i**) and arecoline- (**j**) induced responses of WT and *Ir*-mAChR-A$^{4B}$ or -B$^{4A}$ in CHO/G$_{\alpha15(16)}$ aequorin assay. (**g–j**) Data are from three independent transfections (*N* = 3). Statistics: (**i, j**) nonlinear regression. Data are mean ± SD. Source data are available in the Source Data file.

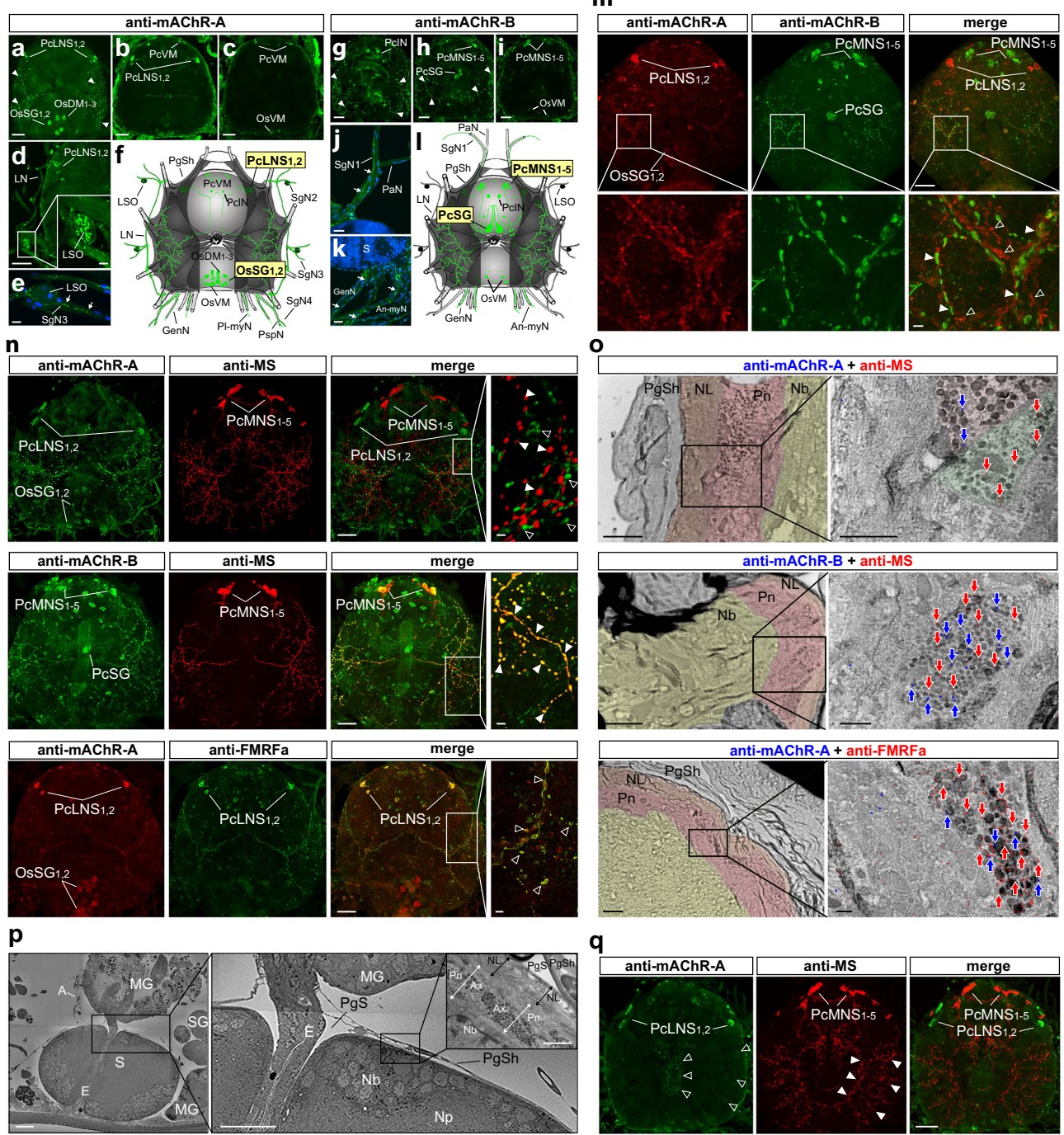

**Fig. 3 | Immunolabelling of *Ir*-mAChR-A and -B in the synganglion.**
**a–l** Immunolocalisation of *Ir*-mAChR-A and -B in neurons and axons of the female synganglion. Z-stack projections from dorsal (**a**, **g**), middle (**b**, **h**), and ventral (**c**, **i**) regions show lateral PcLNS$_{1,2}$ and medial PcMNS$_{1-5}$ NSC in the anterior protocerebrum, with their dorso-lateral surface axons (arrowheads). **d** Immunoreactivity (IR) in lateral nerve (LN) axons and the lateral segmental organ (LSO) of a three-day-fed female. **e** Lack of IR in the LSO of an unfed female, but positive labelling (arrows) in SG nerve 3 (SgN3) originating from OsSG$_{1,2}$ neurons. **f**, **l** Schematic drawings of IR distribution (green); in panel **l**, only representative neurons are annotated. **j** IR detected in the axon (arrows) of SG nerve 1 (SgN1), a branch of the palpal nerve (PaN), from the PcSG neuron. **k** IR-positive axons (arrows) exiting the synganglion (S) via the ano-myosomal (An-myN) and genital (GenN) nerves. DAPI, blue. PgSh, periganglionic sheath; Pl-myN, postlatero-myosomal nerve; PspN, paraspiracular nerve. **m** Double labelling with anti-*Ir*-mAChR-A and -B in PcLNS$_{1,2}$ and PcMNS$_{1-5}$ NSC and their dorso-lateral axons. Open arrowheads, *Ir*-mAChR-A-positive axons (red); filled arrowheads, *Ir*-mAChR-B-positive axons (green). SG-innervating neurons OsSG$_{1,2}$ and PcSG are indicated. **n** Double labelling with anti-*Ir*-mAChR-A and -B plus anti-MS or anti-FMRFa highlights PcMNS$_{1-5}$ and PcLNS$_{1,2}$ and their dorso-lateral axons. Filled arrowheads, PcMNS$_{1-5}$ axons; open arrowheads indicate PcLNS$_{1,2}$ axons; yellow, co-localisation. See schematics in (**f**) and (**l**) and Supplementary Fig. 6d. **o** TEM micrographs showing double immunogold labelling (see insets in **n**) in axons within the dorso-lateral perineurium (Pn, pink). Colour-coded arrows indicate IR. NL neurilemma (brown), Nb neuronal cell bodies (yellow). Supplementary Fig. 6a provides details. **p** SEM of a sagittal section of a 48 h-fed nymph; inset (right) shows TEM of a comparable region. A aorta, MG midgut, E oesophagus, SG salivary gland, Np neuropile, PgSh periganglionic sheath, PgS periganglionic sinus, Ax axon, Nb neuronal cell bodies. **q** Double labelling with anti-*Ir*-mAChR-A and anti-MS. Z-stack projection showing internal axonal networks of PcLNS$_{1,2}$ (green; open arrowheads) and PcMNS$_{1-5}$ (red; filled arrowheads), arborising into distinct lobes (see schematic in Fig. 4d). Validation of anti-*Ir*-mAChR-A and -B antibodies is shown in Supplementary Fig. 4a–c. For neuron nomenclature, see the "Methods" section. Scale bars: 50 μm (**a–d**, **g–i**, **m**, **n**, **p**, **q**); 10 μm (**e**, **j**, **k** insets of **d**, **m** and **n**); 2 μm (**o**); 1 μm (insets of **o** and **p**).

paracrine ACh signalling, we quantified free choline and ACh in SGs during feeding (Fig. 6c). SGs contained variable free choline concentrations in the low nanogram range across feeding stages. ACh displayed greater variability, with detectable increases on days 3 and 6 of feeding. Free choline was detected exclusively in type II and III acini across three independent replicates, while all measurements in type I acini were below the detection limit (Fig. 6c), suggesting an epithelial origin rather than release from cholinergic axons innervating type I acini (Figs. 5f and 6a, b). In the synganglion, both free choline and ACh were present at all timepoints, though at consistently lower levels than in the SGs and with variable ACh detection among replicates (Fig. 6d).

Additional evidence for endogenous ACh synthesis in *I. ricinus* SGs comes from the transcripts encoding ChAT, which mediates ACh biosynthesis, and vesicular ACh transporter (VAChT, which packages ACh into secretory vesicles) (Fig. 6e and Supplementary Fig. 9). Given the absence of anterograde *vacht* and *chat* mRNA transport in cholinergic axons innervating type I acini[8], their detection in SG tissue suggests local synthesis. However, despite extensive attempts, the ChAT protein was unable to be visualised in SGs by immunolabelling of whole-mount or sectioned tissues at any feeding stage, leaving the precise identity of ACh-producing cells in type II and III acini unresolved. *machr-a* and *machr-b* transcripts were detected at low levels in SGs (Ct >36 and 39, respectively, Fig. 6e), likely reflecting expression in epithelial cells of the less abundant type I acini, which receive neuronal ACh input (Figs. 5f, 6a, b). In synganglia, *chat*, *vacht*, and *machr-a* were consistently detected at all timepoints, whereas the *machr-b* transcript was barely detectable throughout feeding (Fig. 6e).

These data support local ACh synthesis in the SGs, where it may act in a paracrine manner to modulate neuropeptide release from cholinoceptive axon terminals. ACh injection into unfed females did not alter anti-SIFa fluorescence in *Ir*-mAChR-B-expressing axon terminals located within type II and III acini (Supplementary Fig. 10), suggesting that exogenous ACh either failed to reach the receptors or inhibited peptide release.

### In vivo effects of muscarinic agonists and antagonists on SG secretion

To assess the physiological roles of axonal *Ir*-mAChR-A and -B in regulating saliva secretion, we injected pharmacological discriminators (Fig. 1g, h) into five-day-fed *I. ricinus* females (Fig. 7a). These included common agonists (ACh, muscarine), *Ir*-mAChR-A-selective agonists (methacholine, pilocarpine), *Ir*-mAChR-A-selective antagonists (scopolamine, atropine), or a broadly acting antagonist (propiverine) (Fig. 7b–j). Using a previously established in vivo dose of ~500 μmol·kg$^{-1}$ of body weight in a pilot experiment[11], ACh and pilocarpine produced no or only weak secretion. Increasing the dose

tenfold to 5 mmol·kg$^{-1}$ of body weight elicited robust secretion with both compounds (Supplementary Fig. 11a, b). These results established a dosing framework for testing muscarinic agonists and antagonists in vivo.

Injection of ACh and muscarine elicited immediate secretion, where ACh-stimulated activity plateaued within 30 min, whereas muscarine resulted in continuously increased secretion throughout the 60-min test period (Fig. 7b, c). Pilocarpine and methacholine also triggered rapid responses that plateaued at 30 min, with reduced secretion for the next 30 min (Fig. 7d, e). Notably, pilocarpine—widely used to induce tick salivation[11]—produced only ~50% of the maximal saliva volume induced by ACh, muscarine, or methacholine (Fig. 7b–f; Supplementary Fig. 11c). The *Ir*-mAChR-A-selective antagonist scopolamine partially inhibited ACh- and muscarine-stimulated secretion (~60%) (Fig. 7b, c, g, h), while virtually abolishing pilocarpine- and methacholine-induced responses (Fig. 7d, e, i, j). This suggests that any residual secretion under scopolamine/ACh or scopolamine/muscarine conditions reflects *Ir*-mAChR-B activity. Atropine fully blocked responses to all agonists, while propiverine nearly abolished ACh-, muscarine-, and pilocarpine-induced secretion, but demonstrated less effective inhibition of methacholine-induced responses (Fig. 7b–e, g–j). Together, these results demonstrate that *Ir*-mAChR-A and -B contribute to salivation with distinct effects on secretion volume and rate.

Notably, injection of atropine or propiverine—but not scopolamine—caused paralysis-like symptoms, including stretching and shaking of the forelegs, and reduced overall leg movement. Whether these effects reflect toxicity at the administered doses, and whether this influenced salivary secretion, remains unclear.

### Impact of mAChR-A and -B activity on saliva protein quantity and composition

Saliva was collected under four pharmacological conditions designed to mimic or inhibit *Ir*-mAChR-mediated responses (Fig. 7k). Protein content was quantified for three of these conditions, and all four were analysed by principal component analysis (PCA) (Fig. 7l, m).

Saliva induced by ACh or muscarine contained comparable protein concentrations (~0.7 and 0.5 μg/μL, respectively). Pilocarpine-induced saliva exhibited higher protein levels (~1 μg/μL), whereas methacholine-induced saliva showed markedly lower protein (~0.26 μg/μL). Strikingly, scopolamine pre-treatment followed by ACh or muscarine injection resulted in significantly elevated protein concentrations (~1.5 and ~2 μg/μL) compared with controls without scopolamine treatment (Fig. 7l).

To assess the impact of receptor-type–specific activity on the saliva proteome, we performed PCA of 218 proteins, including 37 host-derived proteins (Fig. 7m, see also Supplementary Dataset 2). Saliva

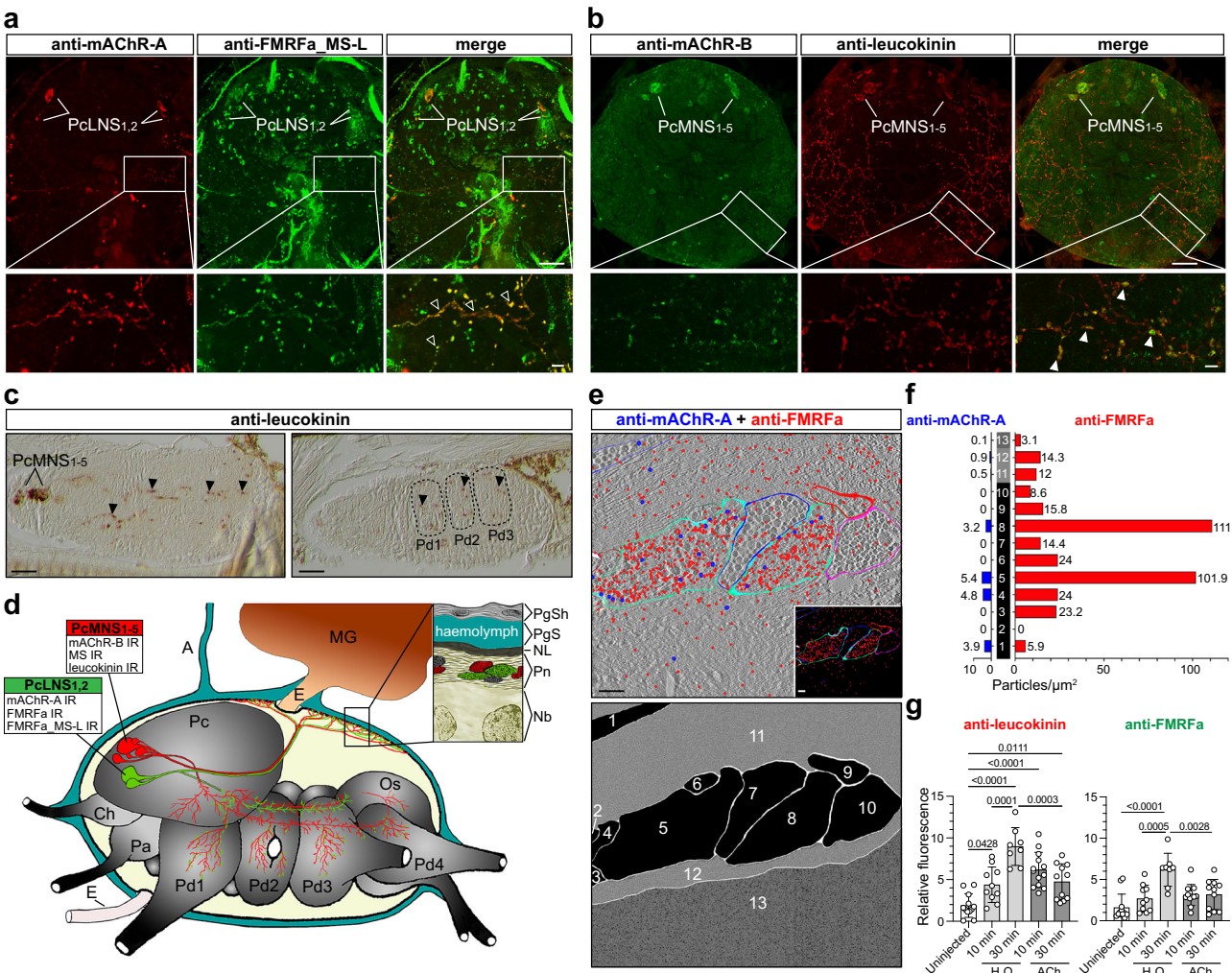

**Fig. 4 | Cholinoceptive axons of peptidergic NSC reach the dorsal perineurium and internal lobes of the synganglion. a, b** Double labelling shows co-localisation (yellow) of anti-*Ir*-mAChR-A with anti-FMRFa_MS-L in PcLNS$_{1,2}$ neurons and of anti-*Ir*-mAChR-B with anti-leucokinin in PcMNS$_{1-5}$ neurons, including their dorso-lateral axons (open and filled arrowheads, respectively). **c** Sagittal section of the synganglion immunolabelled with anti-leucokinin, showing the medial region at the PcMNS$_{1-5}$ level and a deeper plane through the pedal lobes (Pd); arrowheads mark internal axonal projections from PcMNS$_{1-5}$. Dotted lines outline Pd boundaries. **d** Schematic lateral view of the synganglion summarising medial (red) and lateral (green) NSC and their projections based on 3D confocal z-stacks and serial sections. Ch cheliceral lobe, Pa palpal lobe, Pd1–4 pedal lobes, Pc protocerebrum, Os opistosomal lobe, E oesophagus, MG midgut. IR immunoreactivity. Insets: structure of the dorso-lateral surface showing multiple axon terminals (green, red, grey) within the perineurium (Pn). PgSh, periganglionic sheath; PgS, periganglionic sinus; NL, neurilemma; Nb, neuronal cell bodies. **e** TEM micrographs showing double immunogold labelling with anti-*Ir*-mAChR-A (15 nm gold, blue) and anti-FMRFa (10 nm gold, red) in the dorso-lateral perineurium; inset shows contrast-

enhanced detail. Schematic illustrates labelled axons (1–10), perineurium (11, 12), and the region of neuronal cell bodies (13). **f** Quantification of immunogold-labelled nanoparticles from (**e**). Bars represent *Ir*-mAChR-A (blue) and FMRFa (red) labelling. The raw counting data are provided in the Source Data file. **g** Quantification of fluorescence intensities in dorso-lateral axonal arborisation from PcMNS$_{1-5}$ and PcLNS$_{1,2}$ cells labelled with anti-leucokinin and anti-FMRFa, respectively, in uninjected, water-, or ACh-injected unfed females. Methodological details are in Supplementary Fig. 7. Anti-leucokinin and anti-FMRFa intensities: $N = 6, 5, 4, 6, 5$ synganglia; for anti-leucokinin, $n = 12, 10, 8, 13, 11$ lateral sides of synganglia or subdivisions; for anti-FMRFa, $n = 12, 10, 12, 11$ lateral sides or subdivisions (subdivisions used to avoid nonspecific signal). Only statistically significant comparisons ($p < 0.05$) are indicated. Validation of anti-*Ir*-FMRFa_MS-L and anti-leucokinin antibodies is shown in Supplementary Fig. 5. Statistics: **g** One-way ANOVA (two-sided) followed by Tukey's multiple-comparison post hoc test. Data are mean ± SD. Source data are available in the Source Data file. Scale bars: 50 μm (**a**–**c**); 10 μm (insets in **a, b**); 1 μm (**e**).

induced by each drug formed distinct clusters. ACh- and muscarine-induced samples did not clearly separate from pilocarpine- and methacholine-induced samples, which overlapped near the origin. In contrast, ACh- and muscarine-induced saliva following scopolamine or propiverine pre-treatment showed clear separation from all other groups. Muscarine alone separated from ACh along PC2, and for both agonists, this divergence became more pronounced after scopolamine pre-treatment. Protein identities varied across conditions (Fig. 7k, n and Supplementary Fig. 12 and Supplementary Dataset 2). Unique sets were detected in saliva induced by ACh (17 proteins), methacholine (15), and pilocarpine (9). In scopolamine-pre-treated ticks, subsequent

ACh and muscarine injection yielded 15 and 9 unique proteins, respectively. These findings suggest that muscarinic inputs do not generate stimulus-specific secretomes, but rather shift the relative abundance of salivary proteins, reflecting flexible regulation across acinar cell types. Importantly, abundant host proteins, including haemoglobin, albumin, and serum transferrin, and immune effectors such as neutrophilic granule protein and protein S100, were consistently detected across conditions (Supplementary Dataset 2). Their recurrent presence indicates that vertebrate haemostatic and immune factors are not contaminants but integral components of the saliva proteome, underscoring their dynamic role at the host-parasite interface.

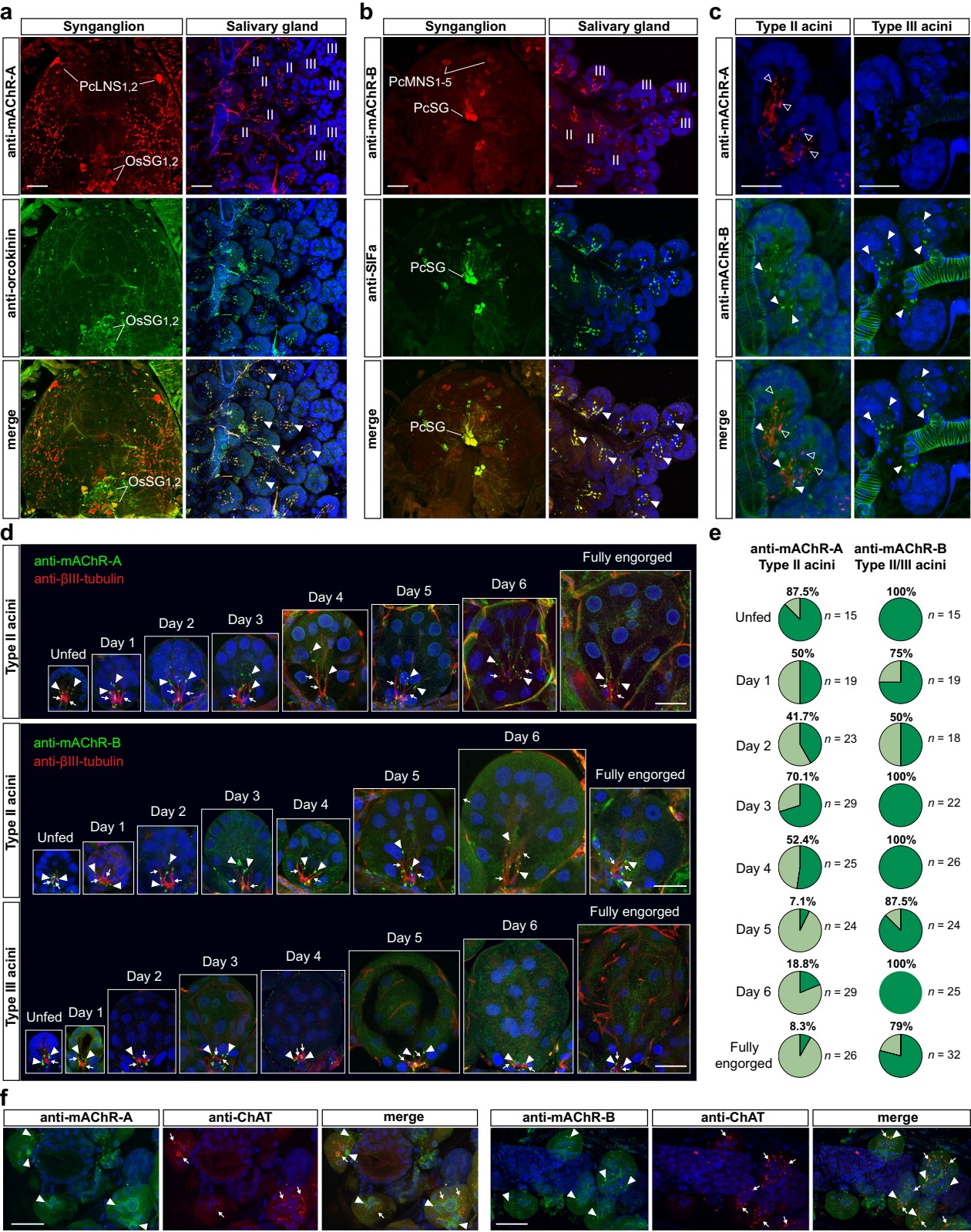

## Discussion

Functional and pharmacological studies on invertebrate mAChRs progress rather slowly compared to research on their mammalian counterparts. As yet, only a few studies have investigated these receptors in nematodes and arthropods, but their specific physiological functions in invertebrates remain largely unknown. Central to our findings is the anatomico-physiological model of two distinct axonal mAChR types (A and B) mediating salivary secretion in *I. ricinus* females. In addition to their sequence, the selective affinity of *Ir*-mAChR-A and -B to a number of classical muscarinic drugs supports a recent classification that all deuterostome mAChRs—including mammalian M1–M5—belong to the type A group, while in protostomes, additional pharmacologically distinct type B mAChRs are present[24,29].

**Fig. 5 | Type II and III SG acini are innervated by cholinoceptive peptidergic axons, while agranular type I acini are targets for synaptic ACh.** Double immunolabelling of anti-*Ir*-mAChR-A with anti-orcokinin (**a**) and anti-*Ir*-mAChR-B with anti-SIFa (**b**) in synganglial neurons and their axonal projections to the SG of an unfed female. Co-localisation of anti-*Ir*-mAChR-A with anti-orcokinin marks OsSG$_{1,2}$ neurons (yellow) and their axon terminals (yellow, arrowheads) targeting type II acini, while anti-*Ir*-mAChR-B co-localises with anti-SIFa in PcSG cells (yellow) and their axon terminals (yellow, arrowheads) innervating type II and III acini. **c** Double immunolabelling of SG acini from an unfed female with anti-*Ir*-mAChR-A (red, open arrowheads) and anti-*Ir*-mAChR-B (green, filled arrowheads) showing that type II acini receive both inputs, whereas type III acini are labelled exclusively with anti-*Ir*-mAChR-B. **d** Association of anti-*Ir*-mAChR-A- and anti-*Ir*-mAChR-B-positive axon terminals (green, arrowheads) with a single myoepithelial cell visualised with anti−βIII-tubulin (red, arrows) across feeding stages, showing acinar enlargement from unfed to fully engorged females. **e** Number of female SGs positive for anti-*Ir*-mAChR-A and anti-*Ir*-mAChR-B immunoreactivity in axon terminals in type II and III acini. Tick numbers (*n*) are shown in the figure. **f** Double labelling of type I acini with anti-*Ir*-mAChR-A or anti-*Ir*-mAChR-B with anti-ChAT shows scattered *Ir*-mAChR-type-positive immunoreactivity (green, arrowheads) in acinar cells adjacent to ChAT-positive axon terminals (red, arrows). DAPI, blue. A schematic of SG-innervating neurons and their axons is shown in Fig. 6a, b. Validation of anti-*Ir*-mAChR-A and anti-*Ir*-mAChR-B antibodies is provided in Supplementary Fig. 4. Scale bars, 50 μm.

*Ir*-mAChR-A and *Ir*-mAChR-B induced ionic currents of different amplitudes in *Xenopus* oocytes, but the identity of the ion channels and the mechanisms linking receptor activation to channel opening remain unknown. These differences likely reflect intrinsic receptor properties, such as ligand-binding affinity and G protein coupling efficiency, assessed within a simplified oocyte membrane environment. In contrast, similar *Ir*-mAChR-B (this study) and *Ir*-mAChR-A (in Mateos-Hernández et al.[8]) EC$_{50}$ values measured in CHO cells are influenced by endogenous signalling pathways, membrane composition, and/or regulatory proteins, which buffer intrinsic receptor differences. Together, these results highlight the need to further investigate how mAChRs and other GPCRs couple to those ion channels playing key roles in tick SG physiology[44,45].

G$_{q/11}$ pathway activation by *Ir*-mAChR-B in CHO cells mirrors signalling previously described for GAR-1 and GAR-2 receptors in the *T. spiralis* nematode[29]. cAMP production was also stimulated in HEK cells, indicating engagement of a second distinct cascade; however, parallel assays revealed cAMP inhibition at lower ligand concentrations, consistent with concentration-dependent bidirectional regulation of adenylate cyclase. These findings challenge the prevailing view that all mAChR-B receptors couple exclusively to G$_i$ proteins and act solely as inhibitory receptors[24]. Indeed, previous studies have reported that G$_i$ or G$_{q/11}$-coupled mAChRs may, under certain conditions, activate adenylyl cyclase and elevate cAMP[46], and that varying ligand concentrations can bias between G$_s$ and G$_i$ in other GPCR systems[47]. These observations underscore the need for detailed analyses of mAChR-B signalling under different physiological contexts, which is essential to defining the full functional repertoire of these receptors.

Immunolabelling revealed an extensive network of cholinoceptive peptidergic axon terminals at the latero-dorsal synganglion perineurium (summarised in Fig. 4d). Although neurosecretory axons at this region were described in the 1980s[48], our study is the first to trace their origin to lateral and medial NSC. By morphology and location, these cells likely correspond to insect protocerebral NSC, which are also cholinoceptive[49]. In ticks, the barrierless perineurium/neurilemma complex may allow direct exposure of these axons to hormonal ACh. In turn, their activation could release neuropeptide hormones into the haemolymph, and/or initiate signalling cascades through internal peptidergic axons across synganglion lobes. Additional support for a hormonal ACh pathway comes from *Ir*-mAChR-A detection on haemolymph-surrounded lateral segmental organs, although the physiological relevance of this system remains unclear. We speculate that protocerebral cholinoceptive NSC in tick are part of the previously suggested sensory pathway[20], responsive to haemolymph volume and acting directly or indirectly on SG-innervating neurons. Furthermore, nicotinic acetylcholine receptors (nAChRs) are expressed in synganglion[50,51], but their precise localisation and role in tick neurophysiology remain unresolved.

Our data support the existence of the long-postulated cholinoceptive secretomotor nerves that extend to the SG[4,20]. Unlike in vertebrates, neuron-neuron communication rarely occurs at the level of cell somata in insects[52], instead taking place along axons or their terminals—a pattern that likely also applies to ticks. However, our prior patch-clamp study detected ACh-, muscarine-, and nicotine-sensitive sites at the somata of neurons resembling PcSG cells[23,53]. Therefore, we suggest that cholinergic signal clusters may diffuse to the somata of these SG-innervating neurons[20] or act directly on their axo-axonic contacts within or near the synganglion to mediate their activation. The presence of mAChR-A and -B at peptidergic axon terminals in SG acini further support their role as presynaptic heteroreceptors, capable of modulating neuropeptide release as shown in mammals[54]. In this case, ACh signals would likely originate from adjacent SG cells. Although we confirmed substantial ACh levels in *I. ricinus* SGs, we were unable to localise cholinergic cells, possibly due to rapid ChAT protein turnover[55] consistent with scattered ACh signal and variable *chat* and *vacht* transcript levels across individuals. To date, no evidence supports expression of nAChRs in tick SG[50]. Thus, mAChRs remain the primary candidates for mediating cholinergic signalling in this tissue.

The reported inability of ACh to stimulate fluid secretion in hard ticks[11] likely reflects the long-standing reliance on the exogenous cholinomimetic pilocarpine. In contrast, we found that injection of ACh or muscarine triggered robust secretion in *I. ricinus*, with approximately twice the efficacy of pilocarpine. Our pharmacological data suggest that pilocarpine's reduced efficacy arises from its inability to activate *Ir*-mAChR-B and from its partial agonism at *Ir*-mAChR-A (this study and Mateos-Hernández et al.[8]). By comparison, methacholine—a potent *Ir*-mAChR-A–specific agonist—induced secretion volumes similar to those evoked by ACh and muscarine.

In 1982, Kaufman and Harris[19] demonstrated that severing lateral SG-innervating nerves 2 and 3 in *Amblyomma hebraeum* reduced carbachol-induced salivation, whereas severing the palpal nerve caused only a minor reduction in saliva volume. At the time, however, the receptor specificity of carbachol action was unknown. Here, we show that carbachol activates both *Ir*-mAChR types, acting as a full agonist at *Ir*-mAChR-A and a partial agonist at *Ir*-mAChR-B. Consistent with this mechanistic insight in *I. ricinus*, selective stimulation of mAChR-A immunolocalised to axon terminals innervating type II acini via SG nerves 2–4, triggered robust secretion, with saliva volumes comparable to those observed when both receptor types were activated. In contrast, blocking mAChR-A and stimulating salivation with a common agonist markedly reduced secretion, revealing a strong dependence on mAChR-A signalling. Under these conditions, secretion would therefore be mediated solely by mAChR-B, which we detected in an axonal branch of the palpal nerve projecting to type II and III acini. Together, these results suggest a predominant role for type A mAChR in driving fluid secretion.

Our model of axonal mAChR type-specific regulation of tick SG secretion aligns closely with Kaufman and Harris' findings[19], and highlights the need to dissect how mAChR-A signalling via OsSG$_{1,2}$ neurons that innervate type II acini drives robust fluid secretion. Although mAChR-A stimulation elicits high saliva volumes, the fluid likely originates from the large-lumen type III acini, because type II acini—with their small lumens and high abundance of granular cells—are generally specialised for protein and molecule secretion[6,7]. The

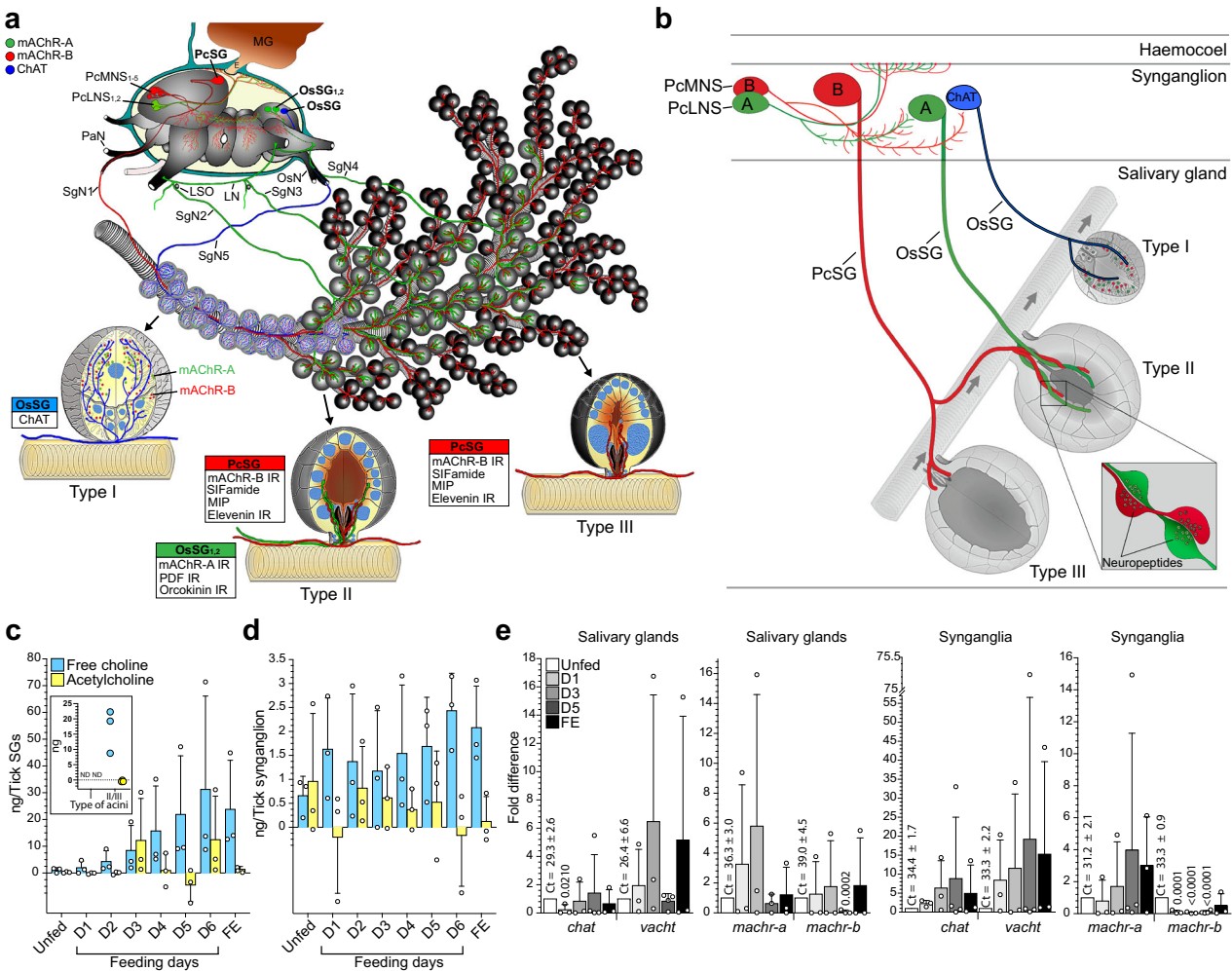

**Fig. 6 | Muscarinic and cholinergic innervation, choline/ACh dynamics, and expression of cholinergic components in SGs and synganglia. a** Schematic overview depicting muscarinic cholinoceptive NSC, cholinoceptive neurons innervating type II and III acini, and cholinergic neurons innervating type I acini (unfed female). SgN1-5, salivary nerves; PaN, palpal nerve; LN, lateral nerve; LSO, lateral segmental organ; OsN, opistosomal nerve; MD, midgut. Previously reported immunoreactivity (IR) for pigment dispersing factor (PDF), myoinhibitory peptide (MIP), elevenin and orcokinin neuropeptides in axons entering acini is indicated[21,22,88]. **b** Simplified diagram highlighting major muscarinic cholinoceptive neurons−including NSC and SG-innervating neurons−and cholinergic neurons projecting to the SG (5 day-fed female). PcMNS and PcLNS neurons extend axons toward the haemocoel-facing surface, while additional projections arborise internally and may contact or modulate SG-innervating axons. Inset: magnified view of synaptic varicosities within a type II acinus where neuropeptides accumulate. Arrows indicate saliva flow direction in the main duct. mAChR-A, -B and ChAT-expressing neurons are colour-coded (A - green, B - red and ChAT - blue). (**a**) and (**b**) include 3D schematics of individual acini adapted and modified from Šimo et al.[16] and Mateos-Hernández et al.[8], reused under the Creative Commons

Attribution 4.0 International License (CC-BY 4.0; https://creativecommons.org/licenses/by/4.0/). Quantification of free choline and ACh in SGs (**c**) and synganglia (**d**) from unfed, 1–6-day-fed, and fully engorged (FE) females. Negative ACh values indicate undetectable levels, after subtracting free choline from total choline (free choline + ACh). The inset in (**c**) presents a dot plot of free choline and ACh levels in type I and type II/III acini isolated from a five-day-fed female. Because tissue is inevitably lost during acini separation, these measurements are not directly proportional to overall SG content. **e** Relative mRNA levels of *chat, vacht, machr-a* and *-b* in SG and synganglion from unfed, 1-, 3-, and 5-day-fed, and FE females. Expression in unfed ticks was normalised to 1; *p*-values above bars indicate significant differences versus the unfed stage. Due to high variability among biological replicates, most comparisons in (**c**−**e**) were not statistically significant (*p* ≥ 0.05). **c**−**e** Each data point represents a pooled organ sample from multiple females, obtained from three independent biological replicates (*N* = 3) for all unfed and fed stages, except D5 in some subpanels of (**e**), which includes four replicates (*N* = 4). Statistics: **c**−**e** two-tailed Student's unpaired *t*-test. Data are mean ± SD. Source data are provided in the Source Data file.

intermittent presence of mAChR-A protein in type II axon terminals further suggests that it may act as a rate-limiting factor in cholinergic control of fluid output. In contrast, mAChR-B−positive axons innervate the basal regions of both type II and III acini, most evident during late feeding, suggesting readiness to regulate secretion and drive acinar emptying through myoepithelial cell contractions that also control the basal valve[10]. mAChR-A-positive axons also enter type II acini via the basal region, suggesting a role in valve regulation. However, unlike

mAChR-B, mAChR-A axons extend further toward apical regions, suggesting additional functions such as specific cell targeting or modulating apical secretory mechanisms, consistent with earlier observations of neuropeptide-positive axonal projections[10].

In this model, mAChR-A-positive axons may initiate secretion by stimulating type II acini and simultaneously transmit neuropeptide signals via axo-axonic synapses within the type II acini to adjacent mAChR-B-positive axons. Such interaction could coordinate activation

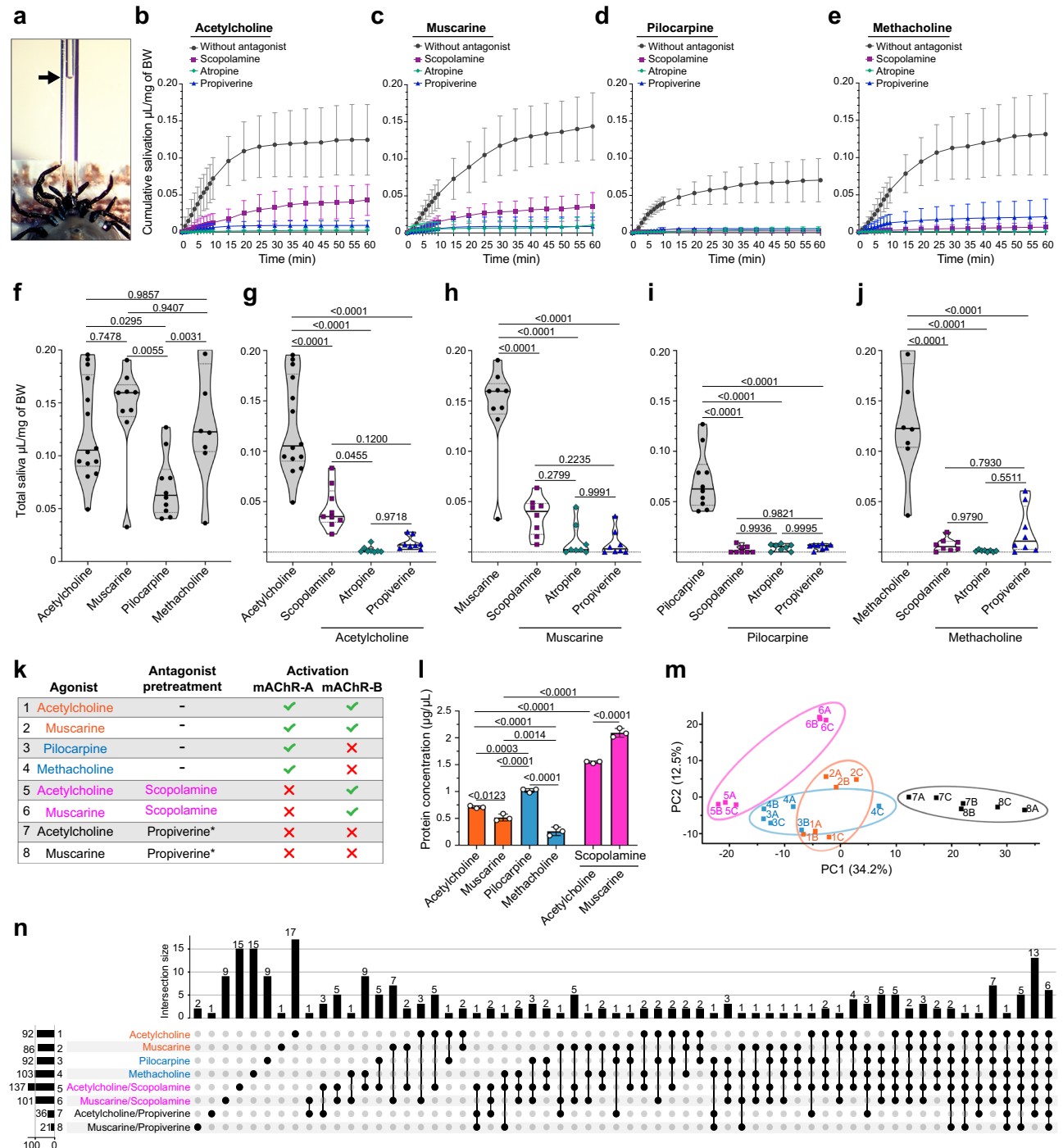

**Fig. 7 | mAChR-A and -B mediate SG secretion rate and protein composition of saliva. a** Experimental setup showing the anterior part of a partially fed *I. ricinus* female salivating into a glass capillary attached to the hypostome. The arrow indicates the saliva level. **b**–**e** Cumulative salivary secretion rates over 60 min following injection of muscarinic agonists or 5-min pre-treatment with a muscarinic antagonist followed by agonist injection. *n* values indicate individual female ticks analysed: **b** *n* = 14, 9, 8, 8; **c** *n* = 9, 8, 8, 8; **d** *n* = 10, 8, 8, 8; and **e** *n* = 8, 9, 8, 8. **f**–**j** Truncated violin plots (solid line: median; dotted line: quartiles; bounds: data limits) showing total saliva volume secreted over 60 min by females treated with the indicated compounds. Each dot, square, diamond, or triangle represents one tick (see also Supplementary Fig. 11c). **k** Eight muscarinic drug combinations used to induce saliva secretion were analysed for protein content (**l**) and composition

(**m**, **n**). Conditions 1–6 correspond to treatments shown in (**b**–**e**); conditions 7 and 8 (asterisk) refer to additional treatments shown in Supplementary Fig. 11d, e. **l** Comparison of protein content in saliva samples 1–6 collected from induced salivation experiments (**k**). **m** Principal component (PC) analysis of saliva from 5-day-fed females, showing clustering by treatment condition; colours and numbers correspond to those in (**k**). **n** UpSet plot showing protein intersections among the eight experimental conditions (see also Supplementary Fig. 12 and Supplementary Dataset 2). **l**–**m** Saliva analyses were performed in three technical replicates (*n* = 3). Statistics: **b**–**e**, **l** data are mean ± SD; **f**–**j** and **l** one-way ANOVA (two-sided) with Tukey's post hoc test for multiple comparisons; *p*-values shown above the bars. Source data are provided in the Source Data file.

of type II and type III acini, coupling protein release with fluid flushing to move contents into the main salivary duct (see Fig. 6b). Although mAChR-A stimulation alone produces saliva volumes comparable to combined mAChR-A and -B activation, the protein concentrations were significantly lower, suggesting that mAChR-A alone cannot generate the full saliva cocktail. In contrast, mAChR-B activity yielded saliva enriched in proteins, but with reduced fluid volume, likely reflecting a primary role in ejecting pre-accumulated material rather than shaping the protein composition of saliva. It remains unresolved whether proteins predominantly derive from type II acini and fluids from type III acini under these experimental conditions. It also remains unclear whether mAChR-A and -B act independently or are co-activated during feeding. Nonetheless, our findings suggest that constitutive mAChR-B activity enables full mAChR-A signalling and coordinated secretion of protein and fluid components, positioning mAChR-B as a permissive regulator of the complete secretory response.

Although saliva volume and protein concentration can be broadly associated with the activity of type II or III acini, we were unable to directly assign specific proteins to individual acinus types. Moreover, the distinct saliva proteomic profiles elicited by drugs that either activate or inhibit the same type of mAChR suggest that multiple, context-dependent modes of action are involved. Resolving these discrepancies will require acinus-specific proteomics in combination with other omics approaches to define the contribution of different structural subunits. In parallel, the roles of additional modulators such as neuropeptides, dopamine, or others[9], and their potential interactions must be investigated to fully understand the dynamic regulation of tick SG secretion.

The localisation of both *Ir*-mAChR-A and -B to type acini I was expected, given the presence of cholinergic innervation previously reported for these structures[8]. This pathway has been implicated in tick rehydration through water uptake[8], as well as in the resorption of ions via type I acini from hyperosmotic primary saliva produced by upstream type II and III acini[56]. Whether this ion resorption is also mediated by muscarinic signalling—via mAChR-A or -B—and whether it operates in synchrony with the secretory activities of type II and III acini, remain open questions.

Given the highly versatile functions of tick SGs[2], their performance depends on the coordinated activity of three distinct acinus types, each under specialised neural control. Among these, two sets of peptidergic neurons[9]—expressing different types of mAChRs—govern saliva production through the innervation of hundreds of acini. Our study reveals how cholinergic signalling through these neurons orchestrates acinar output. Further, it offers new insights into the neural regulation of tick salivation with potential implications for future pharmacological and vector control strategies. In this context, the invertebrate-specific *Ir*-mAChR-B emerges as a promising candidate for selective intervention. More broadly, our study highlights the evolutionary distinctiveness of this cholinergic regulatory system among blood-feeding arthropods, underscoring its central role in enabling ticks to adapt to their specialised parasitic lifestyle.

## Methods

### Ticks and experimental animals
All tick feeding procedures followed established protocols[57,58]. Animals used in France were approved by the ComEth Anses/ENVA/UPEC Ethics Committee for Animal Experimentation (permit No. APAFIS #35511–2022022111197802 v2). In the Czech Republic, procedures complied with the Animal Protection Law of the Czech Republic No. 246/1992 Sb. (Regulation 419/2012) and ethics approval No. 13/2021-P. Animals were maintained ad libitum in an accredited facility (Ministry of Agriculture No. 4253/2019-MZE-17214 and 1643/2019-MZE-17214). *I. ricinus* adults were obtained from the tick rearing facility of UMR BIPAR (Maisons-Alfort, France) and the Institute of Parasitology (České

Budějovice, Czech Republic). Ticks were maintained in 100 mL plastic vials with sterile wood shavings at 22 °C, 90% relative humidity, and 11 h/13 h light-dark cycle (PHCbi, Japan). For adult female feeding stages, up to 80 tick pairs were fed on 11–12-week-old New Zealand White female rabbits (Charles River). To obtain 5-day-fed females for in vivo salivation experiments, two adult pairs were placed on a 5–7-week-old Oncins France 1 female mouse (Charles River).

### Phylogenetic analyses
BLAST searches using previously identified mAChRs from *I. scapularis* (XP_002403135.1, XP_002416160.3), *D. melanogaster* (AFJ23965.1, AGE13748.1), and GAR-1-3 from *Trichinella spiralis* (KRY43096.1, OR220883, KRY27749.1)[8,28,29], were performed against various animal genomes, transcriptomes, and protein databases in NCBI (http://www.ncbi.nlm.nih.gov/). For phylogenetic analysis, the transmembrane regions of putative mAChRs were aligned using ClustalW in MEGA11, and a neighbour-joining tree was constructed with 500 bootstrap replications[59]. A circular phylogenetic tree was coloured and annotated using Adobe Photoshop 26.3.0 and Adobe Illustrator 29.3 (Adobe Inc.).

### Oocyte injection and voltage-clamp recordings
Defolliculated *Xenopus laevis* oocytes were obtained from TEFOR Paris-Saclay University, France and incubated in standard oocyte saline (SOS: 100 mM NaCl, 2 mM KCl, 1 mM MgCl2, 1.8 mM CaCl$_2$, and 5 mM HEPES pH 7.5) supplemented with penicillin (100 U·mL$^{-1}$), streptomycin (100 mg·mL$^{-1}$), gentamycin (50 mg·mL$^{-1}$) and sodium pyruvate (2.5 mM)[60,61]. Oocytes were enzymatically treated with Ca$^{2+}$-free SOS solution containing collagenase IA (2 mg·mL$^{-1}$) and trypsin inhibitor (0.8 mg·mL$^{-1}$) (Sigma, France) and manually defolliculated.

Oocytes were microinjected with 50.6 nL of cRNA (10–20 ng/oocyte) coding for *Ir*-mAChR-A or -B (GenBank: GIXL01002966.1 and PP921913) using a Nanoliter 2020 injector (World Precision Instrument). Recombinant pcDNA3.1+ plasmids containing *Ir*-mAChR-A or -B open reading frames[8] were linearised with PmeI (New England Biolabs, USA), and cRNAs were transcribed in vitro using the T7 mMESSAGE mMACHINE kit (Ambion, USA). G protein-coupled inwardly rectifying potassium channel (GIRK)[61] cRNA was not co-injected. Oocytes were maintained at 18 °C and recorded 3–9 days post-injection.

Electrophysiological recordings were performed with the Roboocyte 2 system (Multi Channel Systems, Germany). Membrane currents were recorded using two 3 M KCl-filled microelectrodes (0.2 and 5 MΩ)[61]. Oocytes were voltage-clamped at −60 mV and perfused with SOS at 3 mL.min$^{-1}$[62]. Drugs were applied for 10 s followed by 30 min washout[63,64]. Oocytes lacking *Ir*-mAChR-A or -B expression were excluded. Negative controls (0.5 μM ACh, muscarine, oxotremorine, or methacholine on non-injected oocytes) produced no inward currents (Supplementary Fig. 13a). For drugs used, see Supplementary Table 1.

### Heterologous expression of *Ir*-mAChR-B in mammalian cell lines and functional assays
The full-length *Ir*-mAChR-B open reading frame (PP921913) was codon-optimised for expression in CHO cells and chemically synthesised (Biomatik, Canada). The sequence was inserted into a pcDNA3.1(+) vector (Invitrogen), with the GCCGCCACC Kozak consensus sequence preceding the start codon. The receptor was expressed in CHO-K1 cells (85051005, Sigma, France) together with the aequorin reporter[65], with or without wild-type human G$_{α15(16)}$ (cDNA Resource Center, Bloomsburg University of Pennsylvania), to monitor intracellular calcium mobilisation-induced bioluminescence upon receptor activation[66]. Bioluminescence was measured using Fluostar Omega (BMG Labtech) or Glomax Discover (Promega) microplate readers. Half maximum activation (EC$_{50}$) values were calculated using GraphPad Prism 9 (GraphPad Software, La Jolla, CA, USA).

To assess cAMP modulation upon *Ir*-mAChR-B stimulation, we used the non-lytic GloSensor™ cAMP assay system (Promega)[16]. HEK293 (85120602, Sigma, France) were co-transfected with CHO-codon-optimised *Ir*-mAChR-B in pcDNA3.1(+) plasmid and the GloSensor-expressing plasmid (Promega). For the activation assay, cells were directly exposed to varying ACh concentrations and luminescence was recorded. For the inhibition assay, cells were pre-treated for 5 min at RT with different ACh doses followed by 10 μM forskolin. Luminescence was recorded every ~1 min for 25 min, and values were normalised to the maximal response (at 15 min for inhibition and 5 min for activation) after removing background.

Mock transfections using only aequorin/$G_{\alpha15(16)}$ (calcium assay)[8] and GloSensor (cAMP assay; Supplementary Fig. 13b) served as negative controls. Detailed transfection and culture procedures are described in Šimo et al.[16].

### Pharmacology of *Ir*-mAChR-A and -B

Agonistic and antagonistic effects of cholinergic agents on *Ir*-mAChR-A and -B were tested in CHO-K1 cells co-expressing each receptor with the aequorin reporter and $G_{\alpha15(16)}$. *Ir*-mAChR-A was expressed using the *Ir*-mAChR-A/pcDNA3/Zeo(+) construct from Mateos-Hernández et al.[8], and *Ir*-mAChR-B using the *Ir*-mAChR-B/pcDNA3.1(+) construct described above.

For agonist assays, cells were exposed to 0.5 or 5 μM ligand, and luminescence was recorded immediately. For antagonist assay, cells were pre-treated with 5 or 50 μM antagonist for 5 min at RT, followed by 3 μM ACh with luminescence immediately recorded. For drugs used, see Supplementary Table 1.

### Protein structure prediction and preparation

Transmembrane regions and loop orientations were predicted using DeepTMHMM[67]. *I. ricinus* mAChR protein models were generated in Phyre2[68] using intensive mode, which aligns the query to remote homologues. Disordered regions, including N-termini (unless stated) and IL3, were truncated. Predicted structures and the *Hs*-M1-atropine complex[33] were prepared and optimised in Maestro[69]. Hydrogen atoms were replaced, steric clashes resolved by local minimisation, and hydrogen-bond networks optimised using the Protein Preparation Wizard[70]. In silico mutations of the *Hs*-M1-atropine complex[33] were performed in Maestro[69]. The *Hs*-M1 was solvated in a 12 Å orthorhombic TIP3P water box[71,72], neutralised and supplemented with 0.15 M NaCl for molecular dynamics preparation.

### Classical molecular dynamics

Protein–ligand complexes and ions were parameterised using the CHARMM36 force field[73], and atropine parameters were generated with SwissParam[74]. Simulations were performed on a GPU-accelerated workstation using Desmond[75], which included multiple equilibration steps followed by a production run under isotropic NPT conditions, employing a Nose-Hoover thermostat[76] and a Martyna-Tobias-Klein barostat[77]. Simulations were conducted at 300 K with a RESPA integrator[78] and a 2-fs inner time step. Images were prepared in Maestro[69], and trajectory analyses were performed using Visual Molecular Dynamics[79].

### Generation of *Ir*-mAChR-A and -B mutants and functional assays

Based on molecular dynamic simulations (see "Results" section: Active-site mutations in *Ir*-mAChR-A and -B destabilise receptor conformations and alter their interactions with atropine), mutant forms of *Ir*-mAChR-A and -B were manually designed (Supplementary Fig. 2). Sequences were codon-optimised for expression in CHO cells, chemically synthesised (Biomatik, Canada), and cloned into pcDNA3.1(+) (Invitrogen) with a Kozak sequence preceding the start codon. Functional assays were performed in CHO cells using the aequorin reporter and $G_{\alpha15(16)}$ as described in the "Methods" section (Pharmacology of *Ir*-

mAChR-A and -B, Mateos-Hernández et al.[8] and Ning et al.[29]). Dose-response curves were generated with GraphPad Prism 9 (GraphPad).

### Whole-mount immunofluorescence and antibodies

Whole-mount staining followed previously established protocols for tick tissues[22,80]. Primary antibodies and dilutions were listed in Supplementary Table 2. Antibodies against *Ir*-mAChR-A or -B were generated in rabbits using carboxy-terminal peptide antigens conjugated to keyhole limpet haemocyanin, followed by affinity purification (Lifetein, USA). The affinity-purified anti-*I. ricinus* FMRFa_MS-L antibody was produced in guinea pigs using an antigen corresponding to the sixth mature peptide of the precursor protein (Leiftein, USA, Supplementary Fig. 6b, c). Secondary antibodies included goat anti-rabbit Alexa 488, goat anti-rabbit Alexa 594, goat anti-mouse Alexa 594 and goat anti-guinea pig Alexa 488 (Life Technologies) diluted 1:1000 in 0.5% Triton X-100 in PBS (PBST). Secondary antibodies (Life Technologies) included goat anti-rabbit Alexa 488, goat anti-rabbit Alexa 594, goat anti-mouse Alexa 594, and goat anti-guinea pig Alexa 488, all diluted 1:1000 in PBST. For double staining of *Ir*-mAChR-B with *Ir*-mAChR-A, SIFamide, or leucokinin (rabbit antisera), tissues were first incubated with the primary antibody, followed by goat anti-rabbit Alexa 594, then incubated overnight in PBST containing anti-*Ir*-mAChR-B directly labelled with NHS-fluorescein (ThermoFisher).

Negative controls (Supplementary Fig. 4, 5) included antibody pre-adsorption with 250 μg.mL$^{-1}$ antigen for 3 h at RT or a replacement with pre-immune serum. Anti-leucokinin antibody[40] (raised against cockroach leucokinin) was pre-adsorbed with *I. ricinus* leucokinin peptide (Synpeptide Co., Ltd, China). Immunocytochemistry of CHO cells transfected with *ir-machr-a* or *-b* compared to mock-transfected cells was used to verify *Ir*-mAChR-A and -B antibody specificity (Supplementary Fig. 4a, b). Stained tissue or cells were mounted in Prolong Antifade Diamond Mountant with DAPI (Life Technologies) and imaged using a Leica DMI8 confocal microscope (Leica Microsystems) with z-stacking. Z-stack composites were adjusted in Adobe Photoshop (Adobe Inc.).

### Immunostaining of sectioned tissue

Immunohistochemical staining of sectioned *I. ricinus* nymphs followed a standard paraplast embedding protocol. Legs and posterior idiosoma were removed before fixation in Bouin's solution for 3 h at RT. Samples were dehydrated and cleared through graded 70%, 96%, and 100% ethanol and chloroform, then embedded in melted paraplast (Leica, Biosystems). Sections (12 μm) were cut using a semi-automated Histocore Multicut microtome (Leica, Biosystems). Xylene-deparaplasted sections were rehydrated through graded 96% and 70% ethanol and distilled water, then blocked for 20 min with normal goat serum (Sigma, France). Sections were incubated overnight with anti-leucokinin (1:1000), followed by a 6 h incubation with HRP-labelled anti-rabbit antibody (New England Biolabs, Canada) at RT. Immunoreactivity was visualised using 3,3′-diaminobenzidine (Sigma, France) and $H_2O_2$ in Tris-HCl buffer (pH 7.6) under a dissection microscope. Images were acquired with a UC90 camera mounted on an Olympus BX53 microscope. Contrast and brightness were adjusted in Adobe Photoshop (Adobe Inc.).

### Nomenclature of the neuropeptidergic cells

Neuronal nomenclature for *I. ricinus* synganglion followed Šimo et al.[81]. The first two letters indicate the neuron's position within a specific lobe: protocerebral (Pc), pedal 1–4 (Pd$_{1-4}$), or opisthosomal (Os). Subsequent letters indicate anatomical orientation: dorsal (D), ventral (V), anterior (A), posterior (P), medial (M), lateral (L), or interneurons (IN). Neurons projecting axons to the syanganglion surface—a putative neurohaemal site—were designated as neurosecretory cells (NS). Neurons projecting axons into SG were designated as SG neurons. Numbers following the letter code indicate the neuron pair number.

## Molecular identification of *I. ricinus* FMRFa_MS-L neuropeptide transcript

The putative *I. ricinus* FMRFa-L_MS prepropeptide (GenBank: GIXL01007324) was identified by BLAST search using the *Rhipicephalus microplus* FMRFa-L_MS prepropeptide (GenBank: XP_037272417.1) against the *I. ricinus* NCBI Bioproject PRJNA657487. A 1324 bp fragment was amplified using primers 5′-AAGATGGTGAAGT-GATTAGGTA-3′ and 5′-ATTCGAAGACAGCTTCCTCGT-3′, cloned into the pGEM®-T Easy vector (Promega), and sequenced (Eurofins Scientific, France). The confirmed sequence was then used to design and generate an antibody against the precursor's mature peptide 6.

## Immunogold TEM

Synganglia from unfed female *I. ricinus* were fixed in 4% formaldehyde with 0.1% glutaraldehyde in 0.1 M HEPES for 1 h at RT. After washing in HEPES buffer, specimens were embedded into 15% gelatine, cryoprotected in 2.3 M sucrose for 72 h at 4 °C and frozen by plunging into liquid nitrogen. Ultrathin cryosections were cut at −100 °C, and collected using 1.15 M sucrose/1% methylcellulose (25 cp, Sigma). Sections were incubated 1 h at RT in 1% fish skin gelatin (FSG), then incubated with primary antibodies diluted 1:40 in FSG 1 h at RT, washed, then incubated 1 h with gold-conjugated secondary probes (BBI; 1:40 in FSG; see Supplementary Table 2). After HEPES washes, sections were post-fixed for 5 min in 1% glutaraldehyde diluted in 0.1 M HEPES, rinsed in distilled $H_2O$, and contrasted/embedded with 2% methylcellulose and 3% aqueous uranyl acetate (9:1). Samples were examined using a JEOL 1400 TEM. Control sections were processed without the primary antibody. See Supplementary Table 2 for a list of antibodies used, their dilutions, and the corresponding secondary antibodies.

Electron tomography was used to localise immunogold nanoparticles in specimens double-labelled with antibodies against FMRFamide and mAChR-A. Anti-FMRFamide antibody was selected over anti-FMRFa_MS-L due to its specific recognition of $PcLNS_{1,2}$ dorso-lateral axons, whereas the latter also labels surface-adjacent neurons (Figs. 4a, 3n and Supplementary Fig. 6d, f), potentially interfering with nanoparticle quantification near neuronal cell bodies. Tomography data were acquired by JEOL 2100 F TEM equipped with the Gatan K2 Summit direct electron detector controlled by SerialEM. Tilt series were collected from ±60° at 1° increments, with each image captured as a 2 × 2 map. Reconstruction and model generation were performed using Imodmop (IMOD package). Black-and-white (B/W) masks delineating individual cell areas were created by converting 3D model contours from tomograms using Imodmop (IMOD package). Coordinates of gold nanoparticle centres in reconstructed tomograms were detected using Findbeads3d (IMOD package). Nanoparticles within cell-specific B/W masks were counted in Matlab, and statistical distributions were generated. Detailed nanoparticle counts are provided in the Source Data.

## Serial block-face scanning electron microscopy

*I. ricinus* nymphs were fed on mice for 48 h. After removing the dorsal cuticle, the body was fixed in 2.5% glutaraldehyde in phosphate buffer for 24 h at 4 °C. Samples were stained sequentially (RT unless stated otherwise) with 2% $OsO_4$ for 2 h, 3% ferrocyanide for 2 h, 1% thiocarbohydrazide for 1.5 h, 2% $OsO_4$ for 3 h, and 1% uranyl acetate overnight at 4 °C. After each staining step (except the first osmium staining), the samples were washed three times for 15 min. Samples were dehydrated in an acetone series and infiltrated with 33%, 50%, 66% acetone/resin mixtures for 2 h each, then infiltrated with 100% Hard Resin Plus 812 (EMS) overnight and polymerised at 62 °C for 48 h. Resin-embedded blocks were trimmed and imaged using MAPS software on an Apreo SEM with VolumeScope (Thermo Fisher Scientific). Serial images were acquired at 3 keV, 1.6 nA, 50 Pa with 20 nm resolution, 140 nm slice thickness, and 1 μs dwell time per pixel. Image data were processed using TrakEM2[82], Microscopy Image Browser[83], and Amira (Thermo Fisher Scientific).

## Quantitative real-time reverse transcriptase PCR (qRT-PCR)

Ten dissected synganglia and ten pairs of SG per time point were pooled for RNA extraction using the RNeasy Micro Kit (Qiagen). qRT-PCR was performed on a LightCycler 480 II (Roche) with the LightCycler 480 SYBR Green I Master (Roche). The ribosomal protein S4 (GenBank: DQ066214) served as the reference gene[84]. Gene-specific primers were as described in Mateos-Hernández et al.[8]. Transcripts were quantified using the ΔΔCt method, corrected by amplification efficiency and expressed as fold differences[85] from three biological replicates. Amplicon sizes were verified by agarose gel electrophoresis, and *chat* and *vacht* amplicons from SG cDNA were sequenced (Eurofins; Supplementary Fig. 9).

## Acetylcholine detection assay

Pools of ten synganglia and ten pairs of SG from unfed, 1–6-day-fed, and replete *I. ricinus* females were dissected in ice-cold PBS, immediately frozen in liquid nitrogen and stored at −80 °C. Choline and ACh levels in tissue homogenates were quantified using the Choline/Acetylcholine Assay Kit (Abcam, Cat. No. ab65345) following the manufacturer's instructions. Each pooled sample was analysed in duplicate on 96-well plates and read using a Glomax Discover microplate reader (Promega). Assays were conducted in three independent biological replicates from separate feeding experiments.

## In vivo fluid secretion assay

Five-day mouse-fed *I. ricinus* females were secured ventral side up on water-resistant double-sided tape (Scotch 3M), and their hypostomes were inserted into calibrated capillary micropipettes (volume 2 or 3 μL, length 55 mm, Drummond MicroCaps) for saliva collection. Ticks were injected ventrally near the third leg pair with cholinomimetic agents using a Nano-Injector pump (Drummond) connected to a Micro 4 controller (World Precision Instruments). All agonists and antagonists were administered at 5 mmol·kg⁻¹ of tick body weight, an effective dose for *I. ricinus* (Supplementary Fig. 11a, b) and a tenfold higher dose than that reported for *Amblyomma hebraeum* by Kaufman[11]. In specific tests, lower doses (1.25 mmol·kg⁻¹) of the antagonist propiverine were used. Drugs were dissolved in sterile water with injection volumes ranging from 50–200 nL·mg⁻¹ body weight. For agonist assays, saliva secretion was monitored for one hour post-injection by marking the saliva level in the micropipette at various timepoints under a Nikon SMZ800 stereomicroscope. For antagonist assays, ticks were first injected with the antagonist, rested for 5 min, and then injected with the agonist. Negative controls (water injections, 50–200 nL·mg⁻¹ body weight) did not elicit secretion. For each treatment condition, saliva from individual ticks (Fig. 7k) was pooled and subsequently processed for proteomic analyses.

## Proteomic analyses

Saliva from each experimental group was directly subjected to in-solution digestion without prior protein extraction, as described in Kozelková et al.[86]. Samples were reduced with 10 mM 1,4-dithiothreitol (DTT) at 56 °C for 45 min and alkylated with 55 mM iodoacetamide at RT in the dark for 20 min. Alkylation was subsequently quenched with 50 mM DTT. Proteins were digested using trypsin (Pierce Trypsin Protease, MS Grade; Thermo Fisher Scientific, #90057) at a ratio of 50:1 (protein:trypsin) overnight at 37 °C. Digestion was terminated by the addition of formic acid to a final concentration of 2.5% (v/v). NanoLC–MS/MS analysis was performed as described in Kozelková et al. (2023)[86]. Briefly, peptides obtained after overnight trypsin digestion were purified using StageTip C18 solid-phase extraction discs (Rappsilber et al., 2007). Purified peptides were dissolved in 30 μl of 3% acetonitrile/0.1% formic acid. Peptide separation was carried out on an UltiMate 3000 RSLCnano system (Thermo Fisher Scientific) online-coupled to a timsTOF Pro mass spectrometer (Bruker Daltonics). An aliquot of 2 μL of peptide solution was loaded onto an

Acclaim PepMap 100 C18 trapping column (300 µm i.d., 5 mm length, 5 µm particles, 100 Å pore size; Thermo Fisher Scientific, #160454) at a flow rate of 2.5 µL min⁻¹ with 2% acetonitrile/0.1% formic acid for 2 min. Peptides were then eluted onto an Acclaim PepMap 100 C18 analytical column (75 µm i.d., 150 mm length, 2 µm particles, 100 Å pore size; Thermo Fisher Scientific, #164534) and separated using a 48-min linear gradient from 5% to 35% acetonitrile/0.1% formic acid at a constant flow rate of 0.3 µL min⁻¹. Column temperature was maintained at 35 °C. Data were acquired in PASEF mode with positive ion polarity. Electrospray ionisation was performed using a CaptiveSpray source (Bruker Daltonics) with a capillary voltage of 1500 V, dry gas flow of 3 L min⁻¹, and a dry temperature of 180 °C. Ions were accumulated for 100 ms, and 10 PASEF MS/MS scans were recorded per topN acquisition cycle. The ion mobility range ($1/K_0$) was set from 0.6 to 1.6 Vs cm⁻², and mass spectra were acquired over an m/z range of 100–1700. Polygon filtering was applied to exclude low-m/z singly charged ions. Target intensity was set to 20,000 to enable repeated precursor selection for PASEF MS/MS, after which precursors were dynamically excluded for 0.4 min. Collision energies ranged from 20 to 59 eV and were adjusted in five equally spaced steps across the ion mobility range. Protein identification and quantification were performed using MaxQuant software (v2.4.2.0) with the Andromeda search engine against the *I. ricinus* (UniProt, 21.08.2023; 61,426 entries) and mouse (UniProt, 27.11.2023; 87,808 entries) databases. Default MaxQuant parameters were applied as described in Kozelková et al.[86], including the TIMS-DDA search type and Bruker TIMS instrument settings. Trypsin/P was specified as the digestion enzyme with up to two missed cleavages allowed. Carbamidomethylation of cysteine was set as a fixed modification, while N-terminal protein acetylation and methionine oxidation were included as variable modifications. Minimum and maximum peptide lengths were set to 8 and 25 amino acids, respectively. Precursor mass tolerance was set to 20 ppm for the first search, and 10 ppm for the main search, and MS/MS fragment ion tolerance was set to 25 ppm. Peptide spectrum matches (PSMs) and protein identifications were filtered using a target–decoy strategy with a false discovery rate (FDR) of 1%. Label-free quantification (LFQ) was performed using the MaxQuant-integrated algorithm with a minimum ratio count of 1. Protein tables generated by MaxQuant were analysed using Perseus software (v1.6.14.0). Experimental conditions were analysed using one biological replicate, each measured in three technical replicates. Results were filtered to remove reverse hits, contaminants, and proteins identified only by modified peptides. Proteins identified by a single peptide, with a score <40, or detected in fewer than two technical replicates were excluded. LFQ intensity values were $\log_2$-transformed, and principal component analysis (PCA) was performed using Perseus.

## Statistics and reproducibility

Statistical analyses were performed using GraphPad Prism 9. Data are presented as mean±SD. Box plots show the median and interquartile range (IQR), with whiskers indicating minimum and maximum inward ion current values for *Ir*-mAChR-A and -B stimulation; all individual data points are shown. Truncated violin plots were used for total saliva secretion, displaying the median and IQR. Drug potency in bioluminescence assays was determined by nonlinear regression, and half-maximal effective concentrations ($EC_{50}$) were calculated where applicable. Sample sizes (*N*, *n*) and statistical tests are specified in the figure legends, except where multiple *n* values are indicated directly in the figure. All *p*-values are reported with uniform precision to four decimal places, and values smaller than 0.0001 are denoted as $p < 0.0001$, with *p*-values shown in the figures.

For imaging-based analyses, multiple samples, sections, and fields of view were examined, and representative images were selected based on optimal tissue preservation, orientation, and signal quality.

Figure 3a–c, e, g–k show synganglia from 4 to 5 independent ticks; staining was reproduced in ≥4 independent experiments and yielded similar labelling patterns. Figure 3d shows five synganglia from two independent staining experiments, with comparable lateral nerve staining patterns in all specimens and lateral segmental organs positivity in three of them. Figures 3m, n, q, and 4a, b are based on ≥6 synganglia across two independent experiments, each producing similar results. Figure 4c is based on four synganglia processed in a single experiment and showing consistent staining. Figure 5a, b are based on three synganglia and three paired SGs, reproduced in two and ≥3 independent biological replicates, respectively, each yielding similar results. Figure 5c comprises two independent experiments, each containing three paired SG, all yielding similar results. Figure 5d is based on >15 ticks per developmental stage across >3 independent biological replicates (three unfed and at least three different tick-rabbit feeding experiments); numbers of positive/negative SGs for individual females are shown in Fig. 5e. Figure 5f is based on 10 independent SG pairs across two experiments, all yielding similar results. Reproducibility and sample sizes (*n*) for immunostaining in the Supplementary Figs. are provided in the corresponding legends.

SEM data in Fig. 3p originate from two independent experiments and show comparable synganglion and associated structural morphologies. TEM immunogold data in Fig. 3o derive from four independent ticks, with multiple ultrathin sections examined per replicate and similar ultrastructural and staining features observed. Figure 4e is based on two synganglia showing consistent features across multiple sections; quantification was performed on one representative axon-rich section, with counting details provided in the Source Data file.

## Reporting summary

Further information on research design is available in the Nature Portfolio Reporting Summary linked to this article.

## Data availability

All data generated or analysed in this study are included in the article and the supplementary information files. Mass spectrometry proteomics data have been deposited in the ProteomeXchange Consortium via the PRIDE[87] repository under dataset identifier PXD055362 (https://proteomecentral.proteomexchange.org). Source data are provided with this paper.

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

## Acknowledgements

This study was supported by the French National Research Agency (ANR-21-CE14-0012, project AxoTick; L.Š). UMR BIPAR was supported by the French Government's Investissement d'Avenir programme, Laboratoire d'Excellence Integrative Biology of Emerging Infectious Diseases (ANR-10-LABX-62-IBEID; institutional support). Additional support was provided by the Czech Science Foundation (22-18424M; J.P.), (22-30920S; R.Š.), (21-08826S; P.K.), and the Ministry of Health of the Czech Republic (NU20-05-00396 R.Š.). The study was also supported by a grant from the Centre-Val de Loire Region, under the Electro-CELL, Appel à projet d'intérêt regional (APR IR 21060LBL; S.H.T.). We acknowledge the BC CAS core facility LEM supported by MEYS CR (LM2023050 Czech-BioImaging and OP VVV CZ.02.1.01/0.0/0.0/18_046/0016045; M.V.). We are grateful to Jan Veenstra from the University of Bordeaux, France, for providing the anti-SIFamide and anti-leucokinin antibodies.

## Author contributions

L. Šimo designed the study and supervised all work. C.N., L.M.-H., S.R., L.A.-D., L. Šofranková, K.B., A.C., E.T., H.F., V.U., T.K., F.D., O.H., R.Š., J.T., T.B., M.T., M.V., J.P., S.H.T., and L. Šimo performed the experiments and analysed and interpreted the data. J.J.V. performed the molecular dynamics analyses and wrote this part of the manuscript. N.H., M.S., H.A., P.K., and J.P. provided resources. L. Šimo prepared the final figures and wrote the manuscript. All authors revised and approved the final version.

## Competing interests

The authors declare no competing interests.
