## [Transparent Peer Review file · Nature Communications]

Two Types of Axonal Muscarinic Acetylcholine Receptors Mediate Formation of Saliva Cocktail in the Tick *Ixodes ricinus*

Corresponding Author: Dr Ladislav Simo

Version 0:

Reviewer comments:

Reviewer #1

(Remarks to the Author)

The manuscript by Ning et al. is well-written. The major finding here is that saliva secretion in *I. ricinus* females is mediated by two different mACh receptors. Interestingly, pilocarpine is widely used in tick salivation, but the data presented here suggest that it exclusively activates mACh-A receptors and not the B-type. How important is the mACh-B receptor for saliva production and tick feeding? The data suggest half of the saliva is likely produced due to mACh-B receptor activity. It is also shown that mACh-B is present throughout the feeding in both acini type II and III (immunostaining), so one would assume that the change in saliva composition might be due to mACh-B receptor activity. However, the proteome and protein concentration results at a single time point during feeding show equal amounts of proteins (more in mACh-A induced) secreted in saliva. Interestingly, ~15% of the proteins were identified as the mouse host proteins, but there was no discussion about where these proteins came from. The likely reason is that since ticks were partially fed (on mice?), these proteins were not saliva proteins but acquired from the host. Since the proteins were not distinct in two groups, it could either be that both mAChR-A and mACh-B work together to produce a full complement of saliva proteins, or it could be that the acini were already stimulated and were making proteins because of the feeding on the host. Artificial feeding systems might help tease apart the actual portion of saliva proteins secreted due to activating one or the other type of receptor. While the authors show that choline and ACh are present in SG throughout feeding, the data have such high variations that they are challenging to interpret. The methodology is sound and detailed.

"These molecular dynamic results predict that mutating four Ir-mAChR-B substitutions to those of Ir-mAChR-A will negatively affect the atropine agonistic effect, whereas mutating Ir-mAChR-B to four active sites of Ir-mAChR-A should intensify atropine affinity for the receptor and thus antagonise its activity" This sentence needs to be re-written for clarity.

Reviewer #2

(Remarks to the Author)

Ning et al. characterize two distinct muscarinic acetylcholine receptors (mAChRs), A and B, in the *Ixodes ricinus* tick. The authors find that the distinct responses of the two mAChRs to muscarinic agonists and antagonists mediate changes in saliva secretion and protein content. While the manuscript presents a substantial amount of data, the key take-home messages and overall significance are not clearly highlighted. Additionally, the text and figures require refinement of language and better organization for clarity. Finally, as a reviewer, I was asked to focus specifically on the *in vitro* data presented in Figure 1, and thus, most of this review concentrates on those aspects."

Major concerns:

1. Figure legibility is an issue throughout the manuscript. Figure 1 will be used as an example here. In figure 1a, the text is far too small to read. In Figure 1i the color choices for highlighting residues render them unidentifiable. There are also missing X-axis labels on the electrophysiology scale bar and there are symbols in the upper inset of figure 1g that are not explained in the figure legend.

Additionally, there are some inconsistencies between figures and the text. There are multiple occasions where supplementary figures that do not exist are referred to in text e.g. supplementary figure 1 d-f is referred to in text but there is no Supplementary figure 1f (Line 140). Some legends also indicate that standard errors have been used, however, in the

statistical analyses section of the methods it states that 'all data are presented as mean \pm standard deviation.' All figures should be revised to ensure that consistent, readable data with descriptive figure legends is presented.

2. The number of biological replicates is an issue in several instances. For example, the legend of Figure 1d reads: "The error bars indicate the standard error of two (left) and three (right) biological replicates." However, SD error bars should never be used for $n=2$ replicates. In my opinion, a minimum of $n=5$ biological replicates should be used generally, and $n=3$ should be considered an absolute minimum for presenting data with error bars. This is also an issue for Figure 5c.

3. Further clarification and references (where appropriate) are needed for some of the statements made. For example, line 184- 887 should be reworded to clearly indicate what mutations are being made, which protein they are being made to, and the predicted effect. Lines 503-505 present a confusing description of the experimental design. Line 997 refers to 'the ten synganglia and ten pairs of salivary glands at each time point'. Which experiment are these from- the acetylcholine detection assay, which is described in the next section?

For structural modelling and molecular dynamics sections, PDB IDs should be added where the published structure of Hs-M1 has been used (line 223, 809). Are the models of Ir-mAChR- A and -B presented here available anywhere? If so, the database should also be referenced. There are two different averages given for Hs-M1 WT (line 169 and 177). Why are they different?

In the results it is stated that 'SIFamide axons are also recognized by both anti-FMRFa and anti-Ir-FMRFa-MS-L antibodies' (line 384-387) but no comment on whether this affected experimental design, or the validity of any results is made. In the discussion, it is stated that 'it is highly probable that in this setting type III acini were not fully, if at all stimulated to secrete fluids' (line 687-688). Why is this?

Intermediate concerns:

1. In Figure 1b, the activation of mAChR-B appears very different from type A; what do the authors think about that? Furthermore, X oocytes are known to express muscarinic receptors, and the expression levels can vary substantially between batches and even between individual oocytes. Did the authors systematically test a fair number of un-injected oocytes for each batch (e.g. 5-10) to ensure that the observed signals are not due to endogenous expression? Did every single cell express mAChR-A and B and were batches with endogenous mAChR expression discarded?

2. While perhaps not a big issue, I am puzzled by the logic behind the variable thresholds for 2* and 3* significance in several figures! This comes across a bit misleading and is really not necessary. Suggest altering this to standard thresholds (** $p < 0.01$; *** $p < 0.001$). Alternatively, the authors can insert the actual p values instead of stars; that is strictly speaking more correct anyway.

Issues/questions mostly related to Figure 1:

Fig 1 general – the layout of this figure requires high resolution and enlargement on a computer screen. With the resolution provided in the pdf file many parts are very hard to read on a computer screen and it is impossible to read when printed.

Suggest significant changes to layout such that no font is smaller than e.g. 7pt in final layout!

Fig 1a – not fond of the circular cladogram representation, as it cannot be read in most cases. This type of representation only works if it is high resolution and can be enlarged substantially. Suggest another representation of the key information.

Fig 1b – seems to be missing the indication of time on the scale bars.

Fig 1c – "Average currents evoked in oocytes from the same batches." You mean different batches? Individual datapoints should be included.

Fig 1d – Regarding the X-axis scale Fig 1d and e, what is "lg"? I assume it is logarithm base 10 (not base 2 that lg is sometimes used to represent) and suggest the use of "Log" or scientific scale.

Fig 1f – very hard to read due to small size. Furthermore, the representation is unnecessarily complex – why not put (%) as unit on the top and avoid writing % twice in each cell? Also, is it meaningful to include two decimal points? Is 0.26% measurably different from 0? Finally, the number of significant digits should be standardized (e.g. 111.09% has 4 significant digits while 0.26% only has 1). Finally, what is the logic behind the order of drug listing?

Fig 1g – not sure I can identify atropine in the binding site given the resolution and choice of colours!

Fig 1i – impossible to read the amino acids highlighted with black due to choice of colours!

Fig 1j - individual datapoints should be included.

Fig 1j and k – two biological replicates is not sufficient to represent data with error bars!

Fig 6b-e – "...as shown by the dots, square, diamond, and triangle symbol number in panels (f-j)." There are no square and diamonds in 6f, and the text needs correction!

Fig 6f-j – why use ANOVA for 6f and Students t-test for g-j?

Minor issues:

Line 159, GPCR TMI, which are. Otherwise, the previous sentence requires a reference.

Line 211- For protein sequences see supplementary dataset 1.

Line 266- name 'those' peptides

Lin 286- there is no reason provided for kinin to be investigated.

Line 291- replace 'NCS' with 'NSC'

Line 296- delete extra 'the'

Line 300- we queried whether if... Do you mean speculate or propose?

Line 392-393 – sentence should be rearranged, replace 'then' with 'than'

Line 430- delete 'and not seen in type I'.

Line 433- generally significantly? Consider rewording.

Line 438- what is meant by mRNA messages?

Line 597 – replace ‘whether’ with ‘that’, replace ‘NCS’ with ‘NSC’
Line 731- replace ‘laevis’ with ‘Iaevis’
Line 777 and 778 – In the cAMP assay,
Line 791- change ‘along’ to ‘with’
Line 797-798, 800– followed by the immediate...
Line 896- the room temperature acronym (RT) is defined after it has already been used multiple times-
Line 904 - 3,3'-Diaminobenzidine
From 920- ensure that an apostrophe (') is not used instead of prime (′)
Line 940- replace ‘3 with 3’
Line 965 – could be part of the previous heading
Line 989- replace ‘hours’ with ‘h’

Reviewer #3

(Remarks to the Author)

In this manuscript, the authors explored the involvement of muscarinic acetylcholine receptors (mAChRs) in the formation of saliva in the hard tick *Ixodes ricinus*. Using heterologous systems, the authors defined the pharmacological profile of *I. ricinus* mAChR type-B. The study uses an impressive array of techniques to describe and characterise the localisation of mAChR-A and -B in the synganglion and salivary glands of the ticks. Based on these data, the authors identified the peptidergic axons connecting central neurons with saliva-producing acini as the cholinceptive pathway regulating tick salivation, which is of particular importance for the specific research field. Finally, the authors performed a functional analysis of saliva secretion in ticks using different cholinergic agents. Overall, this is a well-organized and straight-forward study. That said, the rigor could be improved in specific experiments and some of the authors' conclusions would require additional consideration of the findings or stronger evidence. The manuscript would benefit from the following points:

1. In Fig. 1d, the authors showed that *I. ricinus* mAChR-B mobilises calcium in a CHO heterologous system after ACh stimulation with an EC₅₀ = 142.6 nM, in the presence of Gα₁₅(16). In their previous article (Mateos-Hernandez et al., 2020), they demonstrated that the ACh EC₅₀ for mAChR-A is 236 nM in the same cell model. This is quite surprising considering the rest of the data presented in this study and other literature. I would encourage the authors to show the values or relative values between mAChR-A and B from Fig. 1f and j and discuss the implications of these results.
2. It is not completely clear whether the authors are measuring calcium or cAMP in Fig. 1f, j and k. Please add the information in the figure legend.
3. In line 140, it indicates Supplementary Fig. 1d-f but there is no panel f in the document provided. Similarly, Supplementary Fig. 4f addressed in line 285 is missing.
4. The authors indicate that activation of mAChR-B does not inhibit forskolin-cAMP production (Supplementary Fig. 1d-e). However, ACh concentrations between 0.381-3.43 nM reduce 40% the relative luminescence (the graph colours for each concentration in d and e do not match; lowest cAMP production corresponds to 92.6 nM in d but 1.14 nM in e). Also, the range of concentrations chosen for these experiments is much lower than the one for the calcium mobilisation and the cAMP production. As the coupling of mAChR-B to Gi proteins is important for the authors' discussion, I would recommend them to support the data with additional experiments in the presence of pertussis toxin (PTX).
5. The authors report the p values in an unconventional manner (not exact values or *p < .05, **p < .01, ***p < .001). I wonder if the alpha value in their statistical analysis is different than 0.05.
6. Fig. 1g shows the structural analysis of Hs-M1 to *I. ricinus* mAChR-A and B according to the text and the figure legend, but in the methods section the authors describe that they use the *I. scapularis* mAChRs. According to Mateos-Hernandez et al., 2020, both species share 100% identity in the sequence of the mAChRs but the readers might not be aware of this. Please, provide this information in this manuscript or modify accordingly.
7. In Fig. 1j:
 - a. It is not clear why the cells expressing the full mutants mAChR-A and B require higher concentrations of all the agonists. Please, indicate if the cells expressing the WT receptors were also treated with 45 μM of the agonists for these results. If not, please provide a rational explanation for this concentration (ACh EC₈₀ for the mutants?).
 - b. In order to compare the response of full mutants vs WT for each agonist, all the data in the graphs should be normalised to only one condition (WT ACh) or provide the comparison between full mutant and WT for ACh in a separate graph. Alternatively, the comparisons could be done among the different agonists for the same recombinant receptor. Otherwise, this panel could be misleading because according to the figure, full mutant mAChR-A is twice as sensitive to muscarine than the WT. However, from the graphs in Supplementary Fig. 3c-d, it can be presumed that the response of the full mutant to this ligand is ~40% less than the WT (although data with no atropine not shown). The same happens in the case of mAChR-B for arecoline (Supplementary Fig. 3e-f).
 - c. The authors should provide data showing the expression levels and subcellular distribution of the full mutants in the heterologous system, considering that they already possess specific tools to detect them.
8. Many of the observations in this manuscript depend on the antibodies against *I. ricinus* mAChR-A and B generated by the

authors. Therefore, the validation of these antibodies is critical and Supplementary Fig. 10 should be referred in the main text and not only in the methods section. The same applies to Supplementary Fig. 11.

9. Immunostaining images using mAChR-A and -B antibodies result in a pattern that labels whole axons and somas. GPCRs are usually located only at the plasma membrane or at the membrane and some dispersed intracellularly due to their trafficking. The authors could provide higher magnification images to show the distribution of the *I. ricinus* mAChRs and explain this subcellular localisation.

10. It is not clear what extra information Fig. 2m adds to the study that it is not already included in Fig. 2j and Supplementary Movie 1.

11. Fig. 3b shows co-detection with 2 antibodies raised in rabbit (mAChR-B and kinin). Please, provide more information how this experiment was conducted.

12. Last sentence from figure legend in Fig. 3 missing (Line 375).

13. Supplementary Fig. 5:

a. It is described in detail in the main text. The authors should consider moving one of the graphs with the main observations to Fig. 3, , instead of, for example, the validation of anti-kinin.

b. The data presented would benefit from showing some staining that does not change with the treatment, like DAPI or other structural protein.

c. Longer periods of times or higher concentrations of ACh would reinforce the hypothesis of the authors.

d. It is puzzling why the authors used the anti-FMRFa instead of the anti-FMRFa_MS-L for this experiment and Fig. 3f.

14. Fig. 4b: the protocol for the co-immunostaining with the rabbit anti-mAChR-B and rabbit anti-SIFa is not valid and cross-detection will occur. Please, remove this image as it undermines the quality of the whole manuscript. The double staining can be repeated using the anti-mAChR-B antibody conjugated with fluorescein as described.

15. The authors claim that mAChR-A axon terminals are longer than the mAChR-B ones in type II acini. However, it is difficult to appreciate this observation with the images provided in Fig. 4. Please, add more data to support this conclusion.

16. Fig. 4c indicates the staining for mAChR-A and -B in the figure, while the labelling for β III-tubulin is only described in the figure legend. For consistency with the rest of the panels, please show it next to the image. Also, the authors state that mAChR-A and -B expressing axons connect to myoepithelial cells in the acini, but they do not use specific markers, and the morphology of cells is not clear in these images.

17. It is difficult to determine how the frequency analysis in Fig. 4e was performed or what it represents (number of positive axons/acini, intensity, area?). The analysis for mAChR-B probably should show acini type II and III in separate graphs.

18. In order to conclude that the free choline increases during feeding, please provide the statistics that support this statement in Fig. 5c.

19. The authors also suggest that free choline is produced in salivary glands based on the absence of it in acini I. Please, provide statistical analysis and/or more data to support this hypothesis.

20. Title in line 479 describes the induced SG secretion as an *in vitro* protocol (title) but in the methods section is named as an *in vivo* fluid secretion assay. Please, keep it consistent between both sections.

21. In Fig. 6, the authors assume that the remaining saliva production in ticks after scopolamine treatment in the presence of ACh or muscarine is due to mAChR-B action. However, atropine injection - which blocks only mAChR-A according to the findings in this article - effectively prevents all salivation. There is no evidence that this dose of scopolamine (5 mmol/kg) results in the complete inhibition of mAChR-A and the authors do not provide further information why they chose this concentration for all ligands. Furthermore, methacholine injection, which does not activate mAChR-B, reaches similar levels of saliva production than ACh and muscarine. The authors should clarify this and provide stronger evidence for a role of mAChR-B in the formation of saliva in ticks. This is particularly important as this conclusion is included in the title of the article.

22. In line 514 it indicates "raw data to be added". Please, check that this is what the authors meant to write here or clarify what this means.

23. The authors perform a proteomic analysis of the content of saliva induced by muscarinic agonists (Fig 6m-n). It is surprising that they do not discuss any of the specific proteins found in these experiments.

24. Reference 8 is sometimes referred in the text as Mateos-Hernandez et al., 2021 and other times as Mateos-Hernandez et al., 2020.

25. Provide a reference for the short turnover of ChAT in the discussion, line 621.

26. In the discussion, the authors explain that treatment with atropine induces paralysis in ticks, but this is not shown in the results section. They should provide evidence of this effect, and which other functions are affected by it. Additionally, they should test if scopolamine and propiverine produce similar toxicity.

27. The authors claim that activation of mAChR-B reduces saliva volume a 70% (line 660). However, the experiments presented in the manuscript do not support the specific activation of the type B receptor.

28. They authors should mention that nicotinic AChRs are probably not expressed in salivary glands, but they are in the synganglion (Lees et al., 2014; Le Mauff et al., 2020), and what implications this could have for some of their results (for example, Supplementary Fig. 5).

29. Line 1008 should be Supplementary Fig. 13 and not 12.

30. Some of the drugs listed in Supplementary Table 2 were not used in the study.

Version 1:

Reviewer comments:

Reviewer #2

(Remarks to the Author)

The revised manuscript by Ning et al. presents essentially the same characterization of two muscarinic acetylcholine receptors (mAChRs) in the *Ixodes ricinus* tick as the original submission. The initial manuscript received a substantial number of major and minor comments from three reviewers, and the authors have made a commendable effort to address these concerns. As a result, the revised version shows significant improvement. Nonetheless, certain aspects would benefit from further refinement, and I have outlined several concerns/recommendations.

Major concerns

1. The abstract does not effectively communicate the broader significance of this study to the general scientific community. For example, the final three sentences give the impression that the work only represents incremental advancements rather than offering novel or impactful insights.
2. Throughout the manuscript, the authors frequently rely on imprecise language that diminishes the clarity and impact of their findings. For example, in the sentence: "Pilocarpine, carbachol, arecoline, aceclidine, and methacholine poorly activated the full Ir-mAChR-A mutant, while bethanechol was ineffective," the terms "poorly" and "ineffective" are vague and lack quantitative or descriptive context. It is unclear what criteria are being used to define these levels of activation. The authors should critically assess their use of qualitative descriptors (adjectives) throughout the manuscript and revise for greater precision and consistency.
3. As noted in my initial review, the authors should carefully consider the level of precision reported across their assays and adopt a consistent standard for significant digits throughout the manuscript. For instance, reporting an EC₅₀ value of 682.8 nM for ACh (Fig. 1d) may not be meaningful given the apparent variability in the underlying data. Similarly, the rationale behind reporting p-values with differing levels of precision, e.g., 0.9706 versus 0.0005 (Fig. 4g), is unclear.
4. This manuscript is notably lengthy and would benefit from thorough proofreading, refinement, and substantial shortening to enhance clarity and conciseness. In many sections, unnecessarily verbose language is used to describe relatively straightforward concepts, which detract from the overall readability and impact of the work.
5. Several figures are unnecessarily complex and visually cluttered, which can obscure the key findings. A careful revision to identify and emphasize the most important points would greatly enhance clarity and improve the overall impact of the visual presentation.

Recommendations:

Fig 1a. While the enlarged version of the figure is easier to read, I still believe this type of figure is better suited for the supplementary material. A more compact and less cluttered panel could effectively convey the presence of Type A and B receptors. Additionally, it appears that the labels for Type A and B may have been inadvertently switched in the figure, which should be carefully reviewed and corrected if necessary.

Fig 1b. Raw maximal current amplitudes are not normally distributed and therefore should not be presented using SD or SEM. This issue is evident from the error bar for Oxo on Type A receptors, which extends into negative values - an implausible outcome for current amplitudes. Unless the data are appropriately transformed (e.g., via logarithmic transformation), it would be more appropriate to report variability using IQR, which better reflects the distribution of non-parametric data.

Fig 1d-f. Suggest that: (i) EC₅₀ values are expressed in μM with 1 significant digit (e.g. 0.68 μM); (ii) figure panels should be resized to allow for larger, more legible axis labels; (iii) The X-axis should be formatted using scientific notation to enhance clarity.

Fig 1g-h. It is not meaningful to present data as a mean \pm error when based on only N = 2 replicates. With such a limited sample size, measures of variability like SD or SEM are not statistically informative. Unless additional replicates are included, I recommend reporting the result as a single percentage value, rounded to the nearest whole number, without decimal points.

Fig 2f-g. Please double check this statement vs Fig 2f-g, as it appears that they are discussing two different things – potency (sensitivity) vs efficacy (relative luminescence (%)). “Under these conditions, all tested ligands effectively activated the Ir-mAChR-A WT, with approximately the same potency as ACh. However, the full Ir-mAChR-A mutant was twice as sensitive to the agonist muscarine as to ACh.”

Fig 2f-g. No error bars are shown for ACh, but given the variation in this assay, is this really a meaningful statement?

“Comparing full Ir-mAChR-B mutant activity with its WT form highlighted arecoline and muscarine as more potent agonists than ACh.”

Fig 2h-i. Similar comments as Fig 1d-f.

Fig 2f-i. The naming of the mutant constructs (e.g., “full mutant”) and the associated color scheme are not particularly intuitive and hinder understanding. The two “full mutants” are essentially reverse versions of each other, but this relationship is difficult to grasp from the figures alone without close reading of the main text. I recommend adopting a clearer and more descriptive naming convention for the two mutants, along with a more distinct and informative color scheme to help readers quickly differentiate and interpret the data.

Fig 4g. While the use of p-values is appropriate, it may not be necessary to display every possible statistical comparison, particularly those that are clearly non-significant. Including all comparisons can clutter the figure and detract from the key findings. I suggest streamlining the panel to emphasize the most relevant and meaningful statistical results, which would improve visual clarity and focus.

Fig 5e. Suggest displaying percentages in rounded numbers. Two decimal points are not meaningful and only adds to clutter.

Fig 6. Notice many errors in legend description.

Fig 6c-e. No p-values are presented, and given the substantial variability in the underlying data, their inclusion may not be meaningful in many cases.

Fig 7. In this reviewer’s opinion, the authors should consider what key message they want the reader to take away from this figure, rather than presenting all available data without prioritization. For example, panels 7b and 7c appear nearly identical, as do panels 7g through 7j. Streamlining the figure to highlight the most informative comparisons would improve clarity and help focus the reader’s attention on the main findings.

Fig 7l. Is every possible comparison important?

Reviewer #3

(Remarks to the Author)

I would like to commend the authors for their thorough and thoughtful revisions to the manuscript titled “Two Types of Axonal Muscarinic Acetylcholine Receptors Mediate Formation of Saliva Cocktail in the Tick *Ixodes ricinus*.” The authors have exposed their case convincingly and addressed all my previous concerns. The revised manuscript now presents a clearer (particularly Fig. 1 and 2) and more convincing case for the proposed receptor functions, supported by high-quality data and improved interpretation. The rigor of the experimental design has been strengthened, and the conclusions are now more robustly justified.

I believe this work is a valuable contribution and is suitable for publication in Nature Communications in its current form. Any remaining concerns I might have are minor and largely subjective. I believe it is more appropriate for the broader scientific community to engage with and discuss the authors’ findings in the published literature.

That said, I did notice a few typographical errors that should be corrected prior to publication, for example:

- Line 53: “... a a complex interplay”
- Line 109: “... discovered in the genomes”
- Line 213: “To investigatigate muscarinic regulation”
- Line 216: “... Both antibodies labeled distinct”
- Line 242: “Additionally, we found”
- Line 269: “... with greather abundance in the perineurium”
- Line 420: “... also Supplementray Dataset.”
- Lines 430 and 440: “Supplemnetary Dataset 2”

These are minor and can be easily addressed by the authors or editorial team.

In conclusion, I recommend this manuscript for publication.

Version 2:

Reviewer comments:

Reviewer #2

(Remarks to the Author)

In this R2 version of the manuscript by Ning et al., the authors have done a commendable job of proofreading and refining. The manuscript comes across more polished now and I have no further comments.

Point-by-point responses to reviewer comments

Reviewer #1 (Remarks to the Author):

1 The manuscript by Ning et al. is well-written. The major finding here is that saliva secretion in *I. ricinus* females is mediated by two different mACh receptors. Interestingly, pilocarpine is widely used in tick salivation, but the data presented here suggest that it exclusively activates mACh-A receptors and not the B-type. How important is the mACh-B receptor for saliva production and tick feeding? The data suggest half of the saliva is likely produced due to mACh-B receptor activity. It is also shown that mACh-B is present throughout the feeding in both acini type II and III (immunostaining), so one would assume that the change in saliva composition might be due to mACh-B receptor activity. However, the proteome and protein concentration results at a single time point during feeding show equal amounts of proteins (more in mACh-A induced) secreted in saliva.

Authors response:

We thank the reviewer for this valuable comment. We agree that the role of mAChR-B should be more clearly emphasized in the manuscript, as also noted by Reviewer #3 (point #24). To address this, we have revised the relevant section of the *Discussion* to highlight both the functional and morphological characteristics of the B-type receptor. Briefly, based on the activity patterns and morphology of the mAChR-B axon terminals, we propose that this receptor plays a basal role in maintaining readiness for the expulsion of type II and III acini. However, input from type A receptor axons appears to be necessary to initiate the full secretory cycle. For more details, please see Discussion Lines 537-598.

2 Interestingly, ~15% of the proteins were identified as the mouse host proteins, but there was no discussion about where these proteins came from. The likely reason is that since ticks were partially fed (on mice?), these proteins were not saliva proteins but acquired from the host. Since the proteins were not distinct in two groups, it could either be that both mAChR-A and mACh-B work together to produce a full complement of saliva proteins, or it could be that the acini were already stimulated and were making proteins because of the feeding on the host. Artificial feeding systems might help tease apart the actual portion of saliva proteins secreted due to activating one or the other type of receptor.

Authors response:

We appreciate the reviewer's insightful comments. Indeed, the saliva cocktail results from the coordinated activities of both A and B receptors. Regarding the host proteins, we consider them an integral part of the tick saliva and its biology, and we do not intend to separate them from the tick-derived components.

Specifically, the detection of mouse proteins such as neutrophilic granule protein (NGP), eosinophil-associated ribonuclease (Ear12), lysozyme (Lyz2), and tropomyosin in tick saliva samples provides important insights into the host response at the tick feeding site. Unlike classical plasma proteins such as albumin, which reflect abundant blood components, these proteins are typically localized within immune or structural cells and are not abundant in circulating blood under normal conditions. Their presence suggests localized immune cell activation, degranulation, and potential tissue damage in response to tick attachment and feeding. In particular, the detection of neutrophil- and eosinophil-derived factors indicates that innate immune effectors are mobilized to the bite site, even as the tick deploys salivary immunosuppressive molecules. This co-occurrence of host defense markers and tick salivary proteins underscores the dynamic interaction at the host-parasite interface and highlights how

proteomic analysis of tick saliva can indirectly reveal aspects of the host's early immune and tissue responses to parasitism. Previous studies have described this phenomenon:

<https://doi.org/10.1006/expr.2000.4567>,

<https://academic.oup.com/jime/article/42/3/359/849202>

Nevertheless, to address the reviewer's concern, we have included the following statement in the Results section:

"Moreover, the consistent detection of abundant host-derived proteins, such as haemoglobin, albumin, and serumtransferrin, as well as immune effectors such as neutrophilic granule protein and protein S100 (Supplementary Dataset 2) across multiple conditions highlights that components of the vertebrate haemostatic and immune system are not merely contaminants but constitute an integral part of the tick saliva proteome, reflecting dynamic interactions at the host-parasite interface."

3. While the authors show that choline and ACh are present in SG throughout feeding, the data have such high variations that they are challenging to interpret. The methodology is sound and detailed.

Authors response:

We agree that ACh levels in the salivary glands show considerable variation throughout the feeding course, as do the *chat* and *vacht* transcript levels. Our primary aim was to demonstrate that ACh is synthesized in the tick salivary glands. While we have a well-established system for tick feeding in the laboratory, we considered it valuable to examine the entire feeding course rather than focusing on a single or a few stages. Although the variation is indeed high, we believe it is important to present this data to the readers to reflect the biological dynamics during feeding. To address the reviewer's concern, we have removed any statements regarding statistical significance from the main text. In addition, we have included the following statement in the Discussion:

"Although we confirmed the presence of substantial amounts of ACh in I. ricinus SG, extensive efforts to localise cholinergic cells within this tissue were unsuccessful. This failure may stem from the rapid turnover of ChAT protein⁵², potentially explaining the scattered ACh signal and the high variability in chat and vacht transcript levels observed among individual ticks."

4. "These molecular dynamic results predict that mutating four Ir-mAChR-B substitutions to those of Ir-mAChR-A will negatively affect the atropine agonistic effect, whereas mutating Ir-mAChR-B to four active sites of Ir-mAChR-A should intensify atropine affinity for the receptor and thus antagonise its activity" This sentence needs to be re-written for clarity.

Authors response:

We thank the reviewer for pointing out this confusing statement. We have now revised the two sentences that read as follows:

"On the one hand, these in-silico results predict that mutating the four conserved active binding site residues of Ir-mAChR-A (i.e., Hs-M1 structurally homologous residues Y106, S109, A193, and N382; Fig. 2a-c) to the substitutions of Ir-mAChR-B (i.e., Y106H, S109C, A193I, and N382H) will destabilize atropine within the active site of Ir-mAChR-A, consequently reducing its affinity (Fig. 2e). On the other hand, mutating the Ir-mAChR-B active site to the four conserved residues of Ir-mAChR-A/Hs-M1 should increase atropine affinity with Ir-mAChR-B."

Reviewer #2 (Remarks to the Author):

Ning et al. characterize two distinct muscarinic acetylcholine receptors (mAChRs), A and B, in the *Ixodes ricinus* tick. The authors find that the distinct responses of the two mAChRs to muscarinic agonists and antagonists mediate changes in saliva secretion and protein content. While the manuscript presents a substantial amount of data, the key take-home messages and overall significance are not clearly highlighted. Additionally, the text and figures require refinement of language and better organization for clarity. Finally, as a reviewer, I was asked to focus specifically on the *in vitro* data presented in Figure 1, and thus, most of this review concentrates on those aspects."

Major concerns:

1 Figure legibility is an issue throughout the manuscript. Figure 1 will be used as an example here. In figure 1a, the text is far too small to read. In Figure 1i the color choices for highlighting residues render them unidentifiable. There are also missing X-axis labels on the electrophysiology scale bar and there are symbols in the upper inset of figure 1g that are not explained in the figure legend.

Authors response:

We thank the reviewer for these valuable suggestions, which have helped us improve the clarity and readability of the figures. To address these comments, we have split the original Figure 1 into two separate figures (Fig. 1 and Fig. 2), which allowed us to enlarge individual panels for better visualization. In addition we have increased the size of the text and converted all figures into vector graphics to ensure high resolution and improved readability. In the new Figure 2c (formerly Figure 1i), we have adjusted the colors highlighting the residues to provide better contrast. Furthermore, we have added X-axis labels (Fig. 1c) to the electrophysiology scale bars to enhance interpretability. Lastly, the symbols shown in the upper inset of Figure 2a (formerly Figure 1g) are now clearly explained in the figure legend. Please also note that new replications of the electrophysiological recordings were performed, and the figure was slightly changed accordingly.

2 Additionally, there are some inconsistencies between figures and the text. There are multiple occasions where supplementary figures that do not exist are referred to in text e.g. supplementary figure 1 d-f is referred to in text but there is no Supplementary figure 1f (Line 140).

Authors response:

We have corrected all figure cross-references throughout the entire manuscript.

3 Some legends also indicate that standard errors have been used, however, in the statistical analyses section of the methods it states that 'all data are presented as mean \pm standard deviation.' All figures should be revised to ensure that consistent, readable data with descriptive figure legends is presented.

Authors response:

The reviewer is correct, and we thank the reviewer for this comment. All statistical data are now presented as mean \pm SD, and the figure legends have been appropriately corrected.

4 The number of biological replicates is an issue in several instances. For example, the legend of Figure 1d reads: "The error bars indicate the standard error of two (left) and three (right) biological replicates." However, SD error bars should never be used for n=2 replicates. In my

opinion, a minimum of n=5 biological replicates should be used generally, and n=3 should be considered an absolute minimum for presenting data with error bars. This is also an issue for Figure 5c.

Authors response:

We thank the reviewer for this important point. We have repeated all experiments that had fewer than two biological replicates, and were pointed out by the reviewer. In particular, as requested, we repeated the experiments corresponding to the data presented in the original submission in Figures 1d, 1j, 1k, and 5c (now Figures 1d, e, 2f–i, and 6c, d in the revised version).

We strongly believe that three independent biological replicates are sufficient to calculate standard deviations (SD), as demonstrated in other reports published in this journal.

5. Further clarification and references (where appropriate) are needed for some of the statements made. For example, line 184- 887 should be reworded to clearly indicate what mutations are being made, which protein they are being made to, and the predicted effect.

Authors response:

We thank the reviewer for pointing out this confusing statement. We have now revised the 2 sentences that read as follows:

“On the one hand, these in-silico results predict that mutating the four conserved active binding site residues of Ir-mAChR-A (i.e., Hs-M1 structurally homologous residues Y106, S109, A193, and N382; Fig. 2a-c) to the substitutions of Ir-mAChR-B (i.e., Y106H, S109C, A193I, and N382H) will destabilize atropine within the active site of Ir-mAChR-A, consequently reducing its affinity (Fig. 2e). On the other hand, mutating the Ir-mAChR-B active site to the four conserved residues of Ir-mAChR-A/Hs-M1 should increase atropine affinity with Ir-mAChR-B.”

6. Lines 503-505 present a confusing description of the experimental design.

Authors response:

We completely agree with the reviewer and have rewritten the statement in the following clearer form:

“Saliva samples were collected from four experimental sets, each representing two distinct pharmacological conditions in which different cholinomimetic agents were used to mimic or inhibit the same mAChR-mediated response (Fig. 7k). Three of these conditions were subsequently analysed for protein content, while all four were included in principal component analysis (PCA) (Fig. 6l, m).”

7. Line 997 refers to ‘the ten synganglia and ten pairs of salivary glands at each time point’. Which experiment are these from- the acetylcholine detection assay, which is described in the next section?

Authors response:

We did not find it necessary to make any corrections, as the current statements are already clear. Specifically, the statement in pertains to the Quantitative real-time reverse transcriptase PCR (qRT-PCR) assay, while the statement in pertains to the Acetylcholine detection assay. In both experiments, pools of tick organs were used; however, the samples were collected from different ticks that had been feeding on rabbits.

8. For structural modelling and molecular dynamics sections, PDB IDs should be added where the published structure of Hs-M1 has been used (line 223, 809). Are the models of Ir-mAChR-A and -B presented here available anywhere? If so, the database should also be referenced.

Authors response:

We thank the reviewer for mentioning this oversight. We have now included the PDB accession number of Hs-M1 used in our study. The models of Ir-mAChR-A and Ir-mAChR-B presented in this manuscript are not publicly available.

“Both Ir-mAChR types share the conserved G protein-coupled receptor (GPCR) seven transmembrane (TM) fold (Supplementary Fig. 2) and are structurally similar to the human M1 receptor (Hs-M1, protein database accession: 6WJC)³³”

9. There are two different averages given for Hs-M1 WT (line 169 and 177). Why are they different?

Authors response:

We understand how the two averages could be misleading. The Hs-M1 WT averages were derived from two separate MD simulations. We now distinguish between the two simulations by using figure sub-labels (Fig. 2d, e). Additionally, we have adjusted the text (underlined below) to clarify that the averages are from two separate simulations:

“Performing separate MD simulations for each of the four Ir-mAChR-B divergent active site substitutions (Fig. 2a-c) depict that the most dissimilar substitution, A193I, causes the highest deviation of atropine within the Hs-M1 active site, 4.1 ± 0.6 Å, compared with the Hs-M1 wild type (WT) average, 2.0 ± 0.4 Å (Fig. 2d). The Y106H mutation causes similar fluctuations, with atropine deviations of 3.5 ± 0.3 Å. At ~300-ns MD, there is an apparent juxtaposed shift between these two mutations in coordinating atropine. The remaining two mutations, S109C (3.1 ± 0.4 Å) and N382H (2.6 ± 0.5 Å), also average higher than the Hs-M1 WT (Fig. 2d).

In subsequent MD simulations that include all four Ir-mAChR-B divergent active site substitutions (Fig. 2a-c), Hs-M1full mutant resulted in a higher average of 3.3 ± 0.3 Å after 15-ns MD, compared to the Hs-M1 WT average of 2.1 ± 0.3 Å (Fig. 2e).”

10. In the results it is stated that ‘SIFamide axons are also recognized by both anti-FMRFa and anti-Ir-FMRFa-MS-L antibodies’ (line 384-387) but no comment on whether this affected experimental design, or the validity of any results is made.

Authors response:

We thank the reviewer for pointing this out. As the role of neuropeptides was not investigated in the present manuscript, these findings did not affect any of the experiments. However, it was important to mention this observation for future studies. Furthermore, in the schematic diagram (Fig. 6a), it is clear that the axons react with all three antibodies: SIFamide, FMRFamide, and FMRFa-MS-L. Nevertheless, to address the reviewer's concern, we have modified the corresponding sentence in the Results section.

“Notably, SIFa-positive axons also exhibited reactivity to both anti-FMRFa and anti-Ir-FMRFa_MS-L antibodies, likely due to cross-reactivity related to the conserved C-terminal Phe-NH₂ motif (Supplementary Fig. 8a–c), although this conclusion requires further validation.”

Additionally, we have added the following statement to the legend of Supplementary Figure 8c:

“c Pre-adsorption of anti-FMRFa_MS-L with SIFamide peptide antigen completely abolished the immunoreactivity in SG innervation. The expression of SIFamide in PsSG neurons and their axons projecting to the tick SG was well characterised by proteomic and in situ hybridization approaches (Šimo et al., 2009⁵) and further confirmed in the current study (Fig.

5b in the main text). Our results suggest that both anti-FMRFa_MS-L and anti-FMRFa antibodies cross-react with the SIFamide peptide in axonal projections targeting type II and III SG acini. However, to confirm this conclusion, further investigation into the various classes of neuropeptides present in these peptidergic axon terminals and their physiological roles is required.”

11. In the discussion, it is stated that ‘it is highly probable that in this setting type III acini were not fully, if at all stimulated to secrete fluids’ (line 687-688). Why is this?

Authors response:

We have completely reworked the Discussion section—addressing the concerns raised by the other two reviewers—to better explain the role of type B receptors in salivation, as well as the proposed model involving both receptor types A and B. The previous statement has been removed. In the revised version, we now provide a comprehensive analysis of the functional and morphological characteristics of axons positive for each receptor type and attempt to link these features to their potential physiological roles, avoiding speculative interpretations.

Intermediate concerns:

12. In Figure 1b, the activation of mAChR-B appears very different from type A; what do the authors think about that?

Authors response:

This is an excellent point raised by the reviewer. It is indeed true that in *Xenopus* oocytes, receptor B appears to be less sensitive; however, this is not the case in CHO cells, where the responses of the two receptors are more similar. Although it is difficult to precisely predict the underlying mechanisms—and it is not the primary focus of the present manuscript—we have included the following sentences in the Discussion to address this point:

*“Although differences in ion current intensity between Ir-mAChR-A and Ir-mAChR-B were observed in *Xenopus* oocytes, our data confirmed the stimulatory nature of the previously identified Ir-mAChR-A⁸, and also suggested activating effects of Ir-mAChR-B. These differences likely reflect intrinsic receptor properties, such as ligand binding affinity and G-protein coupling efficiency, assessed in a simplified oocyte membrane environment. In contrast, the more similar EC₅₀ values measured in CHO cells for Ir-mAChR-B (current study) and Ir-mAChR-A⁸ may result from modulation by endogenous signalling pathways, membrane composition, and regulatory proteins, which can buffer intrinsic functional differences between receptors..”*

13. Furthermore, X oocytes are known to express muscarinic receptors, and the expression levels can vary substantially between batches and even between individual oocytes. Did the authors systematically test a fair number of un-injected oocytes for each batch (e.g. 5-10) to ensure that the observed signals are not due to endogenous expression? Did every single cell express mAChR-A and B and were batches with endogenous mAChR expression discarded?

Authors response:

Thank you for raising this important point, which is not always considered by non-electrophysiologists. For each recording series, we tested uninjected oocytes from 11 different batches. We can confirm that at the tested concentrations, acetylcholine (ACh), muscarine, oxotremorine, and methacholine did not induce any detectable currents in uninjected oocytes. This control *N* value is indicated in the Supplementary Fig. 12a.

Each oocyte was injected with cRNA encoding either mAChR-A or mAChR-B, but not both in the same cell. As indicated, at the tested concentrations, no endogenous currents masked or interfered with the responses of the *Ixodes* receptors. Therefore, only batches that failed to express *Ixodes* mAChR-A or mAChR-B were discarded.

14. While perhaps not a big issue, I am puzzled by the logic behind the variable thresholds for 2* and 3* significance in several figures! This comes across a bit misleading and is really not necessary. Suggest altering this to standard thresholds (** p<0.01; *** p<0.001). Alternatively, the authors can insert the actual p values instead of stars; that is strictly speaking more correct anyway.

Authors response:

As recommended, we have replaced the star symbols with the actual p-values for all statistical data.

15. Issues/questions mostly related to Figure 1:

Fig 1 general – the layout of this figure requires high resolution and enlargement on a computer screen. With the resolution provided in the pdf file many parts are very hard to read on a computer screen and it is impossible to read when printed. Suggest significant changes to layout such that no font is smaller than e.g. 7pt in final layout!

Fig 1a – not fond of the circular cladogram representation, as it cannot be read in most cases. This type of representation only works if it is high resolution and can be enlarged substantially. Suggest another representation of the key information.

Authors response: As stated in our response to the first comment (#1) from this reviewer, we have split the original Figure 1 into two separate figures (Fig. 1 and Fig. 2), allowing us to enlarge individual panels for better visualization. We have also increased the text size and converted all figures into vector graphics to ensure high resolution and improved readability.

16. Fig 1b – seems to be missing the indication of time on the scale bars.

Authors response: The time indication has been added to the scale bars.

17. Fig 1c – “Average currents evoked in oocytes from the same batches.” You mean different batches? Individual datapoints should be included.

Authors response:

This is a very good suggestion. We have added data points to all graphs throughout the manuscript. In this particular case, the differences between recordings for different drugs and batches are clearly explained in the Fig. 1 legend as follows:

“For mAChR-A, recordings (n = 19, 14, 18, and 15) were obtained from oocytes derived from six independent oocyte batches (N = 6). For mAChR-B, recordings (n = 8, 9, 8, and 8) were obtained from oocytes derived from seven independent batches (N = 7). In some instances, multiple recordings were acquired from individual oocytes.”

18. Fig 1d – Regarding the X-axis scale Fig 1d and e, what is “lg”? I assume it is logarithm base 10 (not base 2 that lg is sometimes used to represent) and suggest the use of “Log” or scientific scale.

Authors response: “lg” has been corrected to “Log” as suggested (in the current Figure 1d-f and 2hi Supplementary Fig. 3h).

19. Fig 1f – very hard to read due to small size. Furthermore, the representation is unnecessarily complex – why not put (%) as unit on the top and avoid writing % twice in each cell? Also, is it meaningful to include two decimal points? Is 0.26% measurably different from 0? Finally, the number of significant digits should be standardized (e.g. 111.09% has 4 significant digits while 0.26% only has 1). Finally, what is the logic behind the order of drug listing?

Authors response:

The quality of the figure has been enhanced by vectorizing the image. The (%) symbol has been removed, as it is unnecessary—the heat map color scale effectively conveys the data. All values are now presented with one decimal place for consistency. Additionally, the order of the drugs has been adjusted based on their highest response: 5 μM for agonists and 50 μM for antagonists, specifically for mAChR-A.

20. Fig. 1g – not sure I can identify atropine in the binding site given the resolution and choice of colours!

Authors response:

This is an excellent point, and we thank the reviewer for highlighting it. The colors and simulation figure have been adjusted to enhance the visibility and recognition of the binding sites (current Figure 2a, b).

19. Fig 1i – impossible to read the amino acids highlighted with black due to choice of colours!

Authors response: We agree and have adjusted the colors to improve readability. Current Figure 2c.

20. Fig 1j - individual datapoints should be included.

Authors response: The display of the figure has been revised (current Fig. 2f, g) based on the recommendation of the reviewer #3, and all data points have been included to all statistical data.

21. Fig 1j and k – two biological replicates is not sufficient to represent data with error bars!

Authors response: We repeated the experiments, and the data are now presented as three independent biological replicates. Fig. 2 f-i

22. Fig 6b-e – “...as shown by the dots, square, diamond, and triangle symbol number in panels (f–j).” There are no square and diamonds in 6f, and the text needs correction!

Authors response: Thank you for this point. The statistical descriptions in all relevant figure legends have been revised to include *N* numbers and detailed statistical information. The corresponding text has also been corrected. As a result, the previously incorrect statement has been removed. Current Fig. 7

23. Fig 6f-j – why use ANOVA for 6f and Students t-test for g-j?

Authors response: Thank you for this point. We have provided only the ANOVA test for panels f–j and it is clearly stated in the figure legend. (current Figure 7).

Minor issues:

24.

Line 159, GPCR TMI, which are. Otherwise, the previous sentence requires a reference.

This has been corrected.

Line 211- For protein sequences see supplementary dataset 1.

This has been corrected.

Line 266- name 'those' peptides

This has been corrected.

Line 286- there is no reason provided for kinin to be investigated.

We agree with this comment, and the *in situ* hybridization experiment for kinin has been removed from the manuscript, as it was extensive. However, we have retained the kinin staining on sectioned synganglia to support the reconstruction of the schematic drawing based on this staining and the Z-stack images (see legend in Fig. 4d), as explained in the figure caption.

Line 291- replace 'NCS' with 'NSC'

This sentence referred to the *in situ* experiment, which has been removed from the manuscript, so the sentence no longer exists.

Line 296- delete extra 'the'

This has been corrected.

Line 300- we queried whether if... Do you mean speculate or propose?

Current form of the modified sentence: *"This observation, prompted us to question whether neurosecretory cell-to-cell communication via neuropeptide signalling occurs through axon terminals within the synganglion perineurium, although this hypothesis requires further investigation."*

Line 392-393 – sentence should be rearranged, replace 'then' with 'than'

Current sentences: *Notably, mAChR-A axon terminals extend more apically within type II acini compared to the more basally restricted mAChR-B axon terminals (Supplementary Fig. 8d, e).*

Line 430- delete 'and not seen in type I'.

This has been corrected.

Line 433- generally significantly? Consider rewording.

Current form of the sentence: *"Overall, quantities of both molecules were consistently lower than those observed in the SG (free choline > ≈ 3 ng/tick and ACh > ≈ 1 ng/tick) than in SGs (Fig. 6d)."*

Line 438- what is meant by mRNA messages?

Sentence has been changed to: *"Given the absence of anterograde transport of vacht and chat mRNAs in cholinergic axons innervating type I acini⁸, the presence of these transcripts in SG tissue is unlikely to result from neuronal input and instead suggests local synthesis."*

Line 597 – replace 'whether' with 'that', replace 'NCS' with 'NSC'

This has been corrected.

Line 731- replace 'leavis' with 'laevis'

This has been corrected.

Line 777 and 778 – In the cAMP assay,

This has been corrected.

Line 791- change 'along' to 'with'

This has been corrected.

Line 797-798, 800– followed by the immediate...

This has been corrected.

Line 896- the room temperature acronym (RT) is defined after it has already been used multiple times-

This has been corrected.

Line 904 - 3,3'-Diaminobenzidine

This has been corrected.

From 920- ensure that an apostrophe (') is not used instead of prime (')

This has been corrected.

Line 940- replace '3 with 3'

The entire section of *in situ* hybridization technique has been removed from the manuscript.

Line 965 – could be part of the previous heading

We merged those as suggested

Line 989- replace 'hours' with 'h'

This has been corrected.

Reviewer #3 (Remarks to the Author):

In this manuscript, the authors explored the involvement of muscarinic acetylcholine receptors (mAChRs) in the formation of saliva in the hard tick *Ixodes ricinus*. Using heterologous systems, the authors defined the pharmacological profile of *I. ricinus* mAChR type-B. The study uses an impressive array of techniques to describe and characterise the localisation of mAChR-A and -B in the synganglion and salivary glands of the ticks. Based on these data, the authors identified the peptidergic axons connecting central neurons with saliva-producing acini as the cholinceptive pathway regulating tick salivation, which is of particular importance for the specific research field. Finally, the authors performed a functional analysis of saliva secretion in ticks using different cholinergic agents. Overall, this is a well-organized and straight-forward study. That said, the rigor could be improved in specific experiments and some of the authors' conclusions would require additional consideration of the findings or stronger evidence. The manuscript would benefit from the following points:

1 In Fig. 1d, the authors showed that *I. ricinus* mAChR-B mobilises calcium in a CHO heterologous system after ACh stimulation with an $EC_{50} = 142.6$ nM, in the presence of $G\alpha_{15(16)}$. In their previous article (Mateos-Hernandez et al., 2020), they demonstrated that the ACh EC_{50} for mAChR-A is 236 nM in the same cell model. This is quite surprising considering the rest of the data presented in this study and other literature. I would encourage the authors to show the values or relative values between mAChR-A and B from Fig. 1f and j and discuss the implications of these results.

Authors response:

We thank the reviewer for this comment. However, we do not fully understand which aspect of these experiments was considered surprising. If the concern relates to the difference in EC_{50} values between the A and B mAChR receptors, we would like to clarify that we do not view this as a significant issue. The EC_{50} values (148 nM vs. 236 nM) differ only slightly, and on a logarithmic scale, this difference is negligible. In other words in this system, the receptors exhibit very similar sensitivity to the ligand.

However as requested, all values from current Fig. 2f and g (original Fig. 1f and j) can be found in Source Data file. We hope this addition addresses the reviewer's concern, and we would be happy to provide any further clarification if needed. Nevertheless, as requested, for better clarity we have also included in the supplementary material (Suppl. Fig. 1g) the relative raw data (not normalised) responses for ACh (the ligand used for normalisation of all other ligand responses presented in Fig. 1f) responses for both receptor types A and B (the raw numbers are in the Source Data file).

Regarding the relative values, it is important to emphasize that the luminescent responses shown in this new supplementary figure—as well as in other parts of the manuscript, such as Supplementary Fig. 3d–g—are not directly comparable between the two receptors. Differences in relative signal intensity should not be interpreted as indicating higher sensitivity or greater activity of one receptor over the other. Luminescence values are influenced by multiple experimental variables, such as transfection efficiency, cell number (which can vary

slightly), incubation time with coelenterazine (the substrate for aequorin), and other procedural factors. Although consistent experimental conditions in 3 independent replicates were maintained, some variation is inevitable. Therefore, the EC_{50} values are the most reliable indicators of receptor activity. Same principle we also applied in explanation of some parts of reviewer point #8 below.

If the reviewer is referring to the differences in responses between the two receptors observed in the *Xenopus* system, we would like to note that we have addressed this point in response to Reviewer #2 (point #12). Specifically, we have included a corresponding discussion in the revised manuscript to clarify this issue.

“Although differences in ion current intensity between Ir-mAChR-A and Ir-mAChR-B were observed in Xenopus oocytes, our data confirmed the stimulatory nature of the previously identified Ir-mAChR-A⁸, and also suggested activating effects of Ir-mAChR-B. These differences likely reflect intrinsic receptor properties, such as ligand binding affinity and G-protein coupling efficiency, assessed in a simplified oocyte membrane environment. In contrast, the more similar EC_{50} values measured in CHO cells for Ir-mAChR-B (current study) and Ir-mAChR-A⁸ may result from modulation by endogenous signalling pathways, membrane composition, and regulatory proteins, which can buffer intrinsic functional differences between receptors..”

2 It is not completely clear whether the authors are measuring calcium or cAMP in Fig. 1f, j and k. Please add the information in the figure legend.

Authors response:

For all these data, we used the aequorin reporter system to measure intracellular calcium mobilization. As suggested by the reviewer, we have now indicated the assay type in the figure legends (current Fig. 1f and Fig. 2f-i).

3 In line 140, it indicates Supplementary Fig. 1d-f but there is no panel f in the document provided. Similarly, Supplementary Fig. 4f addressed in line 285 is missing.

Authors response:

The reviewer is correct. Panel f has been included in Supplementary Fig. 1 (new data were generated), and the same applies to original Supplementary Fig. 4 (now Supplementary Fig. 6), where panel f has been properly annotated.

4 The authors indicate that activation of mAChR-B does not inhibit forskolin-cAMP production (Supplementary Fig. 1d-e). However, ACh concentrations between 0.381-3.43 nM reduce 40% the relative luminescence (the graph colours for each concentration in d and e do not match; lowest cAMP production corresponds to 92.6 nM in d but 1.14 nM in e). Also, the range of concentrations chosen for these experiments is much lower than the one for the calcium mobilisation and the cAMP production. As the coupling of mAChR-B to Gi proteins is important for the authors' discussion, I would recommend them to support the data with additional experiments in the presence of pertussis toxin (PTX).

Authors response:

We thank the reviewer for this detailed observation. We initially overlooked this feature of the receptor and considered it an artefact. As suggested, the colors in the figure have been corrected accordingly. In the original submission, these data were based on only three technical replicates. Therefore, we repeated the experiment with forskolin stimulation and confirmed the same finding: Ir-mAChR-B mediates a slight inhibition of cAMP levels once activated by low dose of acetylcholine (Suppl. Fig. 1d). As this was an interesting addition and new observation (pointed out by reviewer), to exclude the possibility that the observed forskolin-mediated inhibition of cAMP was due to endogenous CHO cell receptors, we

performed control experiments using mock-transfected CHO cells (Suppl Fig. 1f). These experiments confirmed that the cAMP inhibition is specifically mediated by Ir-mAChR-B. Our results suggest that at low concentrations, acetylcholine (ACh) inhibits cAMP production, whereas at higher concentrations, the receptor shifts to activate the Gs signaling pathway. We hope the reviewer will appreciate that, although we did not use pertussis toxin as initially suggested, our repeated forskolin-based experiments adequately addressed this concern. The corresponding conclusions have been incorporated into the revised text as follows:

Results: “Interestingly, in the same reporter assay, activation of Ir-mAChR-B by low doses of ACh partially inhibited forskolin-mediated cAMP production (Supplementary Fig. 1d-f).”

Discussion: “The activation of the $G_{q/11}$ family in CHO cells by Ir-mAChR-B corroborates the existence of a downstream pathway also utilized by the GAR-1 and GAR-2 receptors in *T. spiralis* nematode²⁸. In addition, we observed that this receptor type can stimulate cAMP production in HEK cells, indicating activation of a second distinct signaling cascade. Conversely, lower ligand concentrations were associated with a reduction in cAMP levels. Collectively, these findings question the established view that all mAChR-B receptors couple exclusively to G_i proteins and function solely as inhibitory receptors²⁴. Previous studies have reported that G_i - or $G_{q/11}$ -coupled mAChRs may, under certain conditions, activate adenylyl cyclase and increase intracellular cAMP concentrations⁴³. Moreover, ligand concentration has been shown to influence the balance between Gs and G_i protein coupling in some GPCR systems⁴⁴. These observations underscore the need for further detailed studies to define the range of downstream signaling mechanisms engaged by these receptors under varying physiological conditions. Clarifying these signaling pathways will be essential for understanding the full functional repertoire of mAChR-B receptors.”

5 The authors report the p values in an unconventional manner (not exact values or *p < .05, **p < .01, ***p < .001). I wonder if the alpha value in their statistical analysis is different than 0.05.

Authors response:

As recommended by reviewer #2 (point #14), we have replaced the star symbols with the actual p-values for all statistical data throughout the manuscript.

6 Fig. 1g shows the structural analysis of Hs-M1 to *I. ricinus* mAChR-A and B according to the text and the figure legend, but in the methods section the authors describe that they use the *I. scapularis* mAChRs. According to Mateos-Hernandez et al., 2020, both species share 100% identity in the sequence of the mAChRs but the readers might not be aware of this. Please, provide this information in this manuscript or modify accordingly.

Authors response:

We thank the reviewer for this important point. We have modified the text and replaced *I. scapularis* with *I. ricinus* in the structural analysis sections. In the Fig. 1a (phylogeny) *I. ricinus/scapularis* and alignment shown in Figure 2c, "Ir/Is" indicates that the sequences from these two species are identical.

7. In Fig. 1j: **a.** It is not clear why the cells expressing the full mutants mAChR-A and B require higher concentrations of all the agonists. Please, indicate if the cells expressing the WT receptors were also treated with 45 μ M of the agonists for these results. If not, please provide a rational explanation for this concentration (ACh EC80 for the mutants?).

Authors response:

This is an excellent point. Both the WT and mutant forms were treated with 45 μ M agonist, and this has been indicated in the Fig 2 legend as well as in the main text as follows :

“For activation and accurate assay readings, both mutated and WT receptor form expressed in CHO cells (Supplementary Fig. 3c) were exposed to 45 μ M, the highest concentration in our dose- response assay for Ir-mAChR-B (Fig. 1e).”

8. Fig.1j: **b.** In order to compare the response of full mutants vs WT for each agonist, all the data in the graphs should be normalised to only one condition (WT ACh) or provide the comparison between full mutant and WT for ACh in a separate graph. Alternatively, the comparisons could be done among the different agonists for the same recombinant receptor. Otherwise, this panel could be misleading because according to the figure, full mutant mAChR-A is twice as sensitive to muscarine than the WT. However, from the graphs in Supplementary Fig. 3c-d, it can be presumed that the response of the full mutant to this ligand is ~40% less than the WT (although data with no atropine not shown). The same happens in the case of mAChR-B for arecoline (Supplementary Fig. 3e-f).

Authors response:

We highly appreciate this suggestion. As recommended, we now show different agonists for the same recombinant receptor (current Figure 2f, g).

Regarding Supplementary Figure 3 (currently panels d–g), (as we mentioned in the answer for the comment #1) the relative values are not directly comparable between WT and mutant forms. This is because each assay represents a different cell (flask) transfection, and only muscarine (3d, e) or arecoline (3f, g) was used in each case. Therefore, the values were not normalized to the ACh response, as was done in Figure 2f, g, in order to specifically observe differences in receptor responses to the individual ligands. In other words, ACh would need to be used as standard in each assay (Suppl. Fig. 3d-g) to accurately conclude whether the WT and mutant forms respond differently to particular ligands. However, the reviewer’s suggestion to rearrange the display in Figure 2f, g has clarified this issue.

8. Fig. 1j: **c.** The authors should provide data showing the expression levels and subcellular distribution of the full mutants in the heterologous system, considering that they already possess specific tools to detect them.

Authors response:

As suggested, we performed immunostaining of CHO cells expressing either mAChR-A or mAChR-B mutant forms and have included these results in the manuscript (current Supplementary Fig. 3c).

9. Many of the observations in this manuscript depend on the antibodies against *I. ricinus* mAChR-A and B generated by the authors. Therefore, the validation of these antibodies is critical and Supplementary Fig. 10 should be referred in the main text and not only in the methods section. The same applies to Supplementary Fig. 11.

Authors response:

As suggested, we have added references to the antibody validation (Supplementary Fig. 4 and 5) in the main text, and also in the legends for Figures 3, 4 and 5. Due to this inclusion, the order of the supplementary figures has been adjusted.

9. Immunostaining images using mAChR-A and -B antibodies result in a pattern that labels whole axons and somas. GPCRs are usually located only at the plasma membrane or at the membrane and some dispersed intracellularly due to their trafficking. The authors could provide higher magnification images to show the distribution of the *I. ricinus* mAChRs and explain this subcellular localisation.

Authors response: The reviewer is right and we agree that they are typically concentrated at the plasma membrane, with additional pools involved in intracellular trafficking. However, in our high-resolution TEM images (Supplementary Fig. 6a), mAChR immunoreactivity is clearly concentrated within axons and is not restricted to the plasma membrane. Although we are unable to provide definitive membrane-restricted staining, we emphasize that the predominant axonal localization is robust and reproducible across preparations. Moreover, it is likely that the imaged profiles represent axonal shafts rather than synaptic terminals, where membrane-bound receptors are typically concentrated, which may account for the limited detection of membrane-associated signal. This phenomenon was also observed in multiple specimens in our previous studies investigating dopamine and neuropeptide receptors in tick salivary gland axons (<https://doi.org/10.1038/s41598-019-43284-6>). To clarify, we have added the following sentence including previous study citation to the Results where immunogold labeling is first introduced:

“As confocal imaging of whole-mount preparations lacks membrane-level resolution, the predominantly intracellular Ir-mAChR localization in axons—confirmed by ultrastructural analysis and as shown previously¹⁰—is interpreted here as reflecting receptor trafficking. Most imaged profiles appeared to represent axonal shafts rather than synaptic terminals, which are presumably small and underrepresented in our sections, where membrane-bound receptors are typically concentrated.”

10. It is not clear what extra information Fig. 2m adds to the study that it is not already included in Fig. 2j and Supplementary Movie 1.

Authors response: Figure 2m shows axons arborizing within the synganglion lobes (as opposed to the surface projections shown in Movie 1 and current Figure 3n), originating from either PcMNS or PcLNS. This distinction is particularly important, as Figure 4d summarizes both surface and internal axons in the schematic overview. To enhance clarity, the following sentence has been added to the Figure 3 legend.:

“Double labelling with anti-mAChR-A and anti-MS. Z-stack projection highlights internal axonal networks of PcLNS_{1,2} (green; open arrowheads) and PcMNS₁₋₅ (red; filled arrowheads), arborising into distinct lobes.”

11. Fig. 3b shows co-detection with 2 antibodies raised in rabbit (mAChR-B and kinin). Please, provide more information how this experiment was conducted.

Authors response:

Thank you for pointing this out. The following information has been added to the Methods section:

“For double staining of mAChR-B with antibodies against either mAChR-A, SIFamide, or leucokinin (all raised in rabbits), the tissues were first incubated with the primary antibody

(anti-mAChR-A, anti-SIFamide, or anti-leucokinin), followed by incubation with a secondary goat anti-rabbit Alexa 594 antibody. The tissues were then incubated overnight in PBST containing the anti-mAChR-B antibody labeled with NHS-fluorescein (succinimidyl ester of 5/6-carboxyfluorescein), according to the manufacturer's protocol (Thermo Fisher)."

12. Last sentence from figure legend in Fig. 3 missing (Line 375).

Authors response: This has been corrected.

13. Supplementary Fig. 5:

a. It is described in detail in the main text. The authors should consider moving one of the graphs with the main observations to Fig. 3, , instead of, for example, the validation of anti-kinin.

Authors response:

We agree with the reviewer on this point. The validation of the kinin antibody by in situ hybridization (previous Figure 3d) has been removed from the manuscript, as also suggested by reviewer #2. However, we decided to retain the lateral view of the Ixodes synganglion stained with anti-kinin (current Figure 4c), as it helps readers understand how the schematic drawing (Figure 4d) was reconstructed. As further suggested, we have moved the previous Supplementary Figure 5c into the current Figure 4g.

14. Supplementary Fig. 5:

b. The data presented would benefit from showing some staining that does not change with the treatment, like DAPI or other structural protein.

Authors response:

This is an excellent suggestion and we have included this experimental data as suggested (Supplementary Fig. 7c).

15. Supplementary Fig. 5:

c. Longer periods of times or higher concentrations of ACh would reinforce the hypothesis of the authors.

Authors response:

We agree with this point; however, the aim of this experiment was solely to demonstrate that these axons are cholinceptive. While extended incubation times and the use of varying drug concentrations could yield additional insights, we believe that our straightforward, proof-of-concept approach is sufficient for the objectives of this study.

16. Supplementary Fig. 5:

d. It is puzzling why the authors used the anti-FMRFa instead of the anti-FMRFa_MS-L for this experiment and Fig. 3f.

Authors response:

This is an excellent detail that the reviewer has identified. Staining with FMRFa, shown in the current Figure 4e and Supplementary Fig. 7, was selected because of its ability to exclusively recognize PcLNS1 and PcLNS2 dorso-lateral axons. In contrast, the FMRFa_MS_L antibody also stains surface-adjacent neurons in this region (current Supplementary Figures 6d,f), which could negatively interfere with the measurement of fluorescent signal or nanoparticle counts in axons at the dorso-lateral surface of the synganglion. For this reason, we have provided an explanation in both the Methods section and the legend of Supplementary Figure 7.

Methods: *“Electron tomography was used to localize immunogold nanoparticles in specimens double-labeled with antibodies against FMRamide and mAChR-A. The choice of the anti-FMRamide antibody, rather than anti-FMRFa_MS-L, was motivated by its selective recognition of PcLNS_{1,2} dorso-lateral axons. In contrast, the latter antibody also labels surface-adjacent neurons in this region (Fig. 4a and Supplementary Fig. 6d, f), which could interfere with nanoparticle quantification in the area of neuronal cell bodies.”*

Suppl. Fig. 7 legend: *“Note that anti-FMRFa was used instead of anti-FMRFa_MS-L in this experiment, due its exclusively recognition of PcLNS_{1,2} dorso-lateral axons. In contrast, the anti-FMRFa_MS-L antibody also labels surface-adjacent neurons in this area (Fig. 4a in the main text and Supplementary Figure 6d, f), which would confound with accurate signal quantification at the dorso-lateral surface of the synganglion.”*

17. Fig. 4b: the protocol for the co-immunostaining with the rabbit anti-mAChR-B and rabbit anti-SIFa is not valid and cross-detection will occur. Please, remove this image as it undermines the quality of the whole manuscript. The double staining can be repeated using the anti-mAChR-B antibody conjugated with fluorescein as described.

Authors response:

We thank the reviewer for this important comment. As suggested, we have repeated this staining using the anti-mAChR-B antibody conjugated with fluorescein (current Figure 5b).

18. The authors claim that mAChR-A axon terminals are longer than the mAChR-B ones in type II acini. However, it is difficult to appreciate this observation with the images provided in Fig. 4. Please, add more data to support this conclusion.

Authors response:

We agree with the reviewer that the differential axon sizes were not clearly evident in the initially presented image. To address this, we have included an additional image with supporting statistical analysis (current Supplementary Fig. 8e), comparing axons in unfed and partially fed ticks. Additionally, current Figure 5c has been modified to improve clarity. As a point of note, we have been working with these axons for over 20 years and are well aware of the presence of both longer and shorter axons in type II acini, although this distinction is not always apparent through immunostaining. We hope that our additional experiment has clarified this concern.

19. Fig. 4c indicates the staining for mAChR-A and -B in the figure, while the labelling for β III-tubulin is only described in the figure legend. For consistency with the rest of the panels, please show it next to the image. Also, the authors state that mAChR-A and -B expressing axons connect to myoepithelial cells in the acini, but they do not use specific markers, and the morphology of cells is not clear in these images.

Authors response:

We thank the reviewer for this valuable point. In the current Figure 5c, we have removed the β III-tubulin staining to enhance clarity. However, to retain key information, we have included enlarged images (Supplementary Fig. 8d) showing type II and III acini stained with both anti-mAChR-A and anti-mAChR-B antibodies in combination with β III-tubulin.

Regarding the connection between mAChR-A and mAChR-B axons and the myoepithelial cells, our previous study (Vancová et al., 2019, <https://www.nature.com/articles/s41598-019-43284-6>) demonstrated that mAChR-A-positive axons (visualized with anti-PDF) and mAChR-B-positive axons (visualized with anti-SIFamide) form close contacts with myoepithelial cells

in type II and III acini, respectively. To clarify this connection, we have now cited this reference in the revised text:

“Moreover, as showed previously¹⁰and confirmed in our current study (Fig. 5d and Supplementary Figure 8d), within the type II and III acini, axon terminals from PcSG or OsSG_{1,2} neurons make close contact with a single myoepithelial cell in both type II and III acini, visualised by anti- β III-tubulin.”

20. It is difficult to determine how the frequency analysis in Fig. 4e was performed or what it represents (number of positive axons/acini, intensity, area?). The analysis for mAChR-B probably should show acini type II and III in separate graphs.

Authors response:

The current Fig. 5 legend has been updated to:

*“**Number** of individual female SG positive for anti-mAChR-A and anti-mAChR-B immunoreactivity in axon terminals in type II and III acini.”*

As the mAChR-B axons entering both type II and III acini originate from a single PcSG cell, we consistently either detect or do not detect immunoreactivity in both type II and III acini simultaneously. In other words, there is never a case where only type II axons are positive and type III are negative, or vice versa. For this reason, we prefer not to separate the graphs for type II and III acini, as they would be identical.

21. In order to conclude that the free choline increases during feeding, please provide the statistics that support this statement in Fig. 5c.

Authors response:

Here we use same answer as for the reviewer #1 regarding statistical significance for ACh detection in SG and synganglia. We agree that ACh levels in the salivary glands show considerable variation throughout the feeding course, as do the *chat* and *vacht* transcript levels. Our primary aim was to demonstrate that ACh is synthesized in the tick salivary glands. While we have a well-established system for tick feeding in the laboratory, we considered it valuable to examine the entire feeding course rather than focusing on a single or a few stages. Although the variation is indeed high, we believe it is important to present this data to the readers to reflect the biological dynamics during feeding. To address the reviewer’s concern, we have removed any statements regarding statistical significance from the main text. In addition, we have included the following statement:

Discussion: “Although we confirmed the presence of substantial amounts of ACh in I. ricinus SG, extensive efforts to localise cholinergic cells within this tissue were unsuccessful. This failure may stem from the rapid turnover of ChAT protein⁵², potentially explaining the scattered ACh signal and the high variability in chat and vacht transcript levels observed among individual ticks.”

22. The authors also suggest that free choline is produced in salivary glands based on the absence of it in acini I. Please, provide statistical analysis and/or more data to support this hypothesis.

Authors response:

We repeated the experiment (completely new tick feeding), as requested also by reviewer 2 (suggestion #4), and included statistical support for the statement as suggested by the

reviewer (current Fig. 6c).

23. Title in line 479 describes the induced SG secretion as an in vitro protocol (title) but in the methods section is named as an in vivo fluid secretion assay. Please, keep it consistent between both sections.

Authors response:

All salivation experiments were performed in vivo, and the text has been corrected as suggested.

24. In Fig. 6, the authors assume that the remaining saliva production in ticks after scopolamine treatment in the presence of ACh or muscarine is due to mAChR-B action. However, atropine injection - which blocks only mAChR-A according to the findings in this article - effectively prevents all salivation. There is no evidence that this dose of scopolamine (5 mmol/kg) results in the complete inhibition of mAChR-A and the authors do not provide further information why they chose this concentration for all ligands. Furthermore, methacholine injection, which does not activate mAChR-B, reaches similar levels of saliva production than ACh and muscarine. The authors should clarify this and provide stronger evidence for a role of mAChR-B in the formation of saliva in ticks. This is particularly important as this conclusion is included in the title of the article.

Authors response:

We truly thank the reviewer for highlighting what we believe is one of the most important aspects of this review. Indeed, accurately proposing the roles of both mAChR-A and mAChR-B is crucial, based on the findings of our study.

In our previous work, we attempted to silence both receptor genes via dsRNA injection but were unsuccessful in this approach (Mateos-Hernández et al., 2020). Therefore, pharmacological activation and inhibition currently represent the most straightforward and viable strategy for investigating receptor function.

As the reviewer correctly pointed out, despite considerable effort, we were unable to identify any mAChR-B-specific agonist. Thus, the only feasible way to study the function of mAChR-B was to selectively block mAChR-A and then apply a common non-selective agonist, such as acetylcholine (ACh) or muscarine. Two selective antagonists of mAChR-A—atropine and scopolamine—have been previously characterized in heterologous systems.

For drug dosing, we followed a previously published study by the Kaufman (1978) [10.1152/ajpregu.1978.235.1.R76](https://doi.org/10.1152/ajpregu.1978.235.1.R76), who used about 500 µmol/kg body weight of various salivation stimulants. We initially tested this concentration with pilocarpine and ACh. At this dose, pilocarpine triggered moderate salivation, but ACh had no effect. Although ACh had not previously been shown to stimulate salivation in ticks, we did not give up and increased the concentration of both agents tenfold (to 5 mmol/kg). This adjustment allowed us to induce salivation with ACh for the first time in tick research, and pilocarpine's effect was significantly enhanced. These preliminary trials are now included in the manuscript as Supplementary Figure 11a, b, and are described in the Results (lines 370-376) and Methods Line 904-908). We agree that testing a range of concentrations could be informative, but this would require substantial additional experimentation with uncertain outcomes, potentially extending the project by another year. Furthermore, muscarinic agonists are generally known for their toxicity, and increasing their doses further may compromise the physiological relevance of the experiments. We believe the selected concentration is optimal and sufficient, as it has been thoroughly tested in various drug combinations.

Although atropine blocked all salivation, we observed toxic effects associated with this compound, as discussed in detail in response to comment #29. Therefore, we cannot confidently attribute its effects to specific receptor antagonism. In contrast, scopolamine did not produce any noticeable toxic effects.

In this context, we respectfully suggest that the reviewer's statement, "There is no evidence that this dose of scopolamine (5 mmol/kg) results in the complete inhibition of

mAChR-A,” may overlook the fact that salivation induced by pilocarpine and methacholine—both mAChR-A-specific agonists—is nearly completely abolished by scopolamine (Fig. 7d, e and i, j). This strongly suggests that scopolamine effectively inhibits mAChR-A without observable toxicity.

Thus, our best available strategy to investigate the role of mAChR-B was to block mAChR-A using scopolamine, followed by stimulation with a non-selective agonist. We chose not to use pilocarpine for these specific experiments, as it is only a partial mAChR-A agonist, whereas methacholine demonstrated stronger and more complete agonist activity.

In addition, we have thoroughly revised the relevant section of the Discussion. We now more deeply analyze the morphological features of axons innervating different types of acini and connect these features to observed salivation patterns and protein concentrations, thereby proposing more physiologically grounded roles for the two receptor types.

Importantly, our findings are consistent with previous results from the Kaufman group, who reported that interruption of salivary gland (SG)-projecting axons affected salivation differently depending on receptor type (described in current study): disruption of mAChR-A-positive axons significantly reduced salivation, while disruption of mAChR-B-positive axons caused a less pronounced effect. In our own assays, activation of mAChR-A triggered robust salivation, whereas stimulation in the context of isolated mAChR-B function resulted in significantly lower salivation which is in a great alignment with Kaufman data. We have now addressed and thoughtfully discussed this observation in the revised.

In summary, in response to the reviewer’s comment:

- We clarified the drug concentrations used.
- We explained why scopolamine is currently the most suitable pharmacological tool to test the role of the B receptor.
- We incorporated both morphological and functional data to strengthen our hypotheses about receptor roles in the Discussion Lines 537-598.
- Since both receptors are localised in distinct axons innervating the salivary gland and their respective roles were investigated, we believe the current manuscript title sufficiently reflects the focus on both receptors in the context of axonal control of the tick salivary gland.

We sincerely hope these revisions satisfactorily address the reviewer’s concerns and that the proposed receptor functions are now more clearly justified by high-quality data and clear interpretation.

25. In line 514 it indicates “raw data to be added”. Please, check that this is what the authors meant to write here or clarify what this means.

Authors response:

We have removed this statement from the Results section and included the following sentence in the Proteomic Analysis section of the Methods:

“The mass spectrometry proteomics data have been deposited in the ProteomeXchange Consortium via the PRIDE¹ partner repository under the dataset identifier PXD055362.”

26. The authors perform a proteomic analysis of the content of saliva induced by muscarinic agonists (Fig 6m-n). It is surprising that they do not discuss any of the specific proteins found in these experiments.

Authors response:

We completely understand the reviewer's concern. Originally, the experiment was designed to identify mAChR-type-specific proteomes. However the overall diversity of tick salivary proteins across samples 1–6 (Fig. 7), representing various pharmacological stimulations, appears broadly comparable, with overlapping sets of secreted effectors including protease inhibitors, lipocalins, metalloproteases, and small immunomodulatory peptides. Rather than distinct,

stimulus-specific proteomes, the data suggest that different stimuli modulate the relative abundance of a shared salivary protein pool, reflecting a graded activation of overlapping acinar types rather than exclusive recruitment of specific glands. In contrast, samples 7+8, treated with common (both A and B receptors) antagonists, show a clear reduction in both protein diversity and secretion intensity, consistent with global suppression of salivary activity. This indicates that while salivary composition is responsive to receptor-specific input, its modularity lies more in quantitative tuning than in qualitative exclusivity. Together, these findings underscore a flexible, redundant salivary system in *Ixodes ricinus*, capable of adapting secretion output to different physiological states without fundamentally altering its effector repertoire.

Rather than overinterpreting the results, we have included a new statement in the Results section to clarify the inconclusive nature of the receptor-specific proteome:

Results:

“Among the eight experimental conditions, 17, 15, and 9 proteins were uniquely identified in saliva induced by ACh, methacholine, and pilocarpine, respectively. In ticks pre-treated with scopolamine, 15 and 9 proteins were uniquely present in following subsequent application of ACh and muscarine, respectively. These findings suggest that rather than producing distinct, stimulus-specific secretomes, different muscarinic inputs modulate the relative abundance of salivary proteinaceous constituents. This supports a model of activation across acinar cell types, allowing flexible secretion output”

Discussion:

“Additionally, although saliva volume and protein concentration can be broadly associated with the activity of type II or III acini, we were unable to directly link specific secreted proteins to individual acinus type. Furthermore, the noticeable differences in the saliva protein spectra elicited by drugs that either activate or inhibit the same type of mAChR suggest the involvement of distinct modes of action. To further elucidate these discrepancies, acinus-specific proteomics, combined with other omics approaches, would be highly beneficial for understanding the functions of different structural subunits within this tissue..”

27. Reference 8 is sometimes referred in the text as Mateos-Hernandez et al., 2021 and other times as Mateos-Hernandez et al., 2020.

Authors response:

This has been corrected throughout the manuscript.

28. Provide a reference for the short turnover of ChAT in the discussion, line 621.

Authors response: We have included the reference as suggested.

29. In the discussion, the authors explain that treatment with atropine induces paralysis in ticks, but this is not shown in the results section. They should provide evidence of this effect, and which other functions are affected by it. Additionally, they should test if scopolamine and piperidine produce similar toxicity.

Authors response:

As suggested, we now mention this effect in the Results section. Specifically, after atropine injection (an effect not observed with scopolamine), ticks stretched their front legs, began shaking, and showed signs of paralysis, although they remained alive. Therefore, we are

unable to conclusively determine whether the atropine-induced paralysis (as a mAChR-A blocker) prevented ACh- and muscarine-mediated salivary gland secretion. We believe this phenomenon is important to mention for future studies, even though atropine has been previously used to block pilocarpine-induced secretion. As noted, we did not observe similar paralysis with scopolamine (and saliva were analyzed). However, upon reviewing our notes, we found that propiverine also induced certain paralysis effect, although some ticks were still capable of salivating (Fig. 7h, j).

Based on these observations, the following statement has been added to the Results section:

“Nevertheless, it is important to note that following injection of either atropine or propiverine—but not scopolamine—ticks exhibited signs of paralysis, characterised by stretching and shaking of the front legs, and markedly reduced overall leg movement. At present, we cannot determine whether the administered doses of propiverine and atropine induced toxic effects on the tick organism, which may have also impacted saliva secretion.”

30. The authors claim that activation of mAChR-B reduces saliva volume a 70% (line 660). However, the experiments presented in the manuscript do not support the specific activation of the type B receptor.

Authors response:

The reviewer is right, and we have rephrased the original sentence as follows:

“In contrast, a dramatical reduction in saliva volume was measured when mAChR-A was specifically blocked and ticks were treated with a common agonist activating both receptor types. Thus, under these conditions, a lower saliva volume was observed, presumably due to the activity of mAChR-B, which was immunodetected in an axonal branch of the palpal nerve projecting to both type II and III acini.”

31. The authors should mention that nicotinic AChRs are probably not expressed in salivary glands, but they are in the synganglion (Lees et al., 2014; Le Mauff et al., 2020), and what implications this could have for some of their results (for example, Supplementary Fig. 5).

Authors response:

This is an excellent point raised by the reviewer. We have included the following two sentences in the Discussion:

“Furthermore, the involvement of neuronal nicotinic acetylcholine receptors (nAChRs) in synganglion function cannot be disregarded, as their expression has been documented in this organ^{46,47}. However, their precise cellular localisation within neuronal populations and their functional role in tick neurophysiology remain to be elucidated ”

“Nevertheless, it is worth mentioning that there is no published evidence for the expression of nAChRs in tick SG⁴⁶. At present, mAChRs remain the primary candidates for mediating cholinergic signalling in this tissue, though further experimental studies are necessary to confirm this.”

32. Line 1008 should be Supplementary Fig. 13 and not 12.

Authors response:

The entire order of the supplementary figures has been revised and thoroughly checked for accuracy.

33. Some of the drugs listed in Supplementary Table 2 were not used in the study.

Authors response:

The drug list has been corrected.

Point-by-point responses to reviewers

Reviewer #2 (Remarks to the Author):

The revised manuscript by Ning et al. presents essentially the same characterization of two muscarinic acetylcholine receptors (mAChRs) in the *Ixodes ricinus* tick as the original submission. The initial manuscript received a substantial number of major and minor comments from three reviewers, and the authors have made a commendable effort to address these concerns. As a result, the revised version shows significant improvement. Nonetheless, certain aspects would benefit from further refinement, and I have outlined several concerns/recommendations.

We are pleased that the reviewer acknowledges the substantial improvements made to the manuscript. However, we are somewhat puzzled that many of the new points raised were not identified during the initial round of review. The majority of these comments relate to stylistic, editorial, or presentation aspects, which are inherently subjective. Nonetheless, we recognise that several of the reviewer's observations are constructive, and we have addressed each point carefully and made corresponding revisions in the manuscript where appropriate.

Major concerns

1. The abstract does not effectively communicate the broader significance of this study to the general scientific community. For example, the final three sentences give the impression that the work only represents incremental advancements rather than offering novel or impactful insights.

We respectfully note that this issue was not raised during the first round of review. We appreciate the reviewer's perspective regarding the abstract. In response, we have completely rewritten the abstract to better highlight the broader significance of our study for the general scientific community, and we believe that this substantial revision adequately addresses the reviewer's concern.

2. Throughout the manuscript, the authors frequently rely on imprecise language that diminishes the clarity and impact of their findings. For example, in the sentence: "Pilocarpine, carbachol, arecoline, aceclidine, and methacholine poorly activated the full Ir-mAChR-A mutant, while bethanechol was ineffective," the terms "poorly" and "ineffective" are vague and lack quantitative or descriptive context. It is unclear what criteria are being used to define these levels of activation. The authors should critically assess their use of qualitative descriptors (adjectives) throughout the manuscript and revise for greater precision and consistency.

We respectfully note that this issue was not raised during the first round of review. We also respectfully disagree with the assertion that the language in the cited sentence is imprecise. In receptor pharmacology, such terminology is standard and widely understood. Nevertheless, to ensure absolute clarity, we have revised the text as follows:

Lines 156-163

“Muscarine activation of the *Ir*-mAChR-A4B approximately doubled the response compared to ACh, whereas carbachol, arecoline, aceclidine, and methacholine elicited responses less than half of the ACh response (<50%), and pilocarpine and bethanechol were almost inactive (<10%) (Fig. 2g). In contrast, both muscarine and arecoline evoked nearly twice that of ACh from the *Ir*-mAChR-B^{4A}. Strikingly, methacholine and oxotremorine—previously selective for *Ir*-mAChR-A (Fig. 1b, g)—elicited responses in the *Ir*-mAChR-B^{4A} nearly equivalent to ACh (~80–120%) (Fig. 2 h).”

In addition, we have revised other relevant sections of the manuscript to avoid any potential ambiguity or vague interpretation.

3. As noted in my initial review, the authors should carefully consider the level of precision reported across their assays and adopt a consistent standard for significant digits throughout the manuscript. For instance, reporting an EC₅₀ value of 682.8 nM for ACh (Fig. 1d) may not be meaningful given the apparent variability in the underlying data. Similarly, the rationale behind reporting *p*-values with differing levels of precision, e.g., 0.9706 versus 0.0005 (Fig. 4g), is unclear.

Although the reviewer indicated this as a major concern, we consider it primarily a cosmetic issue. We thank the reviewer for this comment and agree that the level of precision in reporting EC₅₀ values and *p*-values should be consistent and meaningful relative to the variability in the data. Accordingly, we have revised the manuscript to adopt a uniform standard for significant digits throughout, rounding EC₅₀ values and *p*-values appropriately to ensure clarity and consistency.

Regarding the *p*-values, we have added the following statement to the Methods, Statistical analysis section to clarify our reporting standard:

“All *p*-values are reported with uniform precision to three decimal places, and values smaller than 0.001 are denoted as $p < 0.001$, with *p*-values shown in the figures.”

4. This manuscript is notably lengthy and would benefit from thorough proofreading, refinement, and substantial shortening to enhance clarity and conciseness. In many sections, unnecessarily verbose language is used to describe relatively straightforward concepts, which detract from the overall readability and impact of the work.

We respectfully point out that this issue was not identified in the first round of review. Reviewer #1 described the manuscript as well written, and Reviewer #3 commented that it is a well-organized and straightforward study. Nevertheless, we agree that certain sections could benefit from additional proofreading and minor refinement to improve clarity and conciseness. Accordingly, the manuscript has been scientifically edited and shortened by more than 2,000

words, and thorough proofreading has been completed. As is evident from the tracked changes, no section meanings have been altered during this process.

5. Several figures are unnecessarily complex and visually cluttered, which can obscure the key findings. A careful revision to identify and emphasize the most important points would greatly enhance clarity and improve the overall impact of the visual presentation.

We note that in the first round of review, the reviewer's comments regarding complexity were raised only for Figure 1. In response to the reviewer's request, Fig. 1 was reorganized by dividing it into two separate figures, which improved clarity and was positively noted by Reviewer #3. Reviewer #3 also considered the manuscript to be a well-organized and straightforward study. While we do not agree that the remaining figures are overly complex, we acknowledge the reviewer's general concern regarding clarity, which to some extent is inherently subjective. We believe that the figures, although detailed, accurately and effectively convey the key findings without compromising readability and are fully consistent with *Nature Communications* presentation standards. Moreover, other papers published in *Nature Communications* include figures of comparable or greater complexity, indicating that our figures meet the journal's established standards for data presentation.

Recommendations:

Fig 1a. While the enlarged version of the figure is easier to read, I still believe this type of figure is better suited for the supplementary material. A more compact and less cluttered panel could effectively convey the presence of Type A and B receptors. Additionally, it appears that the labels for Type A and B may have been inadvertently switched in the figure, which should be carefully reviewed and corrected if necessary.

We do not agree that this phylogeny should be moved to the Supplementary Information, as it highlights the invertebrate clade of mAChRs described in this study and is clearly readable with a well-defined message. This phylogeny is central to our findings and forms the conceptual foundation for the entire manuscript, providing a baseline for understanding the evolutionary and pharmacological differences between the two receptor types. It is therefore particularly important to retain this figure in the main text, as it serves as a framework for interpreting the subsequent experimental results, which are relevant not only to tick research but also to the broader invertebrate field. We thank the reviewer for noting that the labels for Type A and B were switched and apologize for this oversight. The figure has now been carefully corrected.

Fig 1b. Raw maximal current amplitudes are not normally distributed and therefore should not be presented using SD or SEM. This issue is evident from the error bar for Oxo on Type A receptors, which extends into negative values - an implausible outcome for current amplitudes. Unless the data are appropriately transformed (e.g., via logarithmic transformation), it would be more appropriate to report variability using IQR, which better reflects the distribution of non-parametric data.

We thank the reviewer for this comment. We note that all current amplitudes reported are inward currents and thus negative by convention. Therefore, error bars extending to more negative values are entirely expected and biologically plausible, reflecting genuine variability in the responses. The apparent extension for oxotremorine on Type A receptor does not in any way affect the quality or interpretation of the data. To address the reviewer's concern and improve data presentation, we have revised the plots to display variability using interquartile range (IQR) representation instead of mean \pm SD.

Fig 1d-f. Suggest that: (i) EC₅₀ values are expressed in μ M with 1 significant digit (e.g. 0.68 μ M) We understand the reviewer's suggestion; however, we consider this a purely cosmetic detail that does not affect the scientific message or conclusions of the study. Nevertheless, to ensure consistency and clarity, we have revised all EC₅₀ values in Fig. 1d–f to be expressed in μ M with uniform precision (two significant figures; e.g., 0.68 μ M, 6.6 μ M), as recommended.

(ii) figure panels should be resized to allow for larger, more legible axis labels; (iii) The X-axis should be formatted using scientific notation to enhance clarity.

We do not agree with these suggestions. The figure panels have already been enlarged (three panels per page width) and are sufficiently legible, comparable to figures in other manuscripts published in this journal. The axis labels are within the size range recommended by the journal's formatting guidelines. Regarding the X-axis annotation, in the first round of review the reviewer specifically requested that we change the labeling from "lg" to "log," which we implemented. The current suggestion to switch to scientific notation is **entirely new** and, in our view, unnecessary. For receptor assays, a logarithmic X-axis is the standard and widely accepted method for presenting responses to varying ligand concentrations, and we believe the current format is both clear and appropriate.

Fig 1g-h. It is not meaningful to present data as a mean \pm error when based on only N = 2 replicates. With such a limited sample size, measures of variability like SD or SEM are not statistically informative. Unless additional replicates are included, I recommend reporting the result as a single percentage value, rounded to the nearest whole number, without decimal points.

We respectfully point out that this issue was not identified in the first round of review.

Regarding this table, the reviewer made clear recommendations in the first round (see below), and we followed them exactly, including simplifying the presentation as requested.

First review round suggestions regarding this Figure: Reviewer #2: - very hard to read due to small size. Furthermore, the representation is unnecessarily complex – why not put (%) as unit on the top and avoid writing % twice in each cell? Also, is it meaningful to include two decimal points? Is 0.26% measurably different from 0? Finally, the number of significant digits should be standardized (e.g. 111.09% has 4 significant digits while 0.26% only has 1). Finally, what is the logic behind the order of drug listing?

Authors response:

The quality of the figure has been enhanced by vectorizing the image. The (%) symbol has been removed, as it is unnecessary—the heat map color scale effectively conveys the data. All values are now presented with one decimal place for consistency. Additionally, the order of the drugs has been adjusted based on their highest response: 5 μ M for agonists and 50 μ M for antagonists, specifically for mAChR-A.

We note that new concerns have been raised at this stage. Nonetheless, we agree with the reviewer and now present the values as percentages, with the corresponding raw data provided in the Source Data file.

Fig 2f-g. Please double check this statement vs Fig 2f-g, as it appears that they are discussing two different things – potency (sensitivity) vs efficacy (relative luminescence (%)). “Under these conditions, all tested ligands effectively activated the Ir-mAChR-A WT, with approximately the same potency as ACh. However, the full Ir-mAChR-A mutant was twice as sensitive to the agonist muscarine as to ACh.”

We thank the reviewer for pointing this out. We agree that the distinction between potency and sensitivity should be clarified. We consider the first part of the statement correct and see no need for changes, as “All ligands activated the Ir-mAChR-A WT, with similar potency to ACh (equivalent to 100%)”. However, we agree that the second part should be revised. We have corrected the text to: “Muscarine activation of the Ir-mAChR-A^{4B} approximately doubled the response compared to ACh, whereas carbachol, arecoline, aceclidine, and methacholine elicited responses less than half of the ACh response (<50%), and pilocarpine and bethanechol were almost inactive (<10%) (Fig. 2g).”

Fig 2f-g. No error bars are shown for ACh, but given the variation in this assay, is this really a meaningful statement? “Comparing full Ir-mAChR-B mutant activity with its WT form highlighted arecoline and muscarine as more potent agonists than ACh.

We thank the reviewer for this comment. In these experiments, all ligand responses were normalized to ACh on each plate (set as 1) to account for inter-plate variability. This normalization strategy is a common and widely accepted approach in receptor pharmacology. Since ACh serves solely as the normalization reference, we do not show error bars for it. This method allows meaningful comparison of the relative potencies of other ligands across plates. Additional details have been added to the figure legend for clarity. For muscarine and arecoline, we agree that the previous statement may have been too strong; we have therefore softened the conclusion to: “In contrast, both muscarine and arecoline elicited activation of the Ir-mAChR-B^{4A} with potency equal to or greater than that of ACh.”

Fig 2h-i. Similar comments as Fig 1d-f.

We thank the reviewer for this comment. Regarding Fig 2h-i, we maintain that the panel sizes and X-axis labeling are appropriate and scientifically accurate, consistent with the explanation provided for Fig 1d-f and therefore do not need to be corrected.

Fig 2f-i. The naming of the mutant constructs (e.g., “full mutant”) and the associated color scheme are not particularly intuitive and hinder understanding. The two “full mutants” are essentially reverse versions of each other, but this relationship is difficult to grasp from the figures alone without close reading of the main text. I recommend adopting a clearer and more descriptive naming convention for the two mutants, along with a more distinct and informative color scheme to help readers quickly differentiate and interpret the data.

We thank the reviewer for the suggestion, although **this issue was not raised during the first round of review**. We have removed the term “full mutant” from the text and updated the nomenclature of the mutated receptors to *Ir-mAChR-A^{4B}* and *Ir-mAChR-B^{4A}*, clearly indicating the reciprocal mutations. The *in silico*-tested human *Hs-M1* mutant has also been renamed to *Hs-M1^{Ir4B}*, reflecting the incorporation of four *I. ricinus* mAChR-B substitutions. In addition, we have added a new panel (Fig. 2f) illustrating the mutagenesis process used to generate these constructs. We believe these revisions fully clarify the relationships between the receptor variants and address the reviewer’s concern.

Fig 4g. While the use of p-values is appropriate, it may not be necessary to display every possible statistical comparison, particularly those that are clearly non-significant. Including all comparisons can clutter the figure and detract from the key findings. I suggest streamlining the panel to emphasize the most relevant and meaningful statistical results, which would improve visual clarity and focus.

We understand the reviewer’s concern and would like to clarify that in the current figures, we did not include every possible statistical comparison, but only those relevant to the main message. For example, comparisons between 10- and 30-minute time points of the same treatment were omitted, and only comparisons between treated groups and the untreated control at the same time point were shown. To further address the reviewer’s comment, we have now retained only statistically significant comparisons. We also added the following note to the figure legend: “Only statistically significant comparisons ($p < 0.05$) from the ANOVA test are indicated.”

Fig 5e. Suggest displaying percentages in rounded numbers. Two decimal points are not meaningful and only adds to clutter.

Our aim was to present the data as precisely as possible. We appreciate the reviewer’s suggestion and have rounded the percentages accordingly to improve clarity.

Fig 6. Notice many errors in legend description.

We thank the reviewer for this point and will correct the spelling errors that occurred.

Fig 6c-e. No p-values are presented, and given the substantial variability in the underlying data, their inclusion may not be meaningful in many cases.

Regarding this issue, we respectfully note that we addressed it in our responses to Reviewers #1 and #3 during the first round of review:

We agree that ACh levels in the salivary glands show considerable variation throughout the feeding course, as do the *chat* and *vacht* transcript levels. Our primary aim was to demonstrate that ACh is synthesized in the tick salivary glands. While we have a well-established system for tick feeding in the laboratory, we considered it valuable to examine the entire feeding course rather than focusing on a single or a few stages. Although the variation is indeed high, we believe it is important to present this data to the readers to reflect the biological dynamics during feeding. To address the reviewer's concern, we have removed any statements regarding statistical significance from the main text. In addition, we have included the following statement in the Discussion:

"Although we confirmed substantial ACh levels in *I. ricinus* SG, we were unable to localise cholinergic cells, possibly due to rapid ChAT protein turnover⁵² consistent with scattered ACh signal and variable *chat* and *vacht* transcript levels across individuals."

Therefore, as we do not rely on statistical significance to interpret these data, we emphasize the high fluctuation levels of the molecules or transcripts studied, which reflect the inherent variability in the biological samples. In general, salivary gland proteins exhibit highly variable expression levels, both within and between individual ticks. Our intention was to present the true biological variability rather than to give the impression of precision through intentionally small error bars, which would not accurately represent the underlying biology. To clarify this point, we also added the following statement to the figure legend: "*Due to high variability among biological replicates, most comparisons in c-e were not statistically significant ($p \geq 0.05$).*"

Fig 7. In this reviewer's opinion, the authors should consider what key message they want the reader to take away from this figure, rather than presenting all available data without prioritization. For example, panels 7b and 7c appear nearly identical, as do panels 7g through 7j. Streamlining the figure to highlight the most informative comparisons would improve clarity and help focus the reader's attention on the main findings.

We respectfully point out that this issue was not raised in the first round of review. This figure represents the most biologically relevant results as the final outcome of the study, and we do not intend to change it. We do not agree with the reviewer's point regarding prioritization. The example cited by the reviewer actually illustrates our rationale: it first shows the dynamics of tick salivation over time (7b,c), and then clearly presents the total saliva secreted with all individual data points (7g, j). We believe this is a clear and coherent presentation of the data, and all supporting information is provided in the supplementary figures. Both datasets support each

other and jointly strengthen the interpretation of the results. Furthermore, we consider that the figure is not overly busy but appropriately detailed to convey the biological context accurately.

Fig 7l. Is every possible comparison important?

The reviewer suggests that we included every possible comparison, but this is not the case. For agonist effects, we presented only the relevant differences necessary to illustrate the pharmacological responses. For antagonist pretreatment, we included only comparisons with the representative agonist, as these are the comparisons necessary to interpret the effect.

Reviewer #3 (Remarks to the Author):

I would like to commend the authors for their thorough and thoughtful revisions to the manuscript titled "Two Types of Axonal Muscarinic Acetylcholine Receptors Mediate Formation of Saliva Cocktail in the Tick *Ixodes ricinus*." The authors have exposed their case convincingly and addressed all my previous concerns. The revised manuscript now presents a clearer (particularly Fig. 1 and 2) and more convincing case for the proposed receptor functions, supported by high-quality data and improved interpretation. The rigor of the experimental design has been strengthened, and the conclusions are now more robustly justified.

I believe this work is a valuable contribution and is suitable for publication in Nature Communications in its current form. Any remaining concerns I might have are minor and largely subjective. I believe it is more appropriate for the broader scientific community to engage with and discuss the authors' findings in the published literature.

That said, I did notice a few typographical errors that should be corrected prior to publication, for example:

- Line 53: "... a a complex interplay"
- Line 109: "... dicovered in the genomes"
- Line 213: "To investigatigate muscarinic regulation"
- Line 216: "... Both antibodies labeled distinct"
- Line 242: "Aditionally, we found"
- Line 269: "... with greather abundance in the perineurium"
- Line 420: "... also Supplementray Dataset."
- Lines 430 and 440: "Supplemnetary Dataset 2"

These are minor and can be easily addressed by the authors or editorial team.

In conclusion, I recommend this manuscript for publication. here are essential for interpreting the effects. In addition, the figure is not overly busy and effectively reflects the most important comparisons.

We are very pleased that this reviewer appreciates our responses. We agree with all of the reviewer's minor suggestions and have implemented those in the manuscript.